# The Limits of Transfer Reinforcement Learning with Latent Low-rank Structure

**Tyler Sam**
Cornell University
tjs355@cornell.edu

**Yudong Chen**
University of Wisconsin-Madison
yudong.chen@wisc.edu

**Christina Lee Yu**
Cornell University
cleeyu@cornell.edu

## Abstract

Many reinforcement learning (RL) algorithms are too costly to use in practice due to the large sizes $S, A$ of the problem's state and action space. To resolve this issue, we study transfer RL with latent low rank structure. We consider the problem of transferring a latent low rank representation when the source and target MDPs have transition kernels with Tucker rank $(S, d, A), (S, S, d), (d, S, A)$, or $(d, d, d)$. In each setting, we introduce the transfer-ability coefficient $\alpha$ that measures the difficulty of representational transfer. Our algorithm learns latent representations in each source MDP and then exploits the linear structure to remove the dependence on $S, A$, or $SA$ in the target MDP regret bound. We complement our positive results with information theoretic lower bounds that show our algorithms (excluding the $(d, d, d)$ setting) are minimax-optimal with respect to $\alpha$.

## 1 Introduction

Recently, reinforcement learning (RL) algorithms have exhibited great success on a variety of problems, e.g., video games [31], robotics [21], etc, that require a series of decisions over time. These algorithms can be used on any problem that can be modeled as a Markov decision process (MDP) with state space cardinality $S$, action space cardinality $A$, and horizon $H$. However, RL algorithms' practical application is constrained due to the large amounts of data and computational resources required to train models that outperform heuristic approaches. This limitation arises due to the inefficient scaling of RL algorithms with respect to the size of the state and action spaces. In an online setting where rewards are incurred during the process of learning, we may want to minimize the regret, or the difference between the reward of an optimal policy and the reward collected by the learner. In the finite-horizon online episodic MDP setting with $K$ episodes, any algorithm must incur regret of at least $\tilde{\Omega}(\sqrt{SAH^3K})$ [18]. This regret bound is often too large in practice as many problems modeled as MDPs have large state and action spaces. For example, in the Atari games [25], the state space is the set of all images as a vector of pixel values.

One way to remove the dependence on the state or action space in RL algorithms is to leverage existing data or models from similar problems through transfer learning. In the transfer learning setting for MDPs, the learner can interact with or query from multiple source MDPs, which the learner can utilize at low-cost to improve their performance in the target MDP [27]. The efficacy of the transfer learning approach depends both on the similarity between the source and the target MDPs, and how information is utilized when transferring knowledge between the source and target.

As feature learning in MDPs is difficult and expensive, reusing learned low rank representations can greatly improve the performance on new problems. Thus, in this work, we consider the setting when the source and target MDPs exhibit latent low rank structure, and the transfer RL task is to learn latent low rank representation of the state, action, or state-action pair from the source MDPs, and subsequently use it to improve learning on the target MDP. If the learned representations were perfect, then one could remove all dependence on the state space or action space in the learning task on the

38th Conference on Neural Information Processing Systems (NeurIPS 2024).

|  | Tucker Rank | Source Sample Complexity | Target Regret Bound |
|---|---|---|---|
| Theorem 2 | $(S,S,d)$ | $\tilde{O}\left(d^4(1+A/S)M^2H^4\alpha^2 T\right)$ | $\tilde{O}\left(\sqrt{(dMH)^3 ST}\right)$ |
| Theorem 10 | $(S,d,A)$ | $\tilde{O}\left(d^4(S/A+1)M^2H^4\alpha^2 T\right)$ | $\tilde{O}\left(\sqrt{(dMH)^3 AT}\right)$ |
| Theorem 14 | $(d,S,A)$ | $\tilde{O}\left(\alpha^2 dM^2 TSA\right)$ | $\tilde{O}\left(\sqrt{(dMH)^3 T}\right)$ |
| Theorem 15 | $(d,d,d)$ | $\tilde{O}\left(d^{10}(S+A)M^2H^4\alpha^4 T\right)$ | $\tilde{O}\left(\sqrt{d^6 M^6 H^3 T}\right)$ |
| Theorem 3.1 [4] | $(d,S,A)$ | $\tilde{O}(\alpha^5 SA^5 H^7 d^5 M^2 T)$ | $\tilde{O}(\sqrt{d^3 H^4 T})$ |

Table 1: Theoretical guarantees of our algorithms alongside results from literature in the different Tucker rank settings. See Table 3 for the regret bounds of algorithms in each Tucker rank setting with known latent representations.

target MDP. This approach relies on a critical assumption that the transition kernels of the source and target MDPs have low Tucker rank when viewed as an $S$-by-$S$-by-$A$ tensor.

The work [4] studied this problem when the transition kernel is low rank along the first mode, i.e., with Tucker rank $(d,S,A)$ where $d < \min\{S,A\}$; this is also referred to as the Low Rank MDP model. We also consider this setting, as well as the complementary settings in which the transition kernel has low Tucker rank along the second or third mode, i.e., Tucker rank $(S,d,A)$ or $(S,S,d)$. In addition, we study the $(d,d,d)$ Tucker rank setting, where we can further exploit the low rank structure to improve performance in the target problem. (As we elaborate later, up to a factor of $d$, the $(d,d,d)$ setting subsumes the remaining settings with Tucker ranks $(d,d,A),(S,d,d)$ and $(d,S,d)$.)

The different modes are not interchangeable algorithmically as there is a directionality due to the time of the transition between the current and future state. Additionally, the algorithm in [4] is not computationally tractable as it utilizes an optimization oracle which may not be practical. They assume that the latent representations are within a known finite function class. Our work proposes a computationally efficient algorithm that can handle latent low rank structure on the transition kernel along any of the three different modes.

As the success of a transfer learning approach depends on the similarity between the source and target problems, we introduce the *transfer-ability coefficient* $\alpha$, which quantifies the similarity between the source and target problems under our low rank assumptions. This coefficient, genearlizing the task-relatedness in [4], captures the unique challenge in the transfer reinforcement learning setting.

**Our Contributions:** We propose new computationally efficient transfer RL algorithms that admit the desired efficient regret bounds under all low Tucker rank settings $(S,d,A),(S,S,d),(d,S,A)$, and $(d,d,d)$. With enough samples from the source MDPs, we remove the dependence on $S,A$, or $SA$ in the regret bound on the target MDP. In addition, we introduce the transfer-ability coefficient $\alpha$ that measures the ease of transferring a latent representation from the source MDPs to target MDP. We establish information theoretic lower bounds showing that our algorithms are optimal with respect to $\alpha$ (excluding the $(d,d,d)$ Tucker rank setting). In particular, in the $(d,S,A)$ Tucker rank setting, we achieve the optimal dependence of $\alpha^2$ while [4] is sub-optimal with a dependence on $\alpha^5$.

Table 1 summarizes our main theoretical results, suppressing the dependence on common matrix estimation terms, and compares them with the results from [4].[1] In the appendix, we further consider the setting where one has access to generative models in both the source and target problems, and show that the benefits of transfer learning go beyond alleviating exploration burden.

## 2 Related Work

**RL in the** $(d,S,A)$ **Tucker rank Setting:** There exists an extensive literature on Linear MDPs and Low-rank MDPs, which correspond to the $(d,S,A)$ Tucker rank setting. Linear MDPs assume

---

[1]Since $\bar{\alpha} \geq \alpha$, moving $\bar{\alpha}$ into the source sample complexity yields a sample complexity that scales with $\alpha^5$. $\Phi$ and $\Gamma$ refer to the function classes that contain the true $\phi$ and $\mu$, respectively. As we make no assumptions on the function class of $\phi$, we replace $\log(|\Phi||\Gamma|)$ with $|S||A|$.

that the dynamics are linear with respect to a known feature mapping $\phi$. This allows one to remove all dependence on the size of the state and action spaces and replace it with the rank $d$ of the latent space [17, 37, 36, 34]. The work [13] proposes an algorithm that admits a regret bound of $\tilde{O}(\sqrt{d^2 H^2 T})$, matching the known lower bounds. Our algorithm in the target phase modifies existing linear MDP algorithms to take advantage of the knowledge learned in the source phase. Low Rank MDPs impose the same low-rank structure as linear MDPs but assume that $\phi$ is unknown. To avoid dependence on the size of the state space, many works in this setting construct algorithms that utilize a computationally inefficient oracle to admit sample complexity or regret bounds that scale with the size of the action space and the log of the size of the function class the true low rank representation [3, 33, 26, 40]. [32] incurs sample complexity on $SA$ instead of the size of the function class to learn near-optimal policies in reward free low rank MDPs.

**Transfer RL:** As feature learning in low rank MDPs is difficult, [4, 10, 23, 15] study representational transfer in low rank MDPs. The work most closely related to ours is [4], which performs reward-free exploration in the source problem to learn a sufficient representation to use in the target problem in the $(d, S, A)$ Tucker rank setting. They propose a task relatedness coefficient that captures the existing representational transfer RL settings [10]. This coefficient is similar to our transfer-ability coefficient but differs as our low rank assumption is imposed on different modes of the probability transition tensor. Their algorithm admits a source sample complexity bound of $\tilde{O}(A^4 \alpha^3 d^5 H^7 K^2 T \log(|\Phi||\Gamma|))$ with a target regret bound of $\tilde{O}(\sqrt{d^3 \alpha^2 H^4 T})$. Other transfer RL works [10, 23] require reach-ability assumptions to learn the latent representations in the source phase. Similar to transfer RL, reward free RL studies the problem of giving the learner access to only the transition kernel in the first phase, and in the second phase the learner must output a good policy when given multiple reward functions [19, 39, 11, 32]. While reward free RL is similar to our setting, enabling the dynamics to differ between the source and target phase greatly increases the difficulty and changes the objective in the source problem, i.e., learning a good latent representation instead of collecting trajectories that sufficiently explore the state space.

**RL in the** $(S, d, A)$ **or** $(S, S, d)$ **Tucker Rank Setting:** The work [30] introduces the RL setting in which the transition kernel has Tucker rank $(S, d, A)$ or $(S, S, d)$. Assuming access to a generative model, their algorithms learn near-optimal polices with sample complexities that scale with $S + A$, thereby circumventing the $SA$ lower bound for tabular reinforcement learning [6]. The work [35] consider the same low Tucker rank assumption on the dynamics but in the offline RL setting. The works [38, 29, 28] empirically show the benefits of combining matrix/tensor estimation with RL algorithms on common stochastic control tasks. [16] consider the low-rank bandits setting and use a similar algorithm to ours by estimating the singular subspaces of the reward matrix to improve the regret bound's dependence on the dimension of the problem.

## 3 Preliminaries

We consider the transfer reinforcement learning setting introduced by [4], in which the learner interacts with $M$ finite-horizon source MDPs and one finite horizon episodic target MDP. All MDPs share the same discrete state space $\mathcal{S}$ with cardinality $S$, discrete action space $\mathcal{A}$ with cardinality $A$, and finite horizon $H \in \mathbb{Z}_+$.[2] Let $r = \{r_h\}_{h \in [H]}$ be the deterministic unknown reward function of the target MDP, where $r_h : \mathcal{S} \times \mathcal{A} \to [0, 1]$. For $m \in [M]$, let $r_m = \{r_{m,h}\}_{h \in [H]}$ be the deterministic unknown reward function for the $m$th source MDP, where $r_{m,h} : \mathcal{S} \times \mathcal{A} \to [0, 1]$.[3] Let $\mathcal{P} = \{P_h\}$ be the transition kernel for the target MDP, where $P_h(s'|s, a)$ is the probability of transitioning to state $s'$ from state-action pair $(s, a)$ at time step $h$. Similarly, for $m \in [M]$, $\mathcal{P}_m$ is the transition kernel for $m$th source MDP and defined in the same way as $\mathcal{P}$. An agent's policy has the form $\pi = \{\pi_h\}_{h \in [H]}, \pi_h : \mathcal{S} \to \Delta(\mathcal{A})$, where $\pi_h(a|s)$ is the probability that the agent chooses action $a$ at state $s$ and time step $h$.

The value function and action-value $(Q)$ function of a policy $\pi$ are the expected reward of following $\pi$ given a starting state and starting action, respectively, beginning at time step $h$. We define these as

$$V_h^\pi(s) := \mathbb{E}\left[ \sum_{t=h}^H r_t(s_t, \pi_t(s_t)) | s_h = s \right], \quad Q_h^\pi(s, a) := \mathbb{E}\left[ \sum_{t=h}^H r_t(s_t, a_t) | s_h = s, a_h = a \right]$$

---

[2]It is possible to relax the assumption of sharing the same state/action spaces. In particular, depending on the Tucker rank of the transition kernel, we only require the source and target MDPs share only one of the spaces.

[3]Our results can easily be extended to stochastic reward functions.

where $a_t \sim \pi_t(s_t), s_{t+1} \sim P_t(\cdot|s_t, a_t)$. The optimal value function and action value function are $V_h^*(s) := \sup_\pi V_h^\pi(s)$ and $Q_h^*(s, a) := \sup_\pi Q_h^\pi(s, a)$ for all $(s, a, h) \in \mathcal{S} \times \mathcal{A} \times [H]$. They satisfy the Bellman equations $V_h^*(s) = \max_{a \in \mathcal{A}} Q_h^*(s, a), Q_h^*(s, a) = r_h(s, a) + E_{s' \sim P_h(\cdot|s,a)}[V_{h+1}^*(s')]$ for all $(s, a, h) \in \mathcal{S} \times \mathcal{A} \times [H]$. We define an $\epsilon$-optimal policy $\pi$ for $\epsilon > 0$ as any policy that satisfies $V_h^*(s) - V_h^\pi(s) \leq \epsilon$ for all $s \in \mathcal{S}, h \in [H]$. Similarly, $Q$ is $\epsilon$-optimal if $Q_h^*(s, a) - Q_h(s, a) < \epsilon$ for all $(s, a, h) \in \mathcal{S} \times \mathcal{A} \times [H]$. Also, we use the shorthand $P_h V(s, a) = \mathbb{E}_{s' \sim P_h(\cdot|s,a)}[V(s')]$.

In the transfer RL problem, the learner first interacts with the $M$ source MDPs without access to the target MDP. In this phase, called the source phase, the learner is given access to a generative model in each of the source MDPs. The generative model takes in a state-action pair $(s, a)$ and time step $h$ and outputs $r_h(s, a)$ and an independent sample from $P_h(\cdot|s, a)$ [20]. Access to a generative model in the source phase is necessary; in particular, it has been shown in [4, Theorem 4.1] that without access to a generative model in the source phase, one cannot learn a near-optimal policy in the target phase using the learned representation. Then, in the next phase, called the target phase, the learner loses access to the source MDPs and interacts only with the target MDP in a standard online model, without access to a generative model. The performance of the learner is evaluated by two metrics: i) the regret in the target MDP, i.e., $Regret(T) = \sum_{k \in [K]} V_1^*(s) - V_1^{\pi^k}(s)$, where $\pi^k$ is the learner's policy in episode $k$, and ii) the number of samples taken in the source phase.

## 3.1 The Tucker Rank of a Tensor

For the reader's convenience, we first recall the standard definition of the Tucker rank of a tensor.

**Definition 1** (Tucker Rank [24]). *A tensor $M \in \mathbb{R}^{n_1 \times n_2 \times n_3}$ has Tucker rank $(d_1, d_2, d_3)$ if $(d_1, d_2, d_3)$ is the smallest such that there exists a core tensor $X \in \mathbb{R}^{d_1 \times d_2 \times d_3}$ and matrices $G_i \in \mathbb{R}^{n_i \times d_i}$ for $i \in [3]$ with orthonormal columns such that*

$$M(a, b, c) = \sum_{i \in [d_1], j \in [d_2], k \in [d_3]} X(i, j, k) G_1(a, i) G_2(b, j) G_3(c, k).$$

We note in passing that all our sample complexity and regret bounds in Table 1 remain valid when $d$ is an upper bound on the exact rank. In standard MDPs, the transition kernel can have Tucker rank $(S, S, A)$ as there is no low rank structure imposed on the tensor. In this work, we investigate the benefits of assuming that the transition kernel is low rank along each mode/dimension of the transition kernel, which corresponds to the $(S, d, A), (S, S, d)$, and $(d, S, A)$ Tucker rank settings. To illustrate the difference in these three settings, we consider $d = 1$ and present the corresponding structure of the transition kernel in Equations (1), (2), and (3), respectively:

$$\text{Tucker rank } (S, 1, A): \qquad P(s'|s, a) = \mu(s)\phi(s', a), \tag{1}$$

$$\text{Tucker rank } (S, S, 1): \qquad P(s'|s, a) = \mu(a)\phi(s', s), \tag{2}$$

$$\text{Tucker rank } (1, S, A): \qquad P(s'|s, a) = \mu(s')\phi(s, a), \tag{3}$$

for some functions $\mu, \phi$ defined on the appropriate domains. Figure 1 pictorially displays the $(S, S, d)$ Tucker rank assumption on the transition kernel. While the three settings may seem similar, the $(S, d, A)$ and $(S, S, d)$ Tucker rank settings differ significantly from the $(d, S, A)$ Tucker rank setting (i.e., Low Rank MDP). In particular, the $(d, S, A)$ setting assumes low rank structure across time, i.e., the current state and action vs. the next state, while the other Tucker rank assumptions impose low rank structure between the current state and action. These differences are significant as each setting requires different algorithms in the target problem and construction for the lower bounds.

Additionally, we analyze the $(d, d, d)$ Tucker rank setting, where the transition kernel fully factorizes, in which case the next state, current state, and action all have separate low rank representations. This assumption allows us to further exploit the low rank structure to improve the regret bound in the target phase when compared to the $(S, d, A)$ and $(S, S, d)$ Tucker rank settings. Note that this $(d, d, d)$ Tucker rank assumption is equivalent to imposing low rank along any two modes of the transition kernel, up to factors of $d$ in the sample complexity and regret bounds. For example, for an $(S, d, d)$ Tucker rank transition kernel $P$, one can factor the $S \times d \times d$ core tensor into a $S \times d$ orthonormal matrix and $d^2 \times d \times d$ core tensor, which implies $P$ has Tucker rank $(d^2, d, d)$.

While the sample complexity and regret bound of each algorithm differ in each Tucker rank setting, one should choose which algorithm to use based on the low rank structure inherent in their problem. For example, in the $(S, S, d)$ Tucker rank setting, one assumes that each action has a low rank

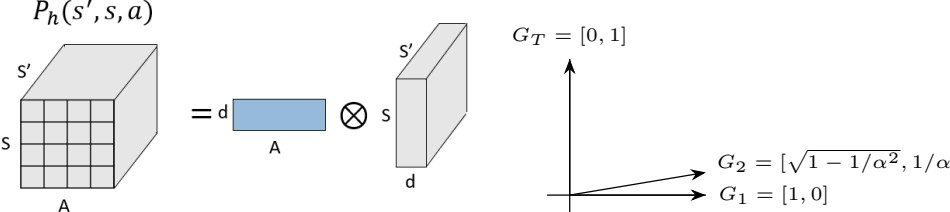

Figure 1: Transition kernel with Tucker rank $(S, S, d)$

Figure 2: A hard transfer learning example where one estimates $G_T$ with $G_1$ and $G_2$

representation. One potential application of this setting is a personal movie recommender system. Specifically, the states are user's different states of mind while each action chooses a movie from a large collection. As movies in the same genre evoke similar responses, each action can be mapped to a lower dimensional space with the axis being each genre. In contrast, each state of mind is distinct. In the $(S, d, A)$ Tucker rank setting, one assumes that the state one transitions from has a low rank representation. The $(d, d, d)$ Tucker rank setting combines the two previous assumptions and asserts that current states and actions have separate low dimensional representations while low rank MDPs with $(d, S, A)$ Tucker rank assume that each state-action pair has a joint low rank representation. For sake of brevity, we will only present our assumption and algorithm for the $(S, S, d)$ Tucker rank setting in the main body of this work and defer the other settings to the appendix.

### 3.2 Low Tucker Rank MDPs

We first define the incoherence $\mu$ of a matrix, which is used in the regularity conditions of our Tucker rank assumption.

**Definition 2** (Incoherence). *Let $X \in \mathbb{R}^{m \times n}$ have rank $d$ with singular value decomposition $X = U\Sigma V^\top$ with $U \in \mathbb{R}^{m \times d}, V \in \mathbb{R}^{n \times d}$. Then, $X$ is $\mu$-incoherent if for all $i \in [m], j \in [n]$, $\|U(i, :)\|_2 \leq \sqrt{d\mu/m}$ and $\|V(j, :)\|_2 \leq \sqrt{d\mu/n}$.*

A small $\mu$ guarantees that the signal in $U$ and $V$ is sufficiently spread out among the rows in those matrices. The notion of incoherence is pervasive in the low-rank estimation literature [7, 8].

Introduced in [30], we assume that reward functions are low rank and the transition kernels have low Tucker rank in each of the source and target MDPs. The main benefit of Assumption 1 is that for any value function $V$, the Bellman update of any value function, $Q_h = r_h + P_h V$, is a low rank matrix due to Proposition 4 [30].

**Assumption 1.** *In each of the $M$ source MDPs, the reward functions have rank $d$, and the transition kernels have Tucker rank $(S, S, d)$. The target MDP has reward function with rank $d'$ and transition kernels with Tucker rank $(S, S, d')$ where $d' \leq dM$. Thus, there exists $S \times S \times d$ tensors $U_{m,h}$, $S \times S \times d'$ tensors $U_h$ s, $A \times d$ $\mu$-incoherent matrices $G_{m,h}$ with orthonormal columns, $A \times d'$ $\mu$-incoherent matrices $G_h$ with orthonormal columns, and $S \times d$ matrices $W_{m,h}$, $S \times d'$ matrices $W_h$ such that*

$$P_{m,h}(s'|s,a) = \sum_{i \in [d]} G_{m,h}(a,i) U_{m,h}(s',s,i), \quad r_{m,h}(s,a) = \sum_{i \in [d]} G_{m,h}(a,i) W_{m,h}(s,i),$$
$$P_h(s'|s,a) = \sum_{i \in [d']} G_h(a,i) U_h(s',s,i), \quad r_h(s,a) = \sum_{i \in [d']} G_h(a,i) W_h(s,i)$$

*where* $\|\sum_{s' \in S} g(s') U_{m,h}(s',s,:)\|_2, \|\sum_{s' \in S} g(s') U_h(s',s,:)\|_2, \|W_h(s)\|_2, \|W_{m,h}(s)\|_2 \leq \sqrt{A/(d\mu)}$ *for all $s \in S$, $a \in A$, $h \in [H]$, and $m \in [M]$ for any function $g : S \to [0, 1]$.*

The latent representations of the reward function and transition kernel are not unique under our assumption. Figure 1 visualizes the low rank structure Assumption 1 imposes on the transition kernel. While Assumption 1 states that the source MDPs and target MDPs have latent low rank structure, the assumption does not guarantee that low rank representations are similar. Additionally, we allow target MDP to be more complex than a single source MDP (at most Tucker rank $(S, S, dM)$ compared to Tucker rank $(S, S, d)$, respectively). Thus, to allow for transfer learning, we assume that for each step $h \in [H]$, the space spanned by the target latent factors is a subset of the space spanned by the latent factors from all the source MDPs.

**Assumption 2.** *Suppose Assumption 1 holds. The target MDP latent factors $G_h$ and source MDP latent factors $G_{m,h}$ satisfy for all $h \in [H]$, $\mathrm{Span}(\{G_h(:,i)\}_{i \in [d']}) \subseteq \mathrm{Span}(\{G_{m,h}(:,i)\}_{i \in [d], m \in [M]})$.*

Under Assumption 2, it is possible that no single source MDP captures the target MDP, but the union of all source MDP does. It is also possible that the source MDPs span a strictly larger subspace than the target MDP.

Thanks to the above assumptions, by estimating the latent features $\{G_{m,h}\}$ of the source MDPs, we can construct an approximation of the target features $\{G_h\}$ **without interacting** with the target MDP, which is essential to transfer learning. Assumption 1 only upper bounds the rank of $Q^*_{m,h}$ by $d$. However, to learn each source latent factor, one needs to obtain a good estimate of a rank $d$ matrix with at least one large entry. Therefore, we assume that the optimal $Q$-functions in each source MDP are full rank with the largest entry lower bounded by a constant.

**Assumption 3.** *For each $m \in [M], h \in [H]$, the optimal Q-function of the $m$-th source MDP at step $h$ satisfies $rank(Q^*_{m,h}) = d$ and $\|Q^*_{m,h}\|_\infty \geq C$ for some $C \in \mathbb{R}^+$.*

This assumption allows one to learn the near-optimal $Q$-functions from noisy samples of the source MDPs. Assumption 3 and its variants are common in the literature of noisy low-rank estimation [2, 9]. Without the above assumption (the rank of $Q^*$ is strictly less than $d$), then one cannot learn every latent factor $G_{m,h}(:,i)$ since it may have a zero singular value, and thus using the approximate feature mapping would result in regret linear in $T$. Similarly, $\|Q^*\|_\infty \geq C$ ensures that some entries of $Q^*$ are sufficiently large and prevents the noise from dominating the low rank signal that one learns in the source phase.

## 4   Transfer-ability

Our transfer-ability coefficient is motivated by the following observation. Under Assumption 2, we can construct each target latent factor $G_h$ with a linear combination of the source latent factors $G_{m,h}$. The magnitudes of the coefficients correspond to the difficulty in estimating the target latent factor because the estimation error of $G_{m,h}$ is amplified by its coefficient. Thus, when these coefficients are too large, transfer learning is essentially impossible; one example is when the source latent factors are almost identical or parallel to each other while the target latent factor is orthogonal to the source latent factors (see Figure 2).

However, these coefficients are not unique and are a function of the source and target latent factors. Furthermore, as the source MDPs latent factors of $Q^*_{m,h}$ and target latent factors $\{G_h\}$ are not unique, each choice of latent factor results in a different set of coefficients. Thus, we introduce $\mathcal{B}_h$ as the set containing all such coefficients given the subspaces from the target and source MDPs. Let $\mathcal{G}_{S,h}$ be the set of all $\mathbb{R}^{S \times d}$ orthonormal latent factor matrices with columns that span the column space of $Q^*_{m,h}$ for all $m \in [M]$, and let $\mathcal{G}_{T,h}$ be the set of all $\mathbb{R}^{S \times d'}$ orthonormal latent factor matrices with columns that span the column space of $P_h(s')$ for all $s' \in \mathcal{S}$. Then, we define the set $\mathcal{B}_h(\mathcal{G}_{T,h}, \mathcal{G}_{S,h})$ to contain all such coefficients, i.e., $B_h \in \mathbb{R}^{d',d,M}$ and $B_h \in \mathcal{B}_h(\mathcal{G}_{T,h}, \mathcal{G}_{S,h})$ if there exists $G_h \in \mathcal{G}_{T,h}, G_{m,h} \in \mathcal{G}_{S,h}$ such that for all $i \in [d'], h \in [H]$, $G_h(\cdot, i) = \sum_{j \in [d], m \in [M]} B_h(i,j,m) G_{m,h}(\cdot, j)$. Now, we define $\alpha$, which precisely measures the challenge involved in transferring the latent representation.

**Definition 3** (Transfer-ability Coefficient). *Given a transfer RL problem that satisfies Assumptions 1, 2, and 3, with the definition of $\mathcal{B}_h$ above, we define $\alpha$ as*

$$\alpha := \max_{h \in [H]} \min_{B \in \mathcal{B}_h(\mathcal{G}_{T,h}, \mathcal{G}_{S,h})} \max_{i \in [d'], j \in [d], m \in [M]} |B(i,j,m)|.$$

Note that the minimum is taken over all latent factor matrices. When there is only one source MDP, $\alpha$ is small, i.e., less than or equal to one, because the columns of $G$ form an orthonormal basis of the space containing the target latent factor. In the best case, $\alpha$ is $1/(dM)$ (see Appendix D). On the other hand, when there are multiple source MDPs, $\alpha$ can be arbitrarily large as in Figure 2; in this case, the estimation error of $G_1$ and $G_2$ are amplified by $\alpha$ when estimating $G_T$ using $G_1$ and $G_2$. We remark that adding more source MDPs or increasing the rank of $Q^*_{m,h}$ will not increase $\alpha$ as one can use the same set of coefficients without the additional source MDP or larger subspace of $Q^*_{m,h}$. See Appendix D for more discussion on $\alpha$.

Our transfer-ability coefficient is similar to the quantity $\alpha_{\max}$ defined in [4]. While these terms both quantify the similarity between the source and target MDPs, in our setting, we allow for the source MDPs' state latent factors, $\{G_{m,h}\}$, to differ from target MDP's state latent factors while [4] assumes that all MDPs share the same latent representation $\phi$. Our setting is more general than the one in [4] even when assuming the same Tucker rank assumption.

## 4.1  Information Theoretic Lower Bound

To formalize the importance of $\alpha$, we prove a lower bound that shows that a dependence on $\alpha$ in the sample complexity of the source phase is necessary to benefit from transfer learning. Before proving our main result, we first present an intermediate lower bound to provide intuition on how $\alpha$ measures the difficulty in transferring a learned representation.

**Theorem 1.** *There exist two transfer RL instances such that (i) they satisfy Assumptions 1 and 2, (ii) they cannot be distinguished without observing $\Omega(\alpha^2)$ samples in the source phase, and (3) they have target action latent features that are orthogonal to each other.*

Learning a latent representation $G$ that is orthogonal to the true representation is no better than randomly guessing a representation $G'$ in the source phase; using either $G$ or $G'$ in the target phase incurs regret linear in $T$. Thus, Theorem 1 states that without observing $\Omega(\alpha^2)$ samples in the source MDPs, one cannot learn a good policy in the target phase with $G$ or $G'$, and one should disregard the information from the source MDPs and use a non-transfer learning RL algorithm to achieve regret bound that scales with $\sqrt{T}$. Therefore, $\Omega(\alpha^2)$ samples are required to benefit from transfer learning.

To prove Theorem 1, we construct two $(S, S, 1)$ transfer RL problems that satisfies Assumption 2 with similar source action latent representations $G_{m,h}^i \in \mathcal{G}$ but orthogonal target action latent representations $G_T^i$ for $i \in \{1, 2\}$. To avoid learning the incorrect representation, one must identify the transfer RL problem by differentiating the $Q^*$s. By construction, the $Q^*$s for the two transfer RL problems differ by at most $1/\alpha$ entrywise, so one must observe $\Omega(\alpha^2)$ samples in the source problem from standard hypothesis testing lower bounds [5]. Observing less samples in the source problem results in an estimate orthogonal to the true target representation with constant probability. See Appendix E for the formal proof. One can easily prove a similar result in the $(S, d, A)$ and $(d, S, A)$ Tucker rank setting with similar constructions by switching the states and actions (see Appendices F and G).

## 5  Algorithm

We now present our algorithm that achieves the optimal dependence on $\alpha$. In the source phase, our algorithm can use any algorithm that learns $\epsilon$-optimal $Q$ functions on each source MDP. As the error bound on the estimated feature representation depends on the incoherence of $Q_{m,h}^*$, we use LR-EVI, which combines empirical value iteration with low rank matrix estimation [30]. After obtaining estimates $\bar{Q}_{m,h}$ of $Q_{m,h}^*$, our algorithm computes the singular value decompositions (SVDs) of each $\bar{Q}_{m,h}$ to construct the latent feature representation $\tilde{G}$ by scaling and concatenating the singular vectors. Then, in the target phase, our algorithm deploys a modification of LSVI-UCB [17] adapted to our Tucker rank setting called LSVI-UCB-(S, S, d) using the $\tilde{G}$. For ease of notation, we define $T_{k,h}^s = \{k' \in [k] | s_h^k = s\}$, which is the set of episodes before $k$, in which one was at $s$ at time step $h$.

---

**Algorithm 1** Source Phase

---

**Input:** $\{N_h\}_{h \in [H]}$
1: **for** $m \in [M]$ **do**
2:     Run LR-EVI($\{N_h\}_{h \in [H]}$) on source MDP $m$ to obtain $\bar{Q}_{m,h}$.
3:     Compute the singular value decomposition of $\bar{Q}_{m,h} = \hat{F}_{m,h} \hat{\Sigma}_{m,h} \hat{G}_{m,h}^\top$ .
4: Compute feature mapping $\hat{\tilde{G}} = \{\hat{\tilde{G}}_h\}_{h \in [H]}$ with $\hat{\tilde{G}}_h = \sqrt{\frac{A}{d\mu}} \begin{bmatrix} \hat{G}_{1,h} & \dots & \hat{G}_{M,h} \end{bmatrix}$ .

---

LSVI-UCB-(S, S, d) uses the same mechanisms as LSVI-UCB, but we compute coefficients and Gram matrices for each action. Specifically, we update the weights $w_h^s$ with the solution to the

**Algorithm 2** Target Phase: LSVI-UCB-(S, S, d)

---

**Input:** $\lambda, \beta_{k,h}^s, \hat{\tilde{G}}$
1: **for** $k \in [K]$ **do**
2:    Receive initial state $s_1^k$.
3:    **for** $h = H, \ldots, 1$ **do**
4:       **for** $s \in S$ **do**
5:          /* Compute Gram matrix $\Lambda_h^s$ for each state $s$ */
6:          Set $\Lambda_h^s \leftarrow \sum_{t \in T_{k-1,h}^s} \hat{\tilde{G}}_h(a_h^t) \hat{\tilde{G}}_h(a_h^t)^\top + \lambda \mathbf{I}$.
7:          /* Estimate $w_h^s$ via regularized least squares */
8:          Set $w_h^s \leftarrow (\Lambda_h^s)^{-1} \sum_{t \in T_{k-1,h}^s} \hat{\tilde{G}}_h(a_h^t) \left[ r_h(s, a_h^t) + \max_{a' \in A} Q_{h+1}(s_{h+1}^t, a') \right]$.
9:       /* Estimate $Q^*$ via the low rank structure with optimism */
10:       Set $Q_h(\cdot, \cdot) \leftarrow \min \left( H, \langle w_h, \hat{\tilde{G}}_h(\cdot) \rangle + \beta_{k,h} \sqrt{\hat{\tilde{G}}_h(\cdot)^\top (\Lambda_h)^{-1} \hat{\tilde{G}}_h(\cdot)} \right)$.
11:    **for** $h \in [H]$ **do**
12:       Take action $a_h^k \leftarrow \max_{a \in A} Q_h(s_h^k, a)$, and observe $s_{h+1}^k$.

---

regularized least squares problem,

$$w_h^s \leftarrow \arg \min_{w \in \mathbb{R}^d} \sum_{t \in T_{k,h}^s} (r_h(s, a_h^t) + \max_{a' \in \mathcal{A}} Q_{h+1}(s_{h+1}^t, a') - w^\top \tilde{G}_h(a_h^t))^2 + \lambda \|w\|_2^2,$$

using the reward when $s$ was observed and add an exploration bonus $\beta_{k,h}^s \sqrt{\hat{\tilde{G}}_h(a)^\top (\Lambda_h^s)^{-1} \hat{\tilde{G}}_h(a)}$ to our estimate of $Q^*$, which is common in linear MDP/bandits algorithms [17, 1]. Multiplying the latent factors by $\sqrt{A/(d\mu)}$ to construct $\hat{\tilde{G}}$ ensures that the feature mapping and coefficients are of similar magnitude, which is crucial in adapting the mechanisms of LSVI-UCB to our setting.

While one may wonder why our modification of LSVI-UCB is necessary, Algorithm 2 improves the regret bound by a factor $\sqrt{S}$ compared to using any linear MDP algorithm off the shelf; reducing our setting to a linear MDP results in using a feature mapping with dimension $dS$. Thus, using any linear MDP algorithm results in a target regret bound of $\tilde{O}(\sqrt{d^2 S^2 H^2 T})$ [41]. However, our algorithm shaves off a factor of $\sqrt{S}$ as our algorithm computes $S$ copies of $d \times d$ dimensional gram matrices in contrast to the $d^2 S^2$ dimensional gram matrices used in linear MDP algorithms.

## 6 Theoretical Results

In this section, we prove our main theoretical results that show with enough samples in the source problem, our algorithms admit regret bounds that are independent of $S, A$, or both $S$ and $A$ in the target problem. We first state our main result in the $(S, S, d)$ Tucker rank setting.

**Theorem 2.** *Suppose Assumptions 1, 2, and 3 hold, and set $\delta \in (0, 1)$. Furthermore, assume that, for any $\epsilon \in (0, \sqrt{SH/T})$, $Q_{m,h} = r_{m,h} + P_{m,h} V_{m,h+1}$ has rank $d$ and is $\mu$-incoherent with condition number $\kappa$ for all $\epsilon$-optimal value functions $V_{h+1}$. Let $\lambda = 1$ and $\beta_{k,h}^s$ be a function of $d, H, |T_{k,h}^s|, |S|, M, T$. Then, for $T \geq \frac{S}{\alpha^2}$, using at most $\tilde{O}(d^4 \mu^5 \kappa^4 (1 + A/S) M^2 H^4 \alpha^2 T)$ samples in the source problems, our algorithm has regret $\tilde{O}(\sqrt{(dMH)^3 ST})$ with probability at least $1 - \delta$.*

Theorem 2 states that using transfer learning improves the performance on the target problem by removing the regret bound's dependence on the action space. Thus, with enough samples from the source MDPs one can recover a regret bound in the target problem that matches the best regret bound for any algorithm in the $(S, S, d)$ Tucker rank setting with **known** latent feature representation $G$ concerning $S$ and $T$. See Appendix E for the proof of Theorem 2.

For sake of brevity, we present our results in our other Tucker rank settings in Table 1 (see Appendices F, G, and H for the formal statements and proofs). In each of our Tucker rank settings, our algorithms use transfer learning to remove the dependence on $S, A$, or $SA$ in the target regret bound, which matches the dependence on $S$ and $A$ of the best regret bounds of algorithms with **known** latent

representation. While the source sample complexities of Theorems 2 and 10 may seem to not scale linearly in $A$ and $S$, respectively, we require the horizon in the target phase to be large enough, i.e., $T\alpha^2(S/A + 1) \geq S + A$ and $T\alpha^2(1 + A/S) \geq S + A$, respectively, so that one observes at least $\Omega(S + A)$ samples in the source phase.

While our dependence on $M$ in the target regret bound may seem sub optimal, our algorithm cannot distinguish which of the $d$ dimensions (out of $dM$ possible dimensions) match the ones in the target MDP without interacting with it. Thus, our algorithm must include all of them, which results in using at worst a $dM$-dimensional feature representation. However, in the case when the feature representations from source MDPs lie in subspaces that intersect, we can apply a singular value thresholding procedure to remove the unneeded dimensions used in the feature representation at the cost of an increased sample complexity (by a factor of $A$) in the source phase (see Appendix C).

**Comparison with [4]:** In comparison to Theorem 3.1 from [4], our result, Theorem 14, has similar dependence on $A, d, H$, and $T$ while our dependence on $\alpha$ is optimal compared to their $\alpha^5$ dependence [4]. While our dependence on $M$ in the regret bound is worse, this is due to our transfer learning setting being more general; we only require that the space spanned by the target features be a subset of the space spanned by the source features instead of having the target features being a linear combination of the source features. Furthermore, our source sample complexity scales with the size of the state space instead of scaling logarithmically in the size of the given function class.

The proofs of 2, Theorem 10, 15, and 14 synthesize the theoretical guarantees of LR-EVI, singular vector perturbation bounds, and LSVI-UCB or our modification of LSVI-UCB. Given $N$ samples from the source MDPs, we first bound the error between the singular subspaces up to a rotation by $\sqrt{d^3M(S + A)/N}$ (suppressing the dependence on common matrix estimation terms) to construct our feature mapping. Thus, with enough samples in the source MDPs, LR-EVI returns $Q$ functions with singular vectors that can be used to construct a sufficient feature representation; the additional regret incurred from using the approximate feature representation in LSVI-UCB or Algorithm 2 is dominated by the original regret of using the true representation.

### 6.1 Discussion of Optimality

In the $(S, d, A), (S, S, d), (d, S, A)$, and $(d, d, d)$ Tucker rank settings, the source sample complexity's scalings concerning $S, A, H$, and $T$ are reasonable when learning $\sqrt{A/T}, \sqrt{S/T}, \sqrt{1/T}$, and $\sqrt{1/T}$-optimal $Q$ functions, respectively, given the $Q$ learning lower bounds in [30]. Similarly, the dependence on $\mu$ and $\kappa$ are standard for RL algorithms incorporating matrix estimation.

In the $(S, d, A), (S, S, d)$, and $(d, S, A)$ Tucker rank settings, our $\alpha^2$ dependence in the source sample complexity is optimal due to our lower bounds. In the $(d, d, d)$ Tucker rank setting, our $\alpha^4$ dependence is likely optimal when eliminating the dependence on $S$ and $A$ in the target problem; one must combine both sets of singular vectors together, $F : \mathcal{S} \to \mathbb{R}^{dM}$ and $G : \mathcal{A} \to \mathbb{R}^{dM}$, to create a feature mapping $\phi : \mathcal{S} \times \mathcal{A} \to \mathbb{R}^{d^2M^2}$, which causes the $\alpha^4$ dependence in our upper bound. However, if $\alpha^4$ is too large, one can use the algorithms from the $(S, d, A)$ or $(S, S, d)$ Tucker rank settings to benefit from transfer learning. Proving an $\alpha^4$ lower bound is an interesting open question. Additionally, the dependence on $d$ and $M$ in this setting is worse as our low rank latent representation of each state-action pair lies in a $d^2M^2$ dimensional space in contrast to a space with dimension $dM$.

The regret bounds of our algorithms used in the target phase, i.e., Algorithm 2, Algorithm 4, and LSVI-UCB [17], in each Tucker rank setting are optimal with respect to $S, A$, and $T$. Since each Tucker rank setting captures tabular MDPs, we can construct lower bounds with a reduction from the bound in [18], e.g., in the $(S, S, d)$ setting, the regret is at least $\tilde{\Omega}(\sqrt{dSH^2T})$. However, our dependence on $d$ and $H$ are sub-optimal. The extra factor of $\sqrt{H}$ is likely due to our use of Hoeffding's inequality instead of tracking the variance of the estimates to use Bernstein's inequality. Our dependence on $d$ is sub-optimal as LSVI-UCB is sub-optimal with respect to $d$; for linear MDPs, LSVI-UCB++ [13] admits a regret bound of $\tilde{O}(\sqrt{d^2H^2T})$, which matches the lower bound from [41]. Thus, to obtain the optimal dependence on $d$ and $H$, we would need to adapt LSVI-UCB++ to our Tucker rank setting.

---

[4]Since $\bar{\alpha} \geq \alpha$, moving $\bar{\alpha}$ into the source sample complexity yields a sample complexity that scales with $\alpha^5$.

# 7 Conclusion

We study transfer RL, in which one transfers a latent representation between source MDPs and target MDP with Tucker rank $(S, S, d), (S, d, A), (d, S, A)$ or $(d, d, d)$ transition kernels. As [4] studied transfer RL in the $(d, S, A)$ Tucker rank setting, our study completes the analysis of representational transfer in RL assuming that one or more modes of the transition kernel is low rank. Furthermore, we propose the transfer-ability coefficient that quantifies the difficulty of transferring latent representations between the source and target problems with information theoretic lower bounds. We propose computationally simple algorithms that admit regret bounds that are independent of $S, A$, or $SA$ depending on the Tucker rank setting given enough samples from the source MDPs. Furthermore, these regret bounds match the bounds of algorithms that are given the true latent representation. Our dependence on $d$ and $H$ is not optimal and can be improved by adapting the methods in [13] to our setting, which is an interesting open problem. Furthermore, proving theoretical guarantees of other forms of transfer RL is worthwhile and an interesting future direction.

**Acknowledgements:** Y. Chen is partially supported by NSF CCF-1704828 and NSF CCF-2233152.

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

# Contents

# A   Notation Table

| Symbol | Definition |
|---|---|
| $\mathcal{S}, S$ | State space, Cardinality of $\mathcal{S}$ |
| $\mathcal{A}, A$ | Action space, Cardinality of $\mathcal{A}$ |
| $H, K$ | Finite Horizon, Number of Episodes ($T = KH$) |
| $M$ | Number of Source MDPs |
| $r_{m,h}, P_{m,h}$ | Reward Function, Transition Kernel for Source MDP $m$ at step $h$ |
| $r_h, P_h$ | Reward Function, Transition Kernel for the Target MDP at step $h$ |
| $Q^\pi_{m,h}, Q^\pi_h$ | Action-value Function at Step $h$ Following $\pi$ for Source MDP $m$ and the Target MDP |
| $Q^*_{m,h}, Q^*_h$ | Optimal Action-value Function at Step $h$ for Source MDP $m$ and the Target MDP |
| $V^\pi_{m,h}, V^\pi_h$ | Value Function at Step $h$ Following $\pi$ for Source MDP $m$ and the Target MDy |
| $V^*_{m,h}, V^*_h$ | Optimal Value Function at Step $h$ for Source MDP $m$ and the Target MDP |
| $d$ | Rank of the Low-dimensional Mode of $P_{m,h}$ |
| $d'$ | Rank of the Low-dimensional Mode of $P_h$ |
| $\alpha$ | Transfer-ability Coefficient |
| $G_{m,h}(a), G_h(a)$ | True Latent Representation of Each Action in the Source and |
|  | Target MDPs in the $(S, S, d)$ Tucker Rank Setting |
| $\tilde{G}_h$ | Concatenation of $G_{m,h}$ for $m \in [M]$ |
| $\hat{\tilde{G}}_h$ | Latent Representation Estimate Computed in the Source Phase |
| LR-EVI($\cdot$) | Low Rank Empirical Value Iteration [30] |
| $\Lambda^s_h$ | Gram Matrix for State $s$ at Step $h$ |
| $w^s_h$ | Regularized Least Squares Solution for State $s$ at Step $h$ |
| $T^s_{k,h}$ | Episodes before Episode $k$, in which one was at State $s$ at Step $h$ |
| $\mu(X)$ | Incoherence of Matrix $X$ |
| $\kappa(X)$ | Condition Number of Matrix $X$ |
| $\sigma_i(X)$ | $i$-th Singular Value of $X$ |
| $\|X\|_{op}$ | Operator Norm of Matrix $X$ |
| $\|X\|_\infty$ | Maximum of the Absolute Value of Each Entry of $X$ |

Table 2: List of notation

# B   Table of Regret Bounds with Known Latent Representations

| | Tucker Rank | Regret Bound with Known Latent Representation |
|---|---|---|
| Theorem 5 | $(S, S, d)$ | $\tilde{O}(\sqrt{d^3 H^3 S T})$ |
| Theorem 8 | $(S, d, A)$ | $\tilde{O}(\sqrt{d^3 H^3 A T})$ |
| Theorem 3.1 [17] | $(d, S, A)$ | $\tilde{O}(\sqrt{d^3 H^3 T})$ |
| Theorem 5.1 [13], Theorem 5.6 [41] | $(d, S, A)$ | $\tilde{\Theta}\left(\sqrt{d^2 H^2 T}\right)$ |

Table 3: Theoretical guarantees of our algorithms alongside results from literature in the different Tucker rank settings. To the best of our knowledge, there are no algorithms tailored to the $(d, d, d)$ Tucker rank setting. Instead, one should use a linear MDP algorithm.

Since each Tucker rank setting captures tabular MDPs, we can construct lower bounds with a reduction from the bound in [18]; in the $(S, d, A)$ Tucker rank setting, the regret is at least $\tilde{\Omega}(\sqrt{dAH^2T})$, and in the $(S, S, d)$ setting, the regret is at least $\tilde{\Omega}(\sqrt{dAH^2T})$. As mentioned, our dependence on $d$ and $H$ are sub optimal because we modify LSVI-UCB [17], which has a sub-optimal dependence on $d$ and $H$. Modifying the algorithm from [13] to achieve the optimal dependence on $d$ and $H$ in the $(S, S, d)$ and $(S, d, A)$ Tucker rank settings is an interesting problem for future research.

# C   Thresholding Procedure in the $(S, S, d)$ Tucker Rank Setting

When the feature representations of each source MDP lies in orthogonal $d$-dimensional subspaces, we cannot discard any unique dimension of the $M$ subspaces as we cannot tell which of the $d$-dimensions match the ones in the target MDP without interacting with it. However, when the feature mappings from the source MDPs lie in intersecting subspaces, i.e.,

$$\text{rank}\left(\tilde{G}_h\right) = \text{rank}\left(\sqrt{\frac{A}{d\mu}}\begin{bmatrix} G_{1,h} & \dots & G_{M,h} \end{bmatrix}\right) = d''$$

for $d' \leq d'' \leq dM$ (recall the target MDP has Tucker rank $(S, S, d')$), we use the following procedure: after computing $\tilde{G}_h$ via concatenation of $\hat{G}_{m,h}$ for all $m \in [M]$, we perform a singular value decomposition of $\tilde{G}_h$ and threshold the singular values to keep only $d''$ dimensions. With this procedure, our regret bound now depends on $d''$ instead of $dM$ at the cost of an increased source sample complexity. We only present the theorem and proof in the $(S, S, d)$ Tucker rank setting, but our results and methods extend to the other Tucker rank settings.

**Theorem 3.** *Suppose Assumptions 1, 2, and 3 hold, and assume the source MDP latent factors span a $d''$-dimensional space. Let $\delta \in (0, 1)$. Furthermore, assume that, for any $\epsilon \in (0, \sqrt{SH/T})$, $Q_{m,h} = r_{m,h} + P_{m,h}V_{m,h+1}$ has rank $d$ and is $\mu$-incoherent with condition number $\kappa$ for all $\epsilon$-optimal value functions $V_{h+1}$. Let $\lambda = 1$ and $\beta_{k,h}^s$ be a function of $d, H, |T_{k,h}^s|, |S|, M, T$. Then, for $T \geq \frac{S}{\alpha^2}$, using at most $\tilde{O}(d^7 \mu^5 \kappa^4 (A + A^2) M^3 H^4 \alpha^2 T/(Sd''))$ samples in the source problems, our algorithm has regret $\tilde{O}(\sqrt{(d''H)^3 ST})$ with probability at least $1 - \delta$.*

The proof of Theorem 3 follows the proof of Theorem 2, except we require a larger source sample complexity to guarantee with high probability that the $d'' + 1$-th singular value of the concatenated feature representation is sufficiently small. With this procedure, we ensure that our feature representation removes the unneeded dimensions. See Appendix E for proof of the above theorem.

## C.1   Thresholding Procedure without Knowledge of $d''$

While the above procedure requires knowledge of the dimension of the subspace spanned by the latent features of the source MDPs, one can also threshold small singular values of the concatenated feature mapping as the misspecification error will still be small. The specific procedure is as follows:

after performing the singular value decomposition on the concatenation of $\hat{G}_{m,h}$, we first find the smallest value of $t \in [dM]$ that satisfies

$$\sigma_{t+1} \leq \sqrt{\frac{tHSd\mu}{\alpha^2 T(dM-t)M^2 A}},$$

and threshold $\sigma_{t+1}, \ldots, \sigma_{dM}$. Let

$$\mathcal{T} = \left\{ t \in [dM] \,|\, \sigma_{t+1} \leq \sqrt{\frac{tHSd\mu}{\alpha^2 T(dM-t)M^2 A}} \right\}$$

be the set of $t$ that satisfies the inequality on the singular values. Then, with the specified procedure, our algorithm admits a target regret bound that depends on $t$ instead of $dM$. Note that $t$ is minimally 1 and at most $dM$.

**Theorem 4.** *Suppose Assumptions 1, 2, and 3 hold. Let $\delta \in (0,1)$ and $t = \min_{t' \in \mathcal{T}} t'$. Furthermore, assume that, for any $\epsilon \in (0, \sqrt{SH/T})$, $Q_{m,h} = r_{m,h} + P_{m,h}V_{m,h+1}$ has rank $d$ and is $\mu$-incoherent with condition number $\kappa$ for all $\epsilon$-optimal value functions $V_{h+1}$. Let $\lambda = 1$ and $\beta_{k,h}^s$ be a function of $d, H, |T_{k,h}^s|, |S|, M, T$. Then, for $T \geq \frac{S}{\alpha^2}$, using at most $\tilde{O}(d^5\mu^5\kappa^4(1 + A/S)M^3 H^4 \alpha^2 T)$ samples in the source problems, our algorithm has regret $\tilde{O}(\sqrt{(tH)^3 ST})$ with probability at least $1 - \delta$.*

With an increase in a factor of $dM$ to the source sample complexity, we can potentially improve the regret bound of the target by decreasing the dimension of the feature representation from $dM$ to $t$. See Appendix E for the proof of the above theorem.

# D  Properties of the Transfer-ability Coefficient $\alpha$

In this section, we prove properties of $\alpha$ and provide easy and hard instances. We first bound the values of $\alpha$.

**Lemma 1.** *Let $\alpha$ be defined according to Definition 4. Then, we have that $\frac{1}{dM} \leq \alpha < \infty$.*

*Proof.* Since Assumption 6 holds and that the latent factors are normalized, it follows that for all $i \in [d']$ and any latent factor matrix,

$$\|F_h(:,i)\|_2 = \|\sum_{j \in [d], m \in [M]} B_h(i,j,m)F_{m,h}(:,j)\|_2$$

$$1 \leq \sum_{j \in [d], m \in [M]} \|B_h(i,j,m)F_{m,h}(:,j)\|_2$$

$$\leq \sum_{j \in [d], m \in [M]} |B_h(i,j,m)| \|F_{m,h}(:,j)\|_2$$

$$= \sum_{j \in [d], m \in [M]} |B_h(i,j,m)|.$$

Given that $1 \leq \sum_{j \in [d], m \in [M]} |B_h(i,j,m)|$, it follows that the minimum of $\max_{j \in [d], m \in [M]} |B_h(i,j,m)|$ is attained when $|B_h(i,j,m)| = \frac{1}{dM}$. Thus, $\alpha \geq \frac{1}{dM}$

Let $\gamma \in (0,1)$. Consider the construction where the target latent factor is $F_T = [1, 0]$ and the source latent factors from source MDPs one and two are

$$F_1 = [1 - \gamma, \gamma\sqrt{2/\gamma - 1}], \quad F_2 = [1 - \gamma, -\gamma\sqrt{2/\gamma - 1}].$$

Clearly, the latent factors are orthonormal. Then, it follows that the only linear combination of $F_1$ and $F_2$ that results in $F$ is

$$F_T = cF_1 + cF_2,$$

which implies that $c = \frac{1}{2-2\gamma}$. Thus, $\alpha = \frac{1}{2-2\gamma}$, and it follows that we can make $\alpha$ arbitrarily large because for any positive constant $C$, we can choose $\gamma$ so that $\alpha > C$. $\qquad\square$

As the proof of Lemma 1 uses a construction in which $\alpha$ can be arbitrarily large, we next provide simple constructions where $\alpha$ is small. In the rank one case where all source latent factors equal the target latent factor, $\alpha$ attains its minimum of $\alpha = \frac{1}{M}$. Next, we consider the construction where

$$F_T = \left[ \frac{1}{\sqrt{d}}, \ldots, \frac{1}{\sqrt{d}} \right],$$

and there are $M$ source MDPs with the standard basis vectors in $\mathbb{R}^d$ as the latent factors equally distributed among the $M$ rank one source MDPs. Thus,

$$F_T = \sum_{m \in M} \frac{\sqrt{d}}{M} F_m,$$

and $\alpha = \sqrt{d}/M$.

We next present a transfer RL instance with large $\alpha$. Consider the following construction with the goal of trying to transfer the state latent factors. Without loss of generality, let the size of the state space and action space be $4n$ for some $n \in \mathbb{N}_+$. Furthermore, we refer to the $i$-th quarter of the states as $s_i$ for $i \in [4]$ and the $i$-th half of the actions as $a_i$ for $i \in [2]$ because our construction is a block MDP with four latent states and two latent actions. Let $H = 2$. In each MDP, the transition kernel does not vary with $h$ while the reward function is time dependent. Assume that the transition kernels have Tucker rank $(S, d, A)$ (Assumption 5). For ease of notation, we refer to the $i$-th row as any state in $s_i$ and the $i$-th column as any state in $a_i$. The target MDP's latent factors are

$$F_h = \begin{bmatrix} \sqrt{1/(2n)} & 0 \\ \sqrt{1/(2n)} & 0 \\ 0 & \sqrt{1/(2n)} \\ 0 & \sqrt{1/(2n)} \end{bmatrix}, \quad W_1 = \begin{bmatrix} \sqrt{1/(2n)} & \sqrt{1/(8n)} \\ \sqrt{1/(2n)} & \sqrt{1/(8n)} \end{bmatrix}, \quad W_2 = \begin{bmatrix} \sqrt{1/(8n)} & \sqrt{1/(2n)} \\ \sqrt{1/(8n)} & \sqrt{1/(2n)} \end{bmatrix}$$

for all $h \in [H]$, and

$$U_h(s_1|a_1) = U_h(s_2|a_1) = [3\sqrt{2}/(8\sqrt{n}), \sqrt{2}/(8\sqrt{n})]$$
$$U_h(s_1|a_2) = U_h(s_2|a_2) = [\sqrt{2}/(8\sqrt{n}), 3\sqrt{2}/(8\sqrt{n})]$$
$$U_h(s_3|a_1) = U_h(s_4|a_1) = [3\sqrt{2}/(8\sqrt{n}), \sqrt{2}/(8\sqrt{n})]$$
$$U_h(s_3|a_2) = U_h(s_4|a_2) = [\sqrt{2}/(8\sqrt{n}), 3\sqrt{2}/(8\sqrt{n})].$$

Thus, the reward functions and transition kernels for the target MDP are

$$r_1 = \begin{bmatrix} \frac{1}{2n} & \frac{1}{2n} \\ \frac{1}{2n} & \frac{1}{2n} \\ \frac{1}{4n} & \frac{1}{4n} \\ \frac{1}{4n} & \frac{1}{4n} \end{bmatrix} \quad r_2 = \begin{bmatrix} \frac{1}{4n} & \frac{1}{4n} \\ \frac{1}{4n} & \frac{1}{4n} \\ \frac{1}{2n} & \frac{1}{2n} \\ \frac{1}{2n} & \frac{1}{2n} \end{bmatrix}$$

and the transition kernels are

$$P_h(s_1|\cdot,\cdot) = P_h(s_2|\cdot,\cdot) = \begin{bmatrix} \frac{3}{8n} & \frac{1}{8n} \\ \frac{3}{8n} & \frac{1}{8n} \\ \frac{1}{8n} & \frac{3}{8n} \\ \frac{1}{8n} & \frac{3}{8n} \end{bmatrix} \quad P_h(s_3|\cdot,\cdot) = P_h(s_4|\cdot,\cdot) = \begin{bmatrix} \frac{1}{8n} & \frac{3}{8n} \\ \frac{1}{8n} & \frac{3}{8n} \\ \frac{3}{8n} & \frac{1}{8n} \\ \frac{3}{8n} & \frac{1}{8n} \end{bmatrix}$$

for all $h \in [H]$. Using the Bellman equations, it follows that the optimal $Q$ function for the above MDP is

$$Q_1^* = \begin{bmatrix} \frac{13}{16n} & \frac{15}{16n} \\ \frac{13}{16n} & \frac{15}{16n} \\ \frac{11}{16n} & \frac{9}{16n} \\ \frac{11}{16n} & \frac{9}{16n} \end{bmatrix}, \quad Q_2^* = \begin{bmatrix} \frac{1}{4n} & \frac{1}{4n} \\ \frac{1}{4n} & \frac{1}{4n} \\ \frac{1}{2n} & \frac{1}{2n} \\ \frac{1}{2n} & \frac{1}{2n} \end{bmatrix}.$$

and both $Q^*$s have $F_h$ as orthonormal latent factors. Next, we present the orthonormal latent factors for the first source MDP

$$F_{1,h} = \begin{bmatrix} 1/\sqrt{2n} & 1/\sqrt{4n} \\ -1/\sqrt{2n} & 1/\sqrt{4n} \\ 0 & 1/\sqrt{4n} \\ 0 & 1/\sqrt{4n} \end{bmatrix}, W_{1,1} = \begin{bmatrix} 3/\sqrt{4n} & 3/\sqrt{4n} \\ 1/\sqrt{4n} & 1/\sqrt{4n} \end{bmatrix}, W_{1,2} = \begin{bmatrix} 1/\sqrt{4n} & 1/\sqrt{4n} \\ 2/\sqrt{4n} & 2/\sqrt{4n} \end{bmatrix}.$$

It follows that the reward functions are

$$r_{1,1} = \begin{bmatrix} (3+3\sqrt{2})/(4n) & (1+\sqrt{2})/(4n) \\ (3-3\sqrt{2})/(4n) & (1-\sqrt{2})/(4n) \\ 3/(4n) & 1/(4n) \\ 3/(4n) & 1/(4n) \end{bmatrix}, r_{1,2} = \begin{bmatrix} (1+\sqrt{2})/(4n) & (1+\sqrt{2})/(2n) \\ (1-\sqrt{2})/(4n) & (1-\sqrt{2})/(2n) \\ 1/(4n) & 1/(2n) \\ 1/(4n) & 1/(2n) \end{bmatrix}.$$

with $P_{1,h}(\cdot|\cdot,\cdot) = 1/(4n)$. Since $P_{1,h}$ is the uniform transition kernel (which has the second column of $F_{1,h}$ as its latent factor), it follows that $PV_{1,2}^*(\cdot,\cdot)$ is a rank one matrix with every entry being the same positive constant. Thus, $Q_{1,1}^* = r_{1,1} + P_{1,1}V_{1,2}^*$ and $Q_{1,2}^* = r_{1,2}$ are rank two matrices with $F_{1,h}$ as latent factors.

The second source MDP has the following latent factors

$$F_{2,h} = \begin{bmatrix} \frac{1}{c\sqrt{2n}} + \frac{1}{c\beta\sqrt{4n}} & 1/\sqrt{4n} \\ -\frac{1}{c\sqrt{2n}} + \frac{1}{c\beta\sqrt{4n}} & 1/\sqrt{4n} \\ -1/(c\beta\sqrt{4n}) & 1/\sqrt{4n} \\ -1/(c\beta\sqrt{4n}) & 1/\sqrt{4n} \end{bmatrix}, W_{2,1} = \begin{bmatrix} 1/\sqrt{2n} & 0 \\ 0 & 1/\sqrt{2n} \end{bmatrix}, W_{2,2} = \begin{bmatrix} 0 & 1/\sqrt{2n} \\ 1/\sqrt{2n} & 0 \end{bmatrix}$$

where $c = \sqrt{1+1/\beta^2}$ for all $h \in [H]$ with $P_{2,h}(\cdot|\cdot,\cdot) = 1/(4n)$. Since $P_{2,h}$ is the uniform transition kernel (which has the second column of $F_{2,h}$ as its latent factor), it follows that $PV_{2,2}^*(\cdot,\cdot)$ is a rank one matrix with every entry being the same positive constant. Thus, $Q_{2,1}^* = r_{2,1} + P_{2,1}V_{2,2}^*$ and $Q_{2,2}^* = r_{2,2}$ are rank two matrices with $F_{2,h}$ as latent factors. Now, we prove that $\alpha \in \Omega(\beta)$.

**Corollary 1.** *Consider the above construction that satisfies Assumptions 5 and 6. Then, $\alpha$ defined in Definition 4 satisfies $\alpha \in \Omega(\beta)$.*

*Proof.* Consider the above construction with $\beta > 0$. First, we show that when expressing the first column of $F_h$ as a linear combination of the columns of $F_{1,h}$ and $F_{2,h}$, one coefficient is $\Omega(\beta)$.

We first note that one must use the first column in $F_{2,h}$ or else the system of linear equations is inconsistent. Thus, we solve

$$a \begin{bmatrix} 1/\sqrt{2n} \\ -1\sqrt{2n} \\ 0 \\ 0 \end{bmatrix} + b \begin{bmatrix} 1/\sqrt{4n} \\ 1/\sqrt{4n} \\ 1/\sqrt{4n} \\ 1/\sqrt{4n} \end{bmatrix} + d \begin{bmatrix} \frac{1}{c\sqrt{2n}} + \frac{1}{c\beta\sqrt{4n}} \\ -\frac{1}{c\sqrt{2n}} + \frac{1}{c\beta\sqrt{4n}} \\ -1/(c\beta\sqrt{4n}) \\ -1/(c\beta\sqrt{4n}) \end{bmatrix} = \begin{bmatrix} 1/\sqrt{22n} \\ 1/\sqrt{n} \\ 0 \\ 0 \end{bmatrix}$$

It follows that $d = bc\beta$. Thus, $b = 1/\sqrt{2}$, and $d \in \Omega(\beta)$ because $c \geq 1$.

To show that $\alpha \in \Omega(\beta)$, we assume for sake of contradiction that there exists some set of basis vectors $F_{1,h}'$ and $F_{2,h'}$ that span the subspace defined by $F_{1,h}$ and $F_{2,h}$, respectively, such that we can express the any rotation of the first column of $F_h$ as a linear combination of the columns of $F_{1,h}'$ and $F_{2,h'}$ using coefficients $a, b, c, d$ such that $|a|, |b|, |c|, |d| \in o(\beta)$. Since $F_h, F_{1,h}$, and $F_{2,h}$ are orthonormal latent factors, it follows that there exist rotation matrices $R, R_1, R_2$ with $\|R\|_\infty, \|R_1\|_\infty, \|R_2\|_\infty \in O(1)$ such that $F_{1,h}' = R_1 F_{1,h}$ and $F_{2,h}' = R_2 F_{2,h}$. It follows that $R^{-1}$ exists and has entries with constant magnitude. By construction we have

$$F_h(:,0) = aR^{-1}R_1F_{1,h}(:,0) + bR^{-1}R_1F_{1,h}(:,1) + cR^{-1}R_2F_{2,h}(:,0) + dR^{-1}R_2F_{2,h}(:,1).$$

Since the entries of the rotation matrices and their inverse are bounded by a constant, the above equation states that we can represent the first column of $F_h$ as a linear combination of the columns of $F_{1,h}$ and $F_{2,h}$ using only constant coefficients. Thus, we reach a contradiction because we've shown that one entry of $dR^{-1}R_2 \in \Omega(\beta)$. Therefore, $\alpha \in \Omega(\beta)$. $\square$

# E   Omitted Proofs in the $(S, S, d)$ Tucker Rank Setting

We first prove our lower bound.

### E.1 Proof of Theorem 1

*Proof.* Suppose all source and target MDPs $(S, A, P, H, r)$ share the same state space, action space, and horizon $H = 1$ and assume that $\mathcal{S} = \mathcal{A} = [2n]$ for some $n \in \mathbb{N}_+$. For ease of notation, we let $s_1$ and $a_1$ refer to any state and action in $[n]$, respectively, and $s_2$ and $a_2$ refer to any state and action in $\{n + 1, \ldots 2n\}$, respectively. We will present the latent factors and optimal Q functions as block vectors and matrices with $s_1, s_2, a_1, a_2$ as the blocks containing $n$ entries. The initial state distribution in the target MDP is uniform over $\mathcal{S}$. We now present two transfer RL problems with similar $Q$ functions (with rows $s_1, s_2$ and columns $a_1, a_2$) that satisfy Assumptions 1 and 2 but have orthogonal target state latent factors. For ease of notation, the superscript $i$ of $Q_{m,h}^{*,i}, Q_h^{*,i}$ denotes transfer RL problem for $i \in \{1, 2\}$. Then, the optimal $Q$ functions for transfer RL problem one are

$$Q_{1,1}^{*,1} = n \begin{bmatrix} \sqrt{1/n} \\ 0 \end{bmatrix} \begin{bmatrix} \sqrt{1/(2n)} & \sqrt{1/(2n)} \end{bmatrix} = \begin{bmatrix} 1/\sqrt{2} & 1/\sqrt{2} \\ 0 & 0 \end{bmatrix},$$

$$Q_{2,1}^{*,1} = n \begin{bmatrix} 0 \\ \sqrt{1/n} \end{bmatrix} \begin{bmatrix} \sqrt{1/(2n)} & \sqrt{1/(2n)} \end{bmatrix} = \begin{bmatrix} 0 & 0 \\ 1/\sqrt{2} & 1/\sqrt{2} \end{bmatrix},$$

$$Q_1^{*,1} = n \begin{bmatrix} \sqrt{1/(2n)} \\ -\sqrt{1/(2n)} \end{bmatrix} \begin{bmatrix} \sqrt{1/(2n)} & \sqrt{1/(2n)} \end{bmatrix} = \begin{bmatrix} 1/2 & 1/2 \\ -1/2 & -1/2 \end{bmatrix},$$

and the optimal $Q$ functions for transfer RL problem two are

$$Q_{1,1}^{*,2} = n \begin{bmatrix} \sqrt{1/n} \\ 0 \end{bmatrix} \begin{bmatrix} \sqrt{1/(2n)} & \sqrt{1/(2n)} \end{bmatrix} = \begin{bmatrix} 1/\sqrt{2} & 1/\sqrt{2} \\ 0 & 0 \end{bmatrix},$$

$$Q_{2,1}^{*,2} = n \begin{bmatrix} 0 \\ \sqrt{1/n} \end{bmatrix} G' = \begin{bmatrix} 0 & 0 \\ (1 + 1/\alpha)/(c\sqrt{2}) & (1 - 1/\alpha)/(c\sqrt{2}) \end{bmatrix},$$

$$Q_1^{*,2} = n \begin{bmatrix} \sqrt{1/(2n)} \\ -\sqrt{1/(2n)} \end{bmatrix} \begin{bmatrix} \sqrt{\frac{1}{(2n)}} & -\sqrt{\frac{1}{(2n)}} \end{bmatrix} = \begin{bmatrix} 1/2 & -1/2 \\ -1/2 & 1/2 \end{bmatrix},$$

where

$$G' = \begin{bmatrix} (\sqrt{1/(2n)} + \sqrt{1/(2n\alpha^2)})/\sqrt{1 + 1/\alpha^2} & (\sqrt{1/(2n)} - \sqrt{1/(2n\alpha^2)})/\sqrt{1 + 1/\alpha^2} \end{bmatrix},$$

and $c = \sqrt{1 + 1/\alpha^2}$. Note that $c \in (1, 2dM)$ because $\alpha \geq 1/(dM)$ from Lemma 1 for $\alpha > 0$. The incoherence $\mu$, rank $d$, and condition number $\kappa$ of the above $Q$ functions are $O(1)$. Furthermore, the above construction satisfies Assumption 5 with Tucker rank $(S, S, 1)$ and Assumption 6 because the source and target MDPs share the same action latent factor $\begin{bmatrix} \sqrt{1/(2n)} & \sqrt{1/(2n)} \end{bmatrix}$ in the first transfer RL problem while in the second transfer RL problem,

$$\begin{bmatrix} \sqrt{1/(2n)} & -\sqrt{1/(2n)} \end{bmatrix} = -\alpha \begin{bmatrix} \sqrt{1/(2n)} & \sqrt{1/(2n)} \end{bmatrix} + \alpha c G'.$$

Note that $G^1 = \begin{bmatrix} \sqrt{1/(2n)} & \sqrt{1/(2n)} \end{bmatrix}$ and $G^2 = \begin{bmatrix} \sqrt{1/(2n)} & -\sqrt{1/(2n)} \end{bmatrix}$, so the target latent factors of the different transfer RL problems are orthogonal to each other.

The learner is given either transfer RL problem one or two with equal probability with a generative model in the source problem. When interacting with the source MDPs, the learner specifies a state-action pair $(s, a)$ and source MDP $m$ and observes a realization of a shifted and scaled Bernoulli random variable $X$. The distribution of $X$ is that $X = 1$ with probability $(Q_{m,1}^{*,i}(s, a) + 1)/2$ and $X = -1$ with probability $(1 - Q_{m,1}^{*,i}(s, a))/2$. Furthermore, the learner is given the knowledge that the state latent features lie in the function class $\mathcal{G} = \{G^1, G^2\}$ and must use the knowledge obtained from interacting with the source MDP to choose one to use in the target MDP, which is no harder than receiving no information about the function class.

To distinguish between the two transfer RL problems, one needs to differentiate between $Q_{2,1}^{*,1}$ and $Q_{2,1}^{*,2}$. By construction, the magnitude of the largest entrywise difference between $Q_{2,1}^{*,1}$ and $Q_{2,1}^{*,2}$ is lower bounded by $\Omega(1/\alpha)$.

If the learner observes $Z$ samples in the source MDPs, where $Z \leq O(\alpha^2)$ for some constant $C > 0$, then the probability of correctly identifying $G$ is upper bounded by $0.76$ (Lemma 5.1 with $\delta = 0.24$ [5]). Thus, if the learner does not observe $\Omega(\alpha^2)$ samples in the source phase, then the learner returns the incorrect feature mapping that is orthogonal to the true feature mapping with probability at least $0.24$. □

## E.2 Proof of Theorem 2

We first present the theoretical guarantees of LSVI-UCB-(S, S, d) and defer their proofs to Appendix J.

**Theorem 5.** *Assume that Assumption 1 holds and the learner is given the true latent action representation $G = \{G_h\}_{h \in [H]}$. Then, there exists a constant $c > 0$ such that for any $\delta \in (0, 1)$, if we set $\lambda = 1, \beta_{k,h}^s = cdH\sqrt{\iota}, \iota = \log(2dTS/\delta)$, then with probability at least $1 - \delta$, the total regret of Algorithm 2 is $\tilde{O}(\sqrt{d^3SH^3T})$.*

Theorem 5 states that LSVI-UCB-(S, S, d) completely removes the regret bound's dependence on the size of the state space by utilizing the latent action representation $G$. Similarly, the algorithm is robust to misspecification error.

**Assumption 4** ($\xi$-approximate $(S, S, d)$ Tucker rank MDP). *Assume $\xi \in [0, 1]$. Then, an MDP $(\mathcal{S}, \mathcal{A}, P, r, H)$ is a $\xi$-approximate $(S, S, d)$ Tucker rank MDP with given feature map $G$ if there exist $H$ unknown S-by-d matrices $W = \{W_h\}_{h \in [H]}$ and S-by-S-by-d tensors $U = \{U_h\}_{h \in [H]}$ such that*

$$|r_h(s, a) - W_h(s)^\top G_h(a)| \leq \xi,$$

$$\left| \sum_{s' \in S} (P_h(s'|s, a) - U_h(s', s)G_h(a)^\top) \right| \leq \xi$$

*for all $h \in [H]$ where $\|W_h\|_2, \|\sum_{s' \in S} g(s')U_h(s', a)\|_2 \leq \sqrt{A/(d\mu)}$ for any function $g : \mathcal{S} \to [0, 1]$.*

**Theorem 6.** *Under the Assumption 4, for any $\delta \in (0, 1)$, for $\lambda = 1, \beta_{k,h}^s = O(Hd\sqrt{\iota} + \xi\sqrt{kdH})$ for $\iota = \log(2dTS/\delta)$, the regret of Algorithm 2 is $\tilde{O}(\sqrt{d^3H^3ST} + \xi dHT)$ with probability at least $1 - \delta$.*

We now prove Theorem 2.

*Proof.* Suppose the assumption stated in Theorem 2 hold. Then, from [30], running LR-EVI with a sample complexity of $C'd^4\mu^5\kappa^4(S + A)MH^4\alpha^2T\log(2SAdMH/\delta)/S$ on each source MDP results in $\bar{Q}_{m,h}$ functions with singular value decomposition $\bar{Q}_{m,h} = \hat{F}_{m,h}\hat{\Sigma}_{m,h}\hat{G}_{m,h}$ that are $\sqrt{HS}/(16\alpha\mu\kappa\sqrt{dMT})$-optimal with probability at least $1 - \delta/2$.

Since our estimates are $\sqrt{HS}/(16\alpha\kappa\sqrt{dMT})$-optimal, it follows that $\bar{\gamma} < 1/2$ in the singular vector perturbation bound, Corollary 3. Since $\|Q_{m,h}^*\|_\infty \geq C$, from Corollary 3, it follows that there exists rotation matrices $R_{m,h}$ such that

$$\|(\hat{G}_{m,h}R_{m,h} - G_{m,h})(a)\|_2 \leq \frac{d\sqrt{H\mu S}}{\alpha\sqrt{AMT}}$$

for all $a \in A$ where the optimal Q function have singular value decomposition $Q_{m,h}^* = F_{m,h}\Sigma_{m,h}G_{m,h}^\top$.

Let $\tilde{G}$ be the $A \times dM$ matrix from joining all the source action latent factor matrices, and $\hat{\tilde{G}}_h$ be the $A \times dM$ matrix from joining all $\bar{G}_{m,h}$. Let $R_h$ be the $d \times dM$ matrix that joins $R_{m,h}$ for all $m \in [M]$. Then, from Assumption 2 and Definition 3, it follows that for all $(s', s, a) \in \mathcal{S} \times \mathcal{S} \times A$, that

$$r_h(s, a) = \sum_{i \in [d]} W_h(s, i)G_h(a, i)$$

$$= \sum_{i \in [d]} \left( \sum_{j \in [d], m \in [M]} B(i, j, m)G_{h,m}(a, j) \right) W_h(s, i)$$

$$= \sum_{i,j \in [d], m \in [M]} B(i, j, m)G_{m,h}(a, j)W_h(s, i)$$

$$= \tilde{W}_h'(s)G_h(a)^\top$$

and $P(s'|\cdot, \cdot) = \tilde{G}_h U'_h(s')^\top$ (with the same argument) for some $S \times dM$ matrices $W'_h, U'_h(s')$ with entries bounded by $\alpha$. Note that $W'_h$ and $U'_h(s')$ are matrices that join $W_{m,h}$ and $U_{m,h}(s')$, respectively, with entries that at most $\alpha$ times larger than the original entry.

Therefore, it follows for all $(s', s, a, h) \in \mathcal{S} \times \mathcal{S} \times \mathcal{A} \times [H]$,

$$
\begin{aligned}
|r_h(s,a) - \hat{\tilde{G}}_h(a) R_h W'_h(s)^\top| &= |\tilde{G}_h(a) W'_h(s)^\top - \hat{\tilde{G}}_h(a) R_h W'_h(s)^\top| \\
&= |(\tilde{G}_h(a) - \hat{\tilde{G}}_h(a) R_h) W'_h(s)| \\
&\leq \alpha | \sum_{m \in M} (G_{m,h}(a) - \hat{G}_{m,h}(a) R_{m,h}) W_{m,h}(s)| \\
&\leq \alpha \sum_{m \in M} \|G_{m,h}(a) - \hat{G}_{m,h}(a) R_{m,h}\|_2 \|W_{m,h}(s)\|_2 \\
&\leq \alpha \sqrt{\frac{A}{d\mu}} \sum_{m \in M} \|G_{m,h}(a) - \hat{G}_{m,h}(a) R_{m,h}\|_2 \\
&\leq \sqrt{\frac{dHMS}{T}}
\end{aligned}
$$

from the Cauchy-Schwarz inequality and our singular vector perturbation bound. From the same logic,

$$
\begin{aligned}
\left| \sum_{s' \in S} (P_h(s'|s,a) - U_h(s', s) G_h(a)^\top) \right| &= | \sum_{s' \in S} \tilde{G}_h(a) U'_h(s', s) - \hat{\tilde{G}}_h(a) R_h U'_h(s', s)^\top| \\
&= |(\tilde{G}_h(a) - \hat{\tilde{G}}_h(a) R_h) \sum_{s' \in S} U'_h(s', s)^\top| \\
&\leq \alpha | \sum_{m \in M} (G_{m,h}(a) - \hat{G}_{m,h}(a) R_{m,h}) \sum_{s' \in S} U_{m,h}(s', s)| \\
&\leq \alpha \sum_{m \in M} \|(G_{m,h}(a) - \hat{G}_{m,h}(a) R_{m,h})\|_2 \| \sum_{s' \in S} U_{m,h}(s', s)^\top\|_2 \\
&\leq \sqrt{\frac{dHMS}{T}}.
\end{aligned}
$$

Therefore, $\hat{\tilde{G}}_h(a)$ satisfies Assumption 4 with $\xi = \sqrt{\frac{dMHS}{T}}$. Then it follows that running LSVI-UCB-TR using $\hat{\tilde{G}}_h(a)$ with $\delta' = \delta/2$ during the target phase of the algorithm admits a regret bound of

$$
Regret(T) \leq C''(\sqrt{d^3 M^3 H^3 ST} + dMHT\sqrt{\frac{dMHS}{T}}) \in \tilde{O}(\sqrt{d^3 M^3 H^3 ST})
$$

with probability at least $1 - \delta$ from a final union bound. Furthermore, the sample complexity in the source phase is

$$
\tilde{O}\left( \frac{d^4 \mu^5 \kappa^4 (S + A) M^2 H^4 \alpha^2 T}{S} \right),
$$

which proves our result. $\qquad \square$

### E.3 Proof of Theorem 3

*Proof.* Let the assumptions of Theorem 3 hold. Then, from [30], running LR-EVI with a sample complexity of $C'd^7\mu^5\kappa^4 A(S + A)M^2 H^4 \alpha^2 T \log(2SAdMH/\delta)/(Sd'')$ on each source MDP results in $\bar{Q}_{m,h}$ functions with singular value decomposition $\bar{Q}_{m,h} = \hat{F}_{m,h} \hat{\Sigma}_{m,h} \hat{G}_{m,h}$ that are $\sqrt{Hd''S}/(16\alpha\mu M\kappa d^2\sqrt{AT})$-optimal with probability at least $1 - \delta/2$.

Since our estimates are $\sqrt{Hd''S}/(16\alpha\mu M\kappa d^2\sqrt{AT})$-optimal, it follows that $\bar{\gamma} < 1/2$ in the singular vector perturbation bound, Corollary 3. Since $\|Q^*_{m,h}\|_\infty \geq C$, from Corollary 3, it follows that there

exists rotation matrices $R_{m,h}$ such that

$$\|(\hat{G}_{m,h}R_{m,h} - G_{m,h})(a)\|_2 \le \frac{\sqrt{Hd''\mu S}}{\alpha AM\sqrt{dT}}$$

for all $a \in A$ where the optimal $Q$ function have singular value decomposition $Q^*_{m,h} = F_{m,h}\Sigma_{m,h}G^\top_{m,h}$.

Let $\tilde{G}$ be the $A \times dM$ matrix from joining all the source action latent factor matrices, and $\hat{\tilde{G}}_h$ be the $A \times dM$ matrix from joining all $\tilde{G}_{m,h}$. Let $R_h$ be the $d \times dM$ matrix that joins $R_{m,h}$ for all $m \in [M]$. Then, from Assumption 2 and Definition 3, it follows that for all $(s', s, a) \in \mathcal{S} \times \mathcal{S} \times A$, that

$$r_h(s,a) = \sum_{i \in [d]} W_h(s,i)G_h(a,i)$$

$$= \sum_{i \in [d]} \left( \sum_{j \in [d], m \in [M]} B(i,j,m)G_{h,m}(a,j) \right) W_h(s,i)$$

$$= \sum_{i,j \in [d], m \in [M]} B(i,j,m)G_{m,h}(a,j)W_h(s,i)$$

$$= \tilde{W}'_h(s)G_h(a)^\top$$

and $P(s'|\cdot,\cdot) = \tilde{G}_h U'_h(s')^\top$ (with the same argument) for some $S \times dM$ matrices $W'_h, U'_h(s')$ with entries bounded by $\alpha$. Note that $W'_h$ and $U'_h(s')$ are matrices that join $W_{m,h}$ and $U_{m,h}(s')$, respectively, with entries that at most $\alpha$ times larger than the original entry.

Let $E = \hat{\tilde{G}}_h - \tilde{G}_h$, where $E = [E_1 \ldots E_M]$. We next bound the $d'' + 1$ smallest singular value of $\hat{\tilde{G}}_h$ with our singular vector perturbation bound ($\|E\|_\infty \le \|E\|_{2,\infty}$).

$$\sigma_{d''+1}(\hat{\tilde{G}}_h) = \sigma_{d''+1}(\tilde{G}_h) + \|E\|_{op} \le \sqrt{AdM}\|E\|_\infty \le \frac{\sqrt{Hd''\mu S}}{\alpha\sqrt{AMT}}$$

With singular value decomposition $\hat{\tilde{G}}_h = X\Sigma Y^\top$, let $G_h^{d''}$ be the thresholded feature representation estimate, i.e., $G_h^{d''} = X(:,: d'')\Sigma(: d'',: d'')Y(:,d'')^\top$. Therefore, it follows for all $(s',s,a,h) \in \mathcal{S} \times \mathcal{S} \times \mathcal{A} \times [H]$,

$$|r_h(s,a) - G_h^{d''}(a)R_hW'_h(s,: d'')^\top|$$

$$= |\tilde{G}_h(a)W'_h(s)^\top - G_h^{d''}(a)R_hW'_h(s,: d'')^\top|$$

$$\le |\tilde{G}_h(a)W'_h(s)^\top - \hat{\tilde{G}}_h(a)R_hW'_h(s)^\top + |\hat{\tilde{G}}_h(a)R_hW'_h(s)^\top - G_h^{d''}(a)R_hW'_h(s,: d'')^\top|$$

$$\le \alpha| \sum_{m \in M} (G_{m,h}(a) - \hat{G}_{m,h}(a)R_{m,h})W_{m,h}(s)| + |\hat{\tilde{G}}_h(a, d''+1 :)W'_h(s, d''+1 :)^\top|$$

$$\le \alpha \sum_{m \in M} \|G_{m,h}(a) - \hat{G}_{m,h}(a)R_{m,h}\|_2\|W_{m,h}(s)\|_2 + \alpha\sigma_{d''+1}(\hat{\tilde{G}}_h)\sqrt{dM} \sum_{m \in [M]} \|W_{m,h}\|_2$$

$$\le \alpha\sqrt{\frac{A}{d\mu}} \left( \sum_{m \in M} \|G_{m,h}(a) - \hat{G}_{m,h}(a)R_{m,h}\|_2 + \sigma_{d''+1}(\hat{\tilde{G}}_h)\sqrt{dM} \right)$$

$$\le \sqrt{\frac{d''HS}{T}}$$

from the Cauchy-Schwarz inequality and our singular vector perturbation bound. From the same logic,

$$\left| \sum_{s' \in S} (P_h(s'|s,a) - U_h(s',s)G_h^{d''}(a)^\top) \right| \le \sqrt{\frac{d''HS}{T}}.$$

Therefore, $G_h^{d''}(a)$ satisfies Assumption 4 with $\xi = \sqrt{\frac{d''HS}{T}}$. Then it follows that running LSVI-UCB-TR using $G_h^{d''}(a)$ with $\delta' = \delta/2$ during the target phase of the algorithm admits a regret bound of

$$Regret(T) \leq C''(\sqrt{(d'')^3 H^3 ST} + d''HT\sqrt{\frac{d''HS}{T}}) \in \tilde{O}(\sqrt{(d'')^3 H^3 ST})$$

with probability at least $1 - \delta$ from a final union bound. Furthermore, the sample complexity in the source phase is

$$\tilde{O}\left(\frac{d^7 \mu^5 \kappa^4 (A + A^2) M^3 H^4 \alpha^2 T}{Sd''}\right),$$

which proves our result.

$\square$

We now prove Theorem 4.

*Proof.* Suppose the assumption stated in Theorem 4 hold. Then, from [30], running LR-EVI with a sample complexity of $C'd^5\mu^5\kappa^4(S + A)M^2 H^4 \alpha^2 T \log(2SAdMH/\delta)/S$ on each source MDP results in $\bar{Q}_{m,h}$ functions with singular value decomposition $\bar{Q}_{m,h} = \hat{F}_{m,h}\hat{\Sigma}_{m,h}\hat{G}_{m,h}$ that are $\sqrt{HS}/(16\alpha\mu\kappa\sqrt{d^2 M^2 T})$-optimal with probability at least $1 - \delta/2$.

Since our estimates are $\sqrt{HS}/(16\alpha\kappa\sqrt{d^2 M^2 T})$-optimal, it follows that $\bar{\gamma} < 1/2$ in the singular vector perturbation bound, Corollary 3. Since $\|Q_{m,h}^*\|_\infty \geq C$, from Corollary 3, it follows that there exists rotation matrices $R_{m,h}$ such that

$$\|(\hat{G}_{m,h}R_{m,h} - G_{m,h})(a)\|_2 \leq \frac{\sqrt{Hd\mu S}}{\alpha\sqrt{AM^2 T}}$$

for all $a \in A$ where the optimal $Q$ function have singular value decomposition $Q_{m,h}^* = F_{m,h}\Sigma_{m,h}G_{m,h}^\top$.

Let $\tilde{G}$ be the $A \times dM$ matrix from joining all the source action latent factor matrices, and $\hat{\tilde{G}}_h$ be the $A \times dM$ matrix from joining all $\bar{G}_{m,h}$. Let $R_h$ be the $d \times dM$ matrix that joins $R_{m,h}$ for all $m \in [M]$. Then, from Assumption 2 and Definition 3, it follows that for all $(s', s, a) \in \mathcal{S} \times \mathcal{S} \times A$, that

$$r_h(s, a) = \sum_{i \in [d]} W_h(s, i)G_h(a, i)$$

$$= \sum_{i \in [d]} \left(\sum_{j \in [d], m \in [M]} B(i, j, m)G_{h,m}(a, j)\right) W_h(s, i)$$

$$= \sum_{i, j \in [d], m \in [M]} B(i, j, m)G_{m,h}(a, j)W_h(s, i)$$

$$= \tilde{W}_h'(s)G_h(a)^\top$$

and $P(s'|\cdot, \cdot) = \tilde{G}_h U_h'(s')^\top$ (with the same argument) for some $S \times dM$ matrices $W_h', U_h'(s')$ with entries bounded by $\alpha$. Note that $W_h'$ and $U_h'(s')$ are matrices that join $W_{m,h}$ and $U_{m,h}(s')$, respectively, with entries that at most $\alpha$ times larger than the original entry. Let $\hat{\tilde{G}}^t$ be the thresholded feature mapping, where $t$ is defined as $t = \min_{t' \in \mathcal{T}} t'$ for

$$\mathcal{T} = \left\{t \in [dM] | \sigma_{t+1} \leq \sqrt{\frac{tHSd\mu}{\alpha^2 T(dM - t)M^2 A}}\right\}.$$

Therefore, with the same logic as the previous proof, it follows for all $(s', s, a, h) \in \mathcal{S} \times \mathcal{S} \times \mathcal{A} \times [H]$,

$$
\begin{aligned}
|r_h(s,a) - \hat{\tilde{G}}^t(a) R_h W'_h(s)^\top| &= |\tilde{G}_h(a) W'_h(s)^\top - \hat{\tilde{G}}^t R_h W'_h(s)^\top| \\
&\le |(\tilde{G}_h(a) - \hat{\tilde{G}}_h(a) R_h) W'_h(s)| + |\hat{\tilde{G}}_h(a) R_h W'_h(s)^\top - \hat{\tilde{G}}^t_h(a)(R_h W'_h(s))^{t,\top}| \\
&\le \sqrt{\frac{HS}{T}} + |\hat{\tilde{G}}_h(a, t+1:)(R_h W'_h)(s, t+1:)^\top| \\
&\le \sqrt{\frac{HS}{T}} + \sigma_{t+1} \alpha \sqrt{dM-t} \|W'_h\|_2 \\
&\le \sqrt{\frac{HS}{T}} + \sqrt{\frac{tHS}{T}} \\
&\le 2\sqrt{\frac{tHS}{T}}
\end{aligned}
$$

where the second and third inequalities comes from the same logic used in the proofs of Theorems 2 and 4, and the fourth inequality comes from the definition of $t$. From the same logic,

$$
\left| \sum_{s' \in S} (P_h(s'|s,a) - U_h(s',s) \hat{\tilde{G}}^t(a)^\top) \right| \le 2\sqrt{\frac{tHS}{T}}.
$$

Therefore, $\hat{\tilde{G}}_h(a)$ satisfies Assumption 4 with $\xi = \sqrt{\frac{tHS}{T}}$. Then it follows that running LSVI-UCB-TR using $\hat{\tilde{G}}^t_h(a)$ with $\delta' = \delta/2$ during the target phase of the algorithm admits a regret bound of

$$
Regret(T) \le C''(\sqrt{t^3 H^3 ST} + 2tHT\sqrt{\frac{tHS}{T}}) \in \tilde{O}(\sqrt{t^3 H^3 ST})
$$

with probability at least $1 - \delta$ from a final union bound. Furthermore, the sample complexity in the source phase is

$$
\tilde{O}\left(\frac{d^5 \mu^5 \kappa^4 (S+A) M^3 H^4 \alpha^2 T}{S}\right),
$$

which proves our result. $\qquad\square$

## F  $(S, d, A)$ Tucker Rank Setting

In this section, we present the assumptions, algorithm, and results for the $(S, d, A)$ Tucker rank setting. In this setting, we assume the following structure on the source and target MDPs.

**Assumption 5.** *In each of the $M$ source MDPs, the reward functions have rank $d$, and the transition kernels have Tucker rank $(S, d, A)$. Let the target MDP's reward functions have rank $d'$ and transition kernels have Tucker rank $(S, d', A)$ where $d' \le dM$. Thus, there exists $S \times d \times A$ tensors $U_{m,h}$, $S \times d' \times A$ tensors $U_h$, $S \times d$ $\mu$-incoherent matrices $F_{m,h}$, $S \times d'$ $\mu$-incoherent matrices $F_h$ with orthonormal columns, and $A \times d$ matrices $W_{m,h}$, $A \times d'$ matrices $W_h$ such that*

$$
P_{m,h}(s'|s,a) = \textstyle\sum_{i \in [d]} F_{m,h}(s,i) U_{m,h}(s',a,i), \quad r_{m,h}(s,a) = \textstyle\sum_{i \in [d]} F_{m,h}(s,i) W_{m,h}(a,i)
$$

*and*

$$
P_h(s'|s,a) = \textstyle\sum_{i \in [d']} F_h(s,i) U_h(s',a,i), \quad r_h(s,a) = \textstyle\sum_{i \in [d']} F_h(s,i) W_h(a,i)
$$

*where* $\|\sum_{s' \in \mathcal{S}} g(s') U_{m,h}(s',:,a)\|_2$, $\|\sum_{s' \in \mathcal{S}} g(s') U_h(s',:,a)\|_2, \|W_h(a)\|_2, \|W_{m,h}(a)\|_2 \le \sqrt{S/(d\mu)}$ *for all* $s \in \mathcal{S}$, $a \in \mathcal{A}$, $h \in [H]$, *and* $m \in [M]$ *for any function* $g : \mathcal{S} \to [0,1]$.

Similarly, to allow transfer of a latent representation between the source and target MDPs, we assume that the subspace spanned by the set of all source latent factors contains the space spanned by the set of target latent factors.

**Assumption 6.** *Suppose Assumption 5 holds. The target MDP latent factors $F_h$ and source MDP latent factors $F_{m,h}$ satisfy for all $h \in [H]$,* $\text{Span}(\{F_h(:,i)\}_{i \in [d']}) \subseteq \text{Span}(\{F_{m,h}(:,i)\}_{i \in [d], m \in [M]})$.

Before defining our transfer-ability coefficient, we first introduce the following notation. We define the set $\mathcal{B}_h(F_h, \{F_{h,m}\}_{m \in M})$ to contain all coefficients of the linear combinations, i.e., $B_h \in \mathbb{R}^{d',d,M}$ and $B_h \in \mathcal{B}_h(F_h, \{F_{h,m}\}_{m \in M})$ if for all $i \in [d'], h \in [H]$, $F_h(\cdot, i) = \sum_{j \in [d], m \in [M]} B_h(i, j, m) F_{m,h}(\cdot, j)$. Furthermore, as the source MDPs latent factors of $Q^*_{m,h}$ are not unique, we define $\alpha$ to measure the difficulty given the best set of latent factors from the source MDPs. Thus, $\alpha$ precisely measures the challenge involved in transferring the latent representation.

**Definition 4** (Transfer-ability Coefficient). *Given a transfer RL problem that satisfies Assumptions 5, 6, and 3, let $\mathcal{F}$ be the set of all $\mathbb{R}^{S \times d}$ matrices with orthonormal columns that span the column space of $Q^*_{m,h}$ for all $m \in [M], h \in [H]$. Then, we define $\alpha$ as*

$$\alpha := \max_{h \in [H]} \min_{F \in \mathcal{F}} \min_{B \in \mathcal{B}_h(F_h, F)} \max_{i \in [d'], j \in [d], m \in [M]} |B(i, j, m)|.$$

We now prove our lower bound in this Tucker rank setting.

### F.1 Information Theoretic Lower Bound

To formalize the importance of $\alpha$ in transfer learning, we prove a lower bound that shows that a dependence on $\alpha$ in the sample complexity of the source phase is necessary to benefit from transfer learning.

**Theorem 7.** *There exist two transfer RL instances such that (i) they satisfy Assumptions 5 and 6, (ii) they cannot be distinguished without observing $\Omega(\alpha^2)$ samples in the source phase, and (3) they have target state latent features that are orthogonal to each other.*

*Proof.* Suppose all source and target MDPs $(S, A, P, H, r)$ share the same state space, action space, and horizon $H = 1$ and assume that $\mathcal{S} = \mathcal{A} = [2n]$ for some $n \in \mathbb{N}_+$. For ease of notation, we let $s_1$ and $a_1$ refer to any state and action in $[n]$, respectively, and $s_2$ and $a_2$ refer to any state and action in $\{n + 1, \ldots 2n\}$, respectively. We will present the latent factors and optimal Q functions as block vectors and matrices with $s_1, s_2, a_1, a_2$ as the blocks containing $n$ entries. The initial state distribution in the target MDP is uniform over $\mathcal{S}$. We now present two transfer RL problems with similar $Q$ functions (with rows $s_1, s_2$ and columns $a_1, a_2$) that satisfy Assumptions 5 and 6 but have orthogonal target state latent factors. For ease of notation, the superscript $i$ of $Q^{*,i}_{m,h}, Q^{*,i}_h$ denotes transfer RL problem. Then, the optimal $Q$ functions for transfer RL problem one are

$$Q^{*,1}_{1,1} = n \begin{bmatrix} \sqrt{1/(2n)} \\ \sqrt{1/(2n)} \end{bmatrix} \begin{bmatrix} \sqrt{1/n} & 0 \end{bmatrix} = \begin{bmatrix} 1/\sqrt{2} & 0 \\ 1/\sqrt{2} & 0 \end{bmatrix}, \quad Q^{*,1}_{2,1} = n \begin{bmatrix} \sqrt{1/(2n)} \\ \sqrt{1/(2n)} \end{bmatrix} \begin{bmatrix} 0 & \sqrt{1/n} \end{bmatrix} = \begin{bmatrix} 0 & 1/\sqrt{2} \\ 0 & 1/\sqrt{2} \end{bmatrix},$$

$$Q^{*,1}_1 = n \begin{bmatrix} \sqrt{1/(2n)} \\ \sqrt{1/(2n)} \end{bmatrix} \begin{bmatrix} \sqrt{\frac{1}{n}} & -\sqrt{\frac{1}{n}} \end{bmatrix} = \begin{bmatrix} 1/2 & -1/2 \\ 1/2 & -1/2 \end{bmatrix},$$

and the optimal $Q$ functions for transfer RL problem two are

$$Q^{*,2}_{1,1} = n \begin{bmatrix} \sqrt{1/(2n)} \\ \sqrt{1/(2n)} \end{bmatrix} \begin{bmatrix} \sqrt{1/n} & 0 \end{bmatrix} = \begin{bmatrix} 1/\sqrt{2} & 0 \\ 1/\sqrt{2} & 0 \end{bmatrix} \quad Q^{*,2}_{2,1} = nF' \begin{bmatrix} 0 & \sqrt{1/n} \end{bmatrix} = \begin{bmatrix} 0 & (1 + 1/\alpha)/(c\sqrt{2}) \\ 0 & (1 - 1/\alpha)/(c\sqrt{2}) \end{bmatrix},$$

$$Q^{*,2}_1 = n \begin{bmatrix} \sqrt{1/(2n)} \\ -\sqrt{1/(2n)} \end{bmatrix} \begin{bmatrix} \sqrt{\frac{1}{n}} & -\sqrt{\frac{1}{n}} \end{bmatrix} = \begin{bmatrix} 1/2 & -1/2 \\ -1/2 & 1/2 \end{bmatrix},$$

where

$$F' = \begin{bmatrix} (\sqrt{1/(2n)} + \sqrt{1/(2n\alpha^2)})/\sqrt{1 + 1/\alpha^2} & (\sqrt{1/(2n)} - \sqrt{1/(2n\alpha^2)})/\sqrt{1 + 1/\alpha^2} \end{bmatrix}$$

and $c \in (1, 2dM)$ because $\alpha \geq 1/(dM)$ from Lemma 1 for $\alpha > 0$. The incoherence $\mu$, rank $d$, and condition number $\kappa$ of the above $Q$ functions are $O(1)$. Furthermore, the above construction satisfies Assumption 5 with Tucker rank $(S, 1, A)$ and Assumption 6 because the source and target MDPs share the same state latent factor $[\sqrt{1/(2n)}, \sqrt{1/(2n)}]$ in the first transfer RL problem while in the second transfer RL problem,

$$\begin{bmatrix} \sqrt{1/(2n)} & -\sqrt{1/(2n)} \end{bmatrix} = -\alpha \begin{bmatrix} \sqrt{1/(2n)} & \sqrt{1/(2n)} \end{bmatrix} + \alpha c F'^\top.$$

Furthermore, note that the target latent factors,

$$F^1 = \begin{bmatrix} \sqrt{1/(2n)} \\ \sqrt{1/(2n)} \end{bmatrix} \quad \text{and} \quad F^2 = \begin{bmatrix} \sqrt{1/(2n)} \\ -\sqrt{1/(2n)} \end{bmatrix}$$

are orthogonal to each other.

The learner is given either transfer RL problem one or two with equal probability with a generative model in the source problem. When interacting with the source MDPs, the learner specifies a state-action pair $(s, a)$ and source MDP $m$ and observes a realization of a shifted and scaled Bernoulli random variable $X$. The distribution of $X$ is that $X = 1$ with probability $(Q_{m,1}^{*,i}(s, a) + 1)/2$ and $X = -1$ with probability $(1 - Q_{m,1}^{*,i}(s, a))/2)$. Furthermore, the learner is given the knowledge that the state latent features lie in the function class $\mathcal{F} = \{F^1, F^2\}$ and must use the knowledge obtained from interacting with the source MDP to choose one to use in the target MDP, which is no harder than receiving no information about the function class.

To distinguish between the two transfer RL problems, one needs to differentiate between $Q_{2,1}^{*,1}$ and $Q_{2,1}^{*,2}$. By construction, the magnitude of the largest entrywise difference between $Q_{2,1}^{*,1}$ and $Q_{2,1}^{*,2}$ is lower bounded by $\Omega(1/\alpha)$.

If the learner observes $Z$ samples in the source MDPs, where $Z \leq O(\alpha^2)$ for some constant $C > 0$, then the probability of correctly identifying $F$ is upper bounded by 0.76 (Lemma 5.1 with $\delta = 0.24$ [5]). Thus, if the learner does not observe $\Omega(\alpha^2)$ samples in the source phase, then the learner returns the incorrect feature mapping that is orthogonal to the true feature mapping with probability at least 0.24. $\qquad\square$

Theorem 7 states that unless one incurs a sample complexity of $\Omega(\alpha^2)$ in the source phase, the latent representation $F$ learned in the source phase is useless in the target phase as one would incur linear regret using $F$.

Our algorithm in this setting is essentially the same as the one used in the $(S, S, d)$ Tucker rank setting except we construct our latent factors using the other set of singular vectors and compute Gram matrices and coefficients for each action instead of each state.

---

**Algorithm 3** Source Phase

---

**Input:** $\{N_h\}_{h \in [H]}$
1: **for** $m \in [M]$ **do**
2:     Run LR-EVI($\{N_h\}_{h \in [H]}$) on source MDP $m$ to obtain $\bar{Q}_{m,h}$.
3:     Compute the singular value decomposition of $\bar{Q}_{m,h} = \hat{F}_{m,h}\hat{\Sigma}_{m,h}\hat{G}_{m,h}^\top$ .
4: Compute feature mapping $\hat{\bar{F}} = \hat{\bar{F}}_h\}_{h \in [H]}$ with $\hat{\bar{F}}_h = \left\{ \sqrt{\frac{S}{d\mu}}\hat{F}_{m,h}(:, i) | i \in [d], m \in [M] \right\}$ .

---

---

**Algorithm 4** Target Phase: LSVI-UCB-(S, d, A)

---

**Input:** $\lambda, \beta_{k,h}^a, \hat{\bar{F}}$
1: **for** $k \in [K]$ **do**
2:     Receive initial state $s_1^k$.
3:     **for** $h = H, \ldots, 1$ **do**
4:         **for** $a \in \mathcal{A}$ **do**
5:             Set $\Lambda_h^a \leftarrow \sum_{t \in T_{k-1,h}^a} \hat{\bar{F}}_h(s_h^t)\hat{\bar{F}}_h(s_h^t)^\top + \lambda\mathbf{I}$.
6:             /* Compute Gram matrix $\Lambda_h^a$ */
7:             Set $w_h^a \leftarrow (\Lambda_h^a)^{-1} \sum_{t \in T_{k-1,h}^a} \hat{\bar{F}}_h(s_h^t) \left[ r_h(s_h^t, a) + \max_{a' \in \mathcal{A}} Q_{h+1}(s_{h+1}^t, a') \right]$.
8:             /* Estimate $w_h^a$ via regularized least squares */
9:         Set $Q_h(\cdot, \cdot) \leftarrow \min \left( H, \langle w_h^\cdot, \hat{\bar{F}}_h(\cdot) \rangle + \beta_{k,h}^\cdot \sqrt{\hat{\bar{F}}_h(\cdot)^\top (\Lambda_h^\cdot)^{-1}\hat{\bar{F}}_h(\cdot)} \right)$.
10:         /* Estimate $w_h^s$ via regularized least squares */
11:     **for** $h \in [H]$ **do**
12:         Take action $a_h^k \leftarrow \max_{a \in \mathcal{A}} Q_h(s_h^k, a)$, and observe $s_{h+1}^k$.

---

We first present the theoretical guarantees of LSVI-UCB-(S, d, A) and defer their proofs to Appendix J.

Before proving our main positive result in the $(S, d, A)$ Tucker rank setting, we first present the definitions and theorems of LSVI-UCB-(S, d, A) and defer their proofs to Appendix J. If one is given the true latent low-rank representation of each state, then LSVI-UCB-(S, d, A) admits the following regret bound.

**Theorem 8.** *Assume that Assumption 5 holds and the learner is given the true latent state representation $F = \{F_h\}_{h \in [H]}$. Then, there exists a constant $c > 0$ such that for any $\delta \in (0, 1)$, if we set $\lambda = 1, \beta_{k,h}^a = cdH\sqrt{\iota}, \iota = \log(2dTA/\delta)$, then with probability at least $1 - \delta$, the total regret of LSVI-UCB-TR is $\tilde{O}(\sqrt{d^3 A H^3 T})$.*

Theorem 8 states that LSVI-UCB-TR completely removes the regret bound's dependence on the size of the state space by utilizing the latent state representation $F$. To measure the algorithm's robustness with respect to misspecification error of the low-rank representation, we first define $\xi$-approximate $(S, d, A)$ Tucker rank MDPs (similarly to Assumption B from [17]).

**Assumption 7** ($\xi$-approximate $(S, d, A)$ Tucker rank MDP). *Let $\xi \in [0, 1]$. Then, an MDP $(\mathcal{S}, \mathcal{A}, P, r, H)$ is a $\xi$-approximate $(S, d, A)$ Tucker rank MDP with given feature map $F$ if there exist $H$ unknown $A$-by-$d$ matrices $W = \{W_h\}_{h \in [H]}$ and $S$-by-$d$-by-$A$ tensors $U = \{U_h\}_{h \in [H]}$ such that*

$$|r_h(s, a) - F_h(s)^\top W_h(a)| \leq \xi,$$

$$\left| \sum_{s' \in S} (P_h(s'|s, a) - F_h(s)^\top U_h(s', a)) \right| \leq \xi$$

*for all $h \in [H]$ where $\|W_h\|_2, \|\sum_{s' \in \mathcal{S}} g(s')U_h(s', a)\|_2 \leq \sqrt{S/(d\mu)}$ for any function $g : \mathcal{S} \to [0, 1]$.*

The following result shows that similar to LSVI-UCB, our algorithm is robust to the misspecified setting.

**Theorem 9.** *Suppose that the learner is given a feature mapping $F'$ that satisfies Assumption 7. Then, using $F = \sqrt{S/(d\mu)}F'$ as the feature mapping for any $\delta \in (0, 1)$ and $\lambda = 1, \beta_{k,h}^a = O(Hd\sqrt{\iota} + \xi\sqrt{d|T_{k,h}^a|}H)$ for $\iota = \log(2dTA/\delta)$, the regret of LSVI-UCB-TR is $\tilde{O}(\sqrt{d^3 H^3 AT} + \xi dHT)$ with probability at least $1 - \delta$.*

The above theorem states that the misspecification error adds a term that is linear in $T$ to the regret bound of LSVI-UCB-(S, d, A). Thus, if one's latent state representation is close enough, i.e., $\xi$ is so small that there exists some positive constant $c$ such that $\xi dHT \leq c\sqrt{d^3 H^3 AT}$, then LSVI-UCB-(S, d, A) admits a regret bound that is independent of the state space, which is the motivation for our main result. We now present our main theorem.

**Theorem 10.** *Suppose Assumptions 5, 6, and 3 hold, and set $\delta \in (0, 1)$. Furthermore, assume that, for any $\epsilon \in (0, \sqrt{AH/T})$, $Q_{m,h} = r_{m,h} + P_{m,h}V_{m,h+1}$ has rank $d$ and is $\mu$-incoherent with condition number $\kappa$ for all $\epsilon$-optimal value functions $V_{h+1}$. Also, let $\lambda = 1$ and $\beta_{k,h}^a$ be a function of $d, H, |T_{k,h}^a|, |A|, M, T$. Then, for $T \geq \frac{A}{\alpha^2}$, using at most $\tilde{O}\left(d^4\mu^5\kappa^4(S/A + 1)M^2H^4\alpha^2T\right)$ samples in the source problems, our algorithm has regret at most $\tilde{O}\left(\sqrt{(dMH)^3AT}\right)$ with probability at least $1 - \delta$.*

Theorem 10 states that using transfer learning improves the performance on the target problem by removing the regret bound's dependence on the state space. Thus, with enough samples from the source MDPs one can recover a regret bound in the target problem that matches the best regret bound for any algorithm in the $(S, d, A)$ Tucker rank setting with **known** latent feature representation $F$ concerning $A$ and $T$. We now prove Theorem 10.

*Proof.* Suppose the assumption stated in Theorem 10 hold. Then, from [30], running LR-EVI with a sample complexity of $C'd^4\mu^5\kappa^4(S + A)MH^4\alpha^2T\log(2SAdMH/\delta)/A$ on each source MDP results in $\bar{Q}_{m,h}$ functions with singular value decomposition $\bar{Q}_{m,h} = \hat{F}_{m,h}\hat{\Sigma}_{m,h}\hat{G}_{m,h}$ that are $\sqrt{HA}/(16\alpha\mu\kappa\sqrt{dMT})$-optimal with probability at least $1 - \delta/2$.

Since our estimates are $\sqrt{HA}/(16\alpha\sqrt{dMT})$-optimal, it follows that $\bar{\gamma} < 1/2$ in the singular vector perturbation bound, Corollary 3. Since $\|Q^*_{m,h}\|_\infty \geq C$, from Corollary 3, it follows that there exists rotation matrices $R_{m,h}$ such that

$$\|(\hat{F}_{m,h}R_{m,h} - F_{m,h})(s)\|_2 \leq \frac{d\sqrt{H\mu A}}{\alpha\sqrt{SMT}}$$

for all $s \in \mathcal{S}$ where the optimal $Q$ function have singular value decomposition $Q^*_{m,h} = F_{m,h}\Sigma_{m,h}G^\top_{m,h}$.

Let $\tilde{F}$ be the $S \times dM$ matrix from joining all the source state latent factor matrices, and $\hat{\tilde{F}}_h$ be the $S \times dM$ matrix from joining all $\bar{F}_{m,h}$. Let $R_h$ be the $d \times dM$ matrix that joins $R_{m,h}$ for all $m \in [M]$. Then, from Assumption 6 and Definition 4, it follows that for all $(s', s, a) \in \mathcal{S} \times \mathcal{S} \times \mathcal{A}$, that

$$r_h(s,a) = \sum_{i \in [d]} F_h(s,i)W_h(a,i)$$

$$= \sum_{i \in [d]} \left( \sum_{j \in [d], m \in [M]} B(i,j,m)F_{h,m}(s,j) \right) W_h(a,i)$$

$$= \sum_{i,j \in [d], m \in [M]} B(i,j,m)F_{m,h}(s,j)W_h(a,i)$$

$$= \tilde{F}_h(s)W'_h(a)^\top$$

and $P(s'|\cdot,\cdot) = \tilde{F}_h U'_h(s')^\top$ (with the same argument) for some $A \times dM$ matrices $W'_h, U'_h(s')$. Note that $W'_h$ and $U'_h(s')$ are matrices that join $W_{m,h}$ and $U_{m,h}(s')$, respectively, with entries that at most $\alpha$ times larger than the original entry.

Therefore, it follows for all $(s', s, a, h) \in \mathcal{S} \times \mathcal{S} \times \mathcal{A} \times [H]$,

$$|r_h(s,a) - \hat{\tilde{F}}_h(s)R_h W'_h(a)^\top| = |\tilde{F}_h(s)W'_h(a)^\top - \hat{\tilde{F}}_h(s)R_h W'_h(a)^\top|$$

$$= |(\tilde{F}_h(s) - \hat{\tilde{F}}_h(s)R_h)W'_h(a)|$$

$$\leq \alpha| \sum_{m \in M} (F_{m,h}(s) - \hat{F}_{m,h}(s)R_{m,h})W_{m,h}(a)|$$

$$\leq \alpha \sum_{m \in M} \|F_{m,h}(s) - \hat{F}_{m,h}(s)R_{m,h}\|_2 \|W_{m,h}(a)\|_2$$

$$\leq \alpha\sqrt{\frac{S}{d\mu}} \sum_{m \in M} \|F_{m,h}(s) - \hat{F}_{m,h}(s)R_{m,h}\|_2$$

$$\leq \sqrt{\frac{dHMA}{T}}$$

from the Cauchy-Schwarz inequality and our singular vector perturbation bound. From the same logic,

$$\left| \sum_{s' \in S} (P_h(s'|s,a) - F_h(s)^\top U_h(s',a)) \right| = |\tilde{F}_h(s)U'_h(s',a) - \hat{\tilde{F}}_h(s)R_h U'_h(s',a)^\top|$$

$$= |(\tilde{F}_h(s) - \hat{\tilde{F}}_h(s)R_h) \sum_{s' \in \mathcal{S}} U'_h(s',a)^\top|$$

$$\leq \alpha| \sum_{m \in M} (F_{m,h}(s) - \hat{F}_{m,h}(s)R_{m,h}) \sum_{s' \in \mathcal{S}} U_{m,h}(s',a)|$$

$$\leq \alpha \sum_{m \in M} \|F_{m,h}(s) - \hat{F}_{m,h}(s)R_{m,h}\|_2 \| \sum_{s' \in \mathcal{S}} U_{m,h}(s',a)^\top\|_2$$

$$\leq \sqrt{\frac{dHMA}{T}}.$$

Therefore, $\hat{\tilde{F}}_h(s)$ satisfies Assumption 7 with $\xi = \sqrt{\frac{dMHA}{T}}$. Then it follows that running LSVI-UCB-TR using $\hat{\tilde{F}}_h(s)$ with $\delta' = \delta/2$ during the target phase of the algorithm admits a regret bound of

$$Regret(T) \leq C"(\sqrt{d^3 M^3 H^3 AT} + dMH\sqrt{\frac{dMHA}{T}}) \in \tilde{O}(\sqrt{d^3 M^3 H^3 AT})$$

with probability at least $1 - \delta$ from a final union bound. Furthermore, the sample complexity in the source phase is

$$\tilde{O}\left(\frac{d^4 \mu^5 \kappa^4 (S+A) M^2 H^4 \alpha^2 T}{A}\right),$$

which proves our result. $\qquad\square$

## F.2 Simulator to Simulator Transfer RL

In this section, we consider the same transfer reinforcement learning setting with the low Tucker rank assumptions except the learner is given access to the generative model in both the source and target problems. In this setting, we isolate the statistical difficulty of the transfer RL problem by removing the challenge of exploration. Thus, instead of regret, the goal in this setting is to learn an $\epsilon$-optimal $Q$ function in the target phase using as few samples in the source and target phases as possible. The main benefit of having access to a generative model in the target phase is that the learner can choose the best states to observe covariates from when performing the regression step at each time step. In this setting, our algorithm still uses LR-EVI in the source phase (Algorithm 3 and adapts the algorithm from [22] for the finite horizon setting in the target phase. Learning $Q^*$ over the support of an approximately optimal design $\rho$ on our latent factors $\hat{\tilde{F}}$ allows us to construct a good estimate of $Q^*$ over all state-action pairs. The optimal design problem that we solve in this algorithm is

$$G(\rho) = \sum_{s \in S} \rho(s)\hat{\tilde{F}}(s)\hat{\tilde{F}}(s)^\top, \quad g(\rho) = \max_{s \in S} \|\hat{\tilde{F}}\|_{G(\rho)^{-1}}$$

where $\rho$ is a probability distribution over $S$. From Theorems 4.3 and 4.4 [22], one can compute a $\rho$ such that $g(\rho) \leq 2d$, with the core set of states, or anchor states, $\mathcal{S}^\# = \{s \in S | \rho(s) > 0\}$ having size at most $4d \log(\log(d)) + 16$ in a polynomial number of computations. This ensures that both the error amplification from using the low-rank structure and the number of states we need to observe $Q^*$ is not too large. Our algorithm is tailored to the $(S, d, A)$ Tucker rank setting but can easily be modified for the other Tucker rank settings.

---

**Algorithm 5** Target Phase

---

**Input:** $\{N_h^t\}_{h \in [H]}, \hat{\tilde{F}}$
1: **for** $h = H, \ldots, 1$ **do**
2:      Compute $\mathcal{S}_h^\#, \rho_h$ according to the above optimal design problem with $\hat{\tilde{F}}_h$.
3:      **for** $(s, a) \in \mathcal{S}_h^\# \times \mathcal{A}$ **do**
4:          Collect $N_h^t$ samples of the reward function and one-step transition to estimate $\hat{Q}_h(s, a)$
     with

$$\hat{Q}_h(s, a) = \hat{r}_h(s, a) + \mathbb{E}_{s' \sim \hat{P}_h(\cdot | s, a)}[\hat{V}_{h+1}(s')],$$

     where $\hat{r}_h(s, a)$ denotes the empirical average of the $N_h^t$ samples of the reward function, and $\hat{P}_h(\cdot | s, a)$ denotes the empirical distribution over the $N_h^t$ samples.
5:      Use the linear structure to estimate $\bar{Q}$ over all state-action pairs with

$$\hat{\theta}_{a,h} = G(\rho)^{-1} \sum_{s \in \mathcal{S}_h^\#} \rho(s)\hat{\tilde{F}}_h(s)\hat{Q}_h(s, a), \quad \bar{Q}_h(s, a) = \hat{\tilde{F}}_h(s)\theta_{a,h}.$$

6:      Compute the value function and policy using the $Q$ function,

$$\hat{V}_h(s) = \max_{a \in \mathcal{A}} \bar{Q}_h(s, a), \quad \hat{\pi}_h(s) = \arg\max_{a \in \mathcal{A}} \bar{Q}_h(s, a).$$

---

In the source phase, our algorithm learns estimates of $Q^*_{m,h}$ and then computes the singular value decomposition to construct approximate feature mappings using the singular subspaces with respect to the state. In step $(a)$ of the target phase, one first approximately solves the optimal design problem to compute the anchor states. After obtaining estimates of the $Q$ function on the anchor states, one uses the linear structure in step $(c)$ to estimate the full $Q$ function over all states with the least squares solution. The above algorithm admits the following sample complexities for learning an $\epsilon$-optimal $Q$ function in the target phase.

**Theorem 11.** *Suppose Assumptions 5, 6, and 3 hold, and set $\delta \in (0,1)$. Furthermore, assume that for any $\epsilon > 0$, $Q_{m,h} = r_{m,h} + P_{m,h}V_{m,h+1}$ has rank $d$ and is $\mu$-incoherent with condition number $\kappa$ for all $\epsilon$-optimal value functions $V_{h+1}$. Then, for $\epsilon \le \alpha d^2$, using at most*

$$\tilde{O}\left(\frac{d^7 \mu^3 \kappa^4 (S+A) H^9 \alpha^2 M^3}{\epsilon^2}\right)$$

*samples in the source problems, Algorithm 5 learns an $\epsilon$-optimal policy in the target MDP using at most*

$$\tilde{O}\left(\frac{d^2 M^2 H^5 A}{\epsilon^2}\right)$$

*samples with probability at least $1 - \delta$.*

Theorem 11 states that with enough samples in the source phase, one can remove the dependence on the size of the state space in the target phase; in tabular finite-horizon MDPs with $(S, d, A)$ Tucker rank transition kernels, one needs to observe at least $\tilde{\Omega}(d(S+A)H^3/\epsilon^2)$ samples from a generative model to learn an $\epsilon$-optimal policy. Furthermore, the source sample complexity's dependence on $S, A,$ and $H$ are reasonable while the dependence on $\alpha$ is optimal due to our lower bound. We can easily construct similar algorithms that admit the corresponding theoretical guarantees in the other Tucker rank settings. We now prove Theorem 11.

*Proof.* In the source phase, using

$$\tilde{O}\left(\frac{d^6 \mu^3 \kappa^4 (S+A) H^9 \alpha^2 M^4}{\epsilon^2}\right)$$

samples guarantees that we learn $\epsilon/(128 H^2 \alpha M^{3/2} \kappa d^2)$-optimal $\hat{Q}_{m,h} = \hat{F}_{m,h} \hat{\Sigma}_{m,h} \hat{G}^\top_{m,h}$ functions for each source MDP with probability at least $1 - \delta/2$. Since $\alpha d^2 \ge \epsilon$ by assumption, we have $\bar{\gamma} < 1/2$ in the singular vector perturbation bound, Corollary 3. Since $\|Q^*_{m,h}\|_\infty \ge C$, it follows from Corollary 3 that there exists rotation matrices $R_{m,h}$ such that

$$\|F_{m,h} - \hat{F}_{m,h} R\|_{2,\infty} \le \frac{\epsilon \sqrt{\mu}}{8 H^2 \alpha M^{3/2} \sqrt{S}}$$

holds for all $m \in [M], h \in [H]$ with probability at least $1 - \delta/2$. Let $\tilde{F}$ be the $S \times dM$ matrix from joining all the source state latent factor matrices, and $\hat{\tilde{F}}_h$ be the $S \times dM$ matrix from joining all $\bar{F}_{m,h}$. Let $R_h$ be the $d \times dM$ matrix that joins $R_{m,h}$ for all $m \in [M]$. Then, from Assumption 6 and Definition 4, it follows that for all $(s', s, a) \in \mathcal{S} \times \mathcal{S} \times \mathcal{A}$, that

$$
\begin{aligned}
r_h(s, a) &= \sum_{i \in [d]} F_h(s, i) W_h(a, i) \\
&= \sum_{i \in [d]} \left( \sum_{j \in [d], m \in [M]} B(i, j, m) F_{h,m}(s, j) \right) W_h(a, i) \\
&= \sum_{i,j \in [d], m \in [M]} B(i, j, m) F_{m,h}(s, j) W_h(a, i) \\
&= \tilde{F}_h(s) W'_h(a)^\top
\end{aligned}
$$

and $P(s' | \cdot, \cdot) = \tilde{F}_h U'_h(s')^\top$ (with the same argument) for some $A \times dM$ matrices $W'_h, U'_h(s')$. Note that $W'_h$ and $U'_h(s')$ are matrices that join $W_{m,h}$ and $U_{m,h}(s')$, respectively, with entries that at most

$\alpha$ times larger than the original entry. Therefore, it follows for all $(s', s, a, h) \in \mathcal{S} \times \mathcal{S} \times \mathcal{A} \times [H]$,

$$
\begin{aligned}
|r_h(s,a) - \hat{\tilde{F}}_h(s)R_h W_h'(a)^\top| &= |\tilde{F}_h(s)W_h'(a)^\top - \hat{\tilde{F}}_h(s)R_h W_h'(a)^\top| \\
&= |(\tilde{F}_h(s) - \hat{\tilde{F}}_h(s)R_h)W_h'(a)| \\
&\leq \alpha| \sum_{m \in M} (F_{m,h}(s) - \hat{F}_{m,h}(s)R_{m,h})W_{m,h}(a)| \\
&\leq \alpha \sum_{m \in M} \|F_{m,h}(s) - \hat{F}_{m,h}(s)R_{m,h}\|_2 \|W_{m,h}(a)\|_2 \\
&\leq \alpha \sqrt{\frac{S}{d\mu}} \sum_{m \in M} \|F_{m,h}(s) - \hat{F}_{m,h}(s)R_{m,h}\|_2 \\
&\leq \frac{\epsilon}{8\sqrt{dM}H^2}
\end{aligned}
$$

from the Cauchy-Schwarz inequality and our singular vector perturbation bound. From the same logic,

$$
\begin{aligned}
|PV(s,a) - [\hat{\tilde{F}}_h R_h U_h' V](s,a)| &= | \sum_{s' \in S} P_h(s'|s,a)V(s') - \hat{\tilde{F}}_h(s)R_h U_h'(s',a)^\top V(s')| \\
&= | \sum_{s' \in S} \tilde{F}_h(s)U_h'(s',a)^\top V(s') - \hat{\tilde{F}}_h(s)R_h U_h'(s',a)^\top V(s')| \\
&\leq H|(\tilde{F}_h(s) - \hat{\tilde{F}}_h(s)R) \sum_{s' \in S} U_h'(s',a)| \\
&\leq H\alpha \sum_{m \in M} \|F_{m,h}(s) - \hat{F}_{m,h}(s)R\|_2 \| \sum_{s' \in S} U_{m,h}(s',a)\|_2 \\
&\leq \frac{H\alpha\sqrt{S}}{\sqrt{d\mu}} \sum_{m \in M} \|F_{m,h}(s) - \hat{F}_{m,h}(s)R\|_2 \\
&\leq \frac{\epsilon}{8\sqrt{dM}H}.
\end{aligned}
$$

From the above result, it follows that for any value function $V$, the misspecification error of using the approximate feature mapping of the corresponding $Q$ function $Q' = r + PV$ satisfies

$$
\begin{aligned}
|Q_h'(s,a) - Q_{h,d}'(s,a)| &\leq |r_h(s,a) - \hat{F}(s)RW(a)^\top| + |PV(s,a) - [\hat{\tilde{F}}_h R_h U_h' V](s,a)| \\
&\leq \frac{\epsilon}{8\sqrt{dM}H^2} + \frac{\epsilon}{8\sqrt{dM}H} \leq \frac{\epsilon}{4H\sqrt{dM}}.
\end{aligned}
$$

To prove our main theorem, we first present the following helper lemma and defer the proof to later in this section.

**Lemma 2.** *Suppose the setting of Theorem 11 holds and that the misspecification error of the bellman update for any value function using an approximate feature mapping satisfies*

$$
|Q_h'(s,a) - Q_{h,d}'(s,a)| \leq \frac{\epsilon}{4H\sqrt{dM}}.
$$

*Then, the learned Q function from our algorithm at time step $h$ satisfies*

$$
\|\bar{Q}_h - Q_h^*\|_\infty, \|\bar{Q}_h - Q_h^{\hat{\pi}}\|_\infty \leq \frac{\epsilon(H - h + 1)}{H}
$$

*for $h \in [H]$ for $N_h = H^4 dMC^2 \log(SAH/\delta)/\epsilon^2$ with probability at least $1 - (H - h + 1)\delta/(2H)$.*

From a union bound and applying Lemma 2, it follows that Linear $Q$-Learning learns an $\epsilon$-optimal policy with probability at least $1 - \delta$. Therefore, the sample complexity of the algorithm is

$$
\sum_{h \in H} N_h |\mathcal{S}_h^\#| A \leq \tilde{O}\left(\frac{d^2 M^2 H^5 A}{\epsilon^2}\right).
$$

It follows from the triangle inequality that the learned policy is $2\epsilon$-optimal by replacing $2\epsilon$ with $\epsilon'$.

$\square$

We now prove Lemma 2.

*Proof.* We proceed via induction. At time step $H$ (the base case), it follows from Hoeffding's inequality for $N_H = H^4 dMC^2 \log(SAH/\delta)/\epsilon^2$ for some constant $C > 0$,

$$|\hat{Q}_H(s,a) - Q_H^*(s,a)| \leq \frac{\epsilon}{4\sqrt{dM}H}$$

for $(s,a) \in \mathcal{S}_h^{\#} \times \mathcal{A}$. From Proposition 4.5 from [22] and our assumption on the misspecification error of the approximate feature mapping, it follows that

$$|\bar{Q}_H(s,a) - Q_H^*(s,a)| \leq \frac{\epsilon}{H}.$$

Since $Q_H^{\hat{\pi}} = Q_H^*$, it follows that the base case holds.

Next, assume that

$$\|\bar{Q}_{h+1} - Q_{h+1}^*\|_\infty, \|\bar{Q}_{h+1} - Q_{h+1}^{\hat{\pi}}\|_\infty \leq \frac{\epsilon(H-h)}{H}$$

holds with probability at least $1 - (H-h)\delta/(2H)$. We define $Q_h'$ as $Q_h' = r_h + P_h\hat{V}_{h+1}$. Let $Q_{h,d}' = F_h(W_h + \sum_{s' \in S} \hat{V}_{h+1}(s')U_h(s'))$. Then, it follows that from Hoeffding's inequality for $N_h = H^4 C^2 dM \log(SAH/\delta)/\epsilon^2$ for some constant $C > 0$, for all $(s,a) \in \mathcal{S}_h^{\#} \times \mathcal{A}$, in step (b) of our algorithm,

$$|\hat{Q}_h(s,a) - Q_h'(s,a)| \leq \frac{\epsilon}{4H\sqrt{dM}}.$$

From our assumption on the misspecification error of the approximate feature mapping, it follows that $\|Q_h' - Q_{h,d}'\|_\infty \leq \frac{\epsilon}{4H\sqrt{dM}}$. Then, from Proposition 4.5 [22], we have

$$|\bar{Q}_h(s,a) - Q_{h,d}'(s,a)| \leq \frac{\epsilon}{2H} + \frac{\epsilon}{2H} = \frac{\epsilon}{H}$$

for all $(s,a) \in S \times \mathcal{A}$. Since

$$|Q_h^*(s,a) - Q_h'(s,a)| \leq |\sum_{s' \in S} (V_{h+1}^*(s') - \hat{V}_{h+1}(s'))P_h(s'|s,a)| \leq \frac{\epsilon(H-h)}{H}$$

and

$$|Q_h^{\hat{\pi}}(s,a) - Q_h'(s,a)| \leq |\sum_{s' \in S} (V_{h+1}^*(s') - \hat{V}_{h+1}(s'))P_h(s'|s,a)| \leq \frac{\epsilon(H-h)}{H},$$

holds from the inductive hypothesis for all $(s,a) \in S \times \mathcal{A}$, it follows that

$$|\bar{Q}_h(s,a) - Q_h^*(s,a)| \leq |\bar{Q}_h(s,a) - Q_h'(s,a)| + |Q_h'(s,a) - Q_h^*(s,a)|$$
$$\leq \frac{\epsilon}{H} + \frac{\epsilon(H-h)}{H} = \frac{\epsilon(H-h+1)}{H},$$

and similarly,

$$|Q_h^{\hat{\pi}}(s,a) - \bar{Q}_h(s,a)| \leq \frac{\epsilon(H-h+1)}{H}.$$

Thus, the inductive hypothesis holds as a union bound asserts that the above result holds with probability at least $1 - (H-h+1)\delta/(2H)$.

$\square$

## G  $(d, S, A)$ **Tucker Rank Setting**

In the low-rank MDP transfer RL setting, [4] provide an algorithm that admits a source sample complexity of $\tilde{O}(A\alpha^3 MT \log(\Phi))$ and target regret of $\tilde{O}(\bar{\alpha}H^2 d^{3/2}\sqrt{T})$ with high probability in the transfer RL setting with low-rank MDPs. We first present the transfer learning assumptions in [4] needed to prove a lower bound in their setting. Low rank MDPs assume the following stucutre with shared latent representation $\phi$,

$$P_{m,h}(s'|s,a) = \mu_{m,h}(s')^\top \phi_h(s,a),$$
$$P_h(s'|s,a) = \mu_h(s')^\top \phi_h(s,a),$$

where $\|\phi_h(s,a)\|_2 \leq 1$ and $\|\int \mu(s)g(s)dx\|_2 \leq \sqrt{d}$ for any function $g : S \to [0,1]$ [3] for all $m \in [M], h \in [H]$. To allow transfer learning to occur, [4] assume the following:

**Assumption 8** (Assumption 2.2 [4])**.** *For any $h \in [H]$ and $s' \in S$, there exists $\alpha_{m,h}(s') \in \mathbb{R}$ such that $\mu_h(s') = \sum_{m\in[M]} \alpha_{m,h}(s')\mu_{m,h}(s')$,*

and define the task-relatedness coefficient as $\alpha = \max_{m\in[M],h\in[H],s\in S} \alpha_{m,h}(s)$. They leave determining the optimal dependence on $\alpha$ as an interesting open question. Similarly to our lower bound in Theorem 1, we prove that one must incur a source sample complexity of $\Omega(\alpha^2)$ to benefit from using transfer learning. Note that in this version of transfer RL, there is no reward function in the source tasks.

**Theorem 12.** *There exist two transfer RL instances such that (i) they satisfy Assumptions 9 and 8, (ii) they cannot be distinguished without observing $\Omega(\alpha^2)$ samples in the source phase, and (3) they have target state-action latent features that are orthogonal to each other.*

*Proof.* We consider two low-rank transfer RL problems, in which the source MDPs have similar transition kernels but differing $\phi$. All source and target MDPs $(\mathcal{S}, \mathcal{A}, P, H, r)$ share the same state space, action space, and horizon $H = 2$ with $M = 2$. The learner is given access to a generative model in the source problems. To introduce the MDPs (without loss of generality assume that $|S|, |A|$ are even), we first define the feature representations: the feature mapping $\phi_i$ for transfer RL problem $i$ is

$$\phi_1(s_1,a_1) = \phi_1(s_2,a_2) = [1,0]^\top, \phi_1(s_1,a_2) = \phi_2(s_2,a_1) = [0,1]^\top,$$

$$\phi_2(s_1,a_1) = \phi_2(s_2,a_2) = [0,1]^\top, \phi_2(s_1,a_2) = \phi_2(s_2,a_1) = [1,0]^\top$$

where $s_1$ refers to states $1, \ldots, |S|/2$, $s_2$ refers to states $|S|/2 + 1, \ldots, |S|$, and similarly for actions. We will present the transition kernels matrices with $s_1, s_2, a_1, a_2$ as the blocks containing $|S|/2$ or $|A|/2$ entries. Clearly, the two feature representations are orthogonal to each other. Next, the

transition kernels $P_{m,1}^i$ for transfer RL problem $i$ and source MDP $m$ are

$$P_{1,1}^1(s_1|\cdot,\cdot) = [1/2, 1/2 - 1/\alpha]\phi_1(\cdot,\cdot)$$
$$= \begin{bmatrix} 1/2 & 1/2 - 1/\alpha \\ 1/2 - 1/\alpha & 1/2 \end{bmatrix}$$
$$P_{1,1}^1(s_2|\cdot,\cdot) = [1/2, 1/2 + 1/\alpha]\phi_1(\cdot,\cdot)$$
$$= \begin{bmatrix} 1/2 & 1/2 + 1/\alpha \\ 1/2 + 1/\alpha & 1/2 \end{bmatrix}$$
$$P_{1,1}^2(s_1|\cdot,\cdot) = [1/2, 1/2 - 1/\alpha]\phi_2(\cdot,\cdot)$$
$$= \begin{bmatrix} 1/2 - 1/\alpha & 1/2 \\ 1/2 & 1/2 - 1/\alpha \end{bmatrix}$$
$$P_{1,1}^2(s_2|\cdot,\cdot) = [1/2, 1/2 + 1/\alpha]\phi_2(\cdot,\cdot)$$
$$= \begin{bmatrix} 1/2 + 1/\alpha & 1/2 \\ 1/2 & 1/2 + 1/\alpha \end{bmatrix}$$
$$P_{2,1}^1(s_1|\cdot,\cdot) = P_{2,1}^1(s_2|\cdot,\cdot) = P_{2,1}^2(s_2|\cdot,\cdot) = P_{2,1}^2(s_2|\cdot,\cdot)$$
$$= [1/2, 1/2]\phi_1(\cdot,\cdot) = [1/2, 1/2]\phi_2(\cdot,\cdot)$$
$$= \begin{bmatrix} 1/2 & 1/2 \\ 1/2 & 1/2 \end{bmatrix}$$
$$P_T^i(\cdot|\cdot,\cdot) = \mu_T(\cdot)\phi_i(\cdot,\cdot),$$

where $\mu_T(\cdot) = [0,1]$ in both transfer RL problems. It follows that

$$\mu_T(s_1) = \alpha(\mu_1^2(s_1) - \mu_1^1(s_1)) = \alpha(\mu_2^2(s_1) - \mu_2^1(s_1))$$

and similar results hold for $s_2$, so $\alpha$ satisfies the definition of the task-relatedness coefficient. It follows that in both settings, $\alpha_{\max} = \alpha$. The target phase reward functions are defined as

$$r_1^1(\cdot,\cdot) = r_2^1(\cdot,\cdot) = [1,0]\phi_1(\cdot,\cdot)$$
$$r_1^2(\cdot,\cdot) = r_2^2(\cdot,\cdot) = [1,0]\phi_2(\cdot,\cdot),$$

and the initial state distribution is uniform across all states. By construction, in transfer RL problem 1, the only state action pairs that receive reward are $(s_1, a_1), (s_2, a_2)$. In contrast, in transfer RL problem 2, the only state action pairs that receive reward are $(s_1, a_1), (s_2, a_1)$.

The learner is given either transfer RL problem one or two with equal probability with a generative model in the source problem. When interacting with the source MDPs, the learner specifies a state-action pair $(s, a)$ and source MDP $m$ and observes a transition from $P_{1,m}^i$. Furthermore, the learner is given the knowledge that the state latent features lie in the function class $\Phi = \{\phi_1, \phi_2\}$ and must use the knowledge obtained from interacting with the source MDP to choose one to use in the target MDP, which is no harder than receiving no information about the function class.

To distinguish between the two transfer RL problems, one needs to differentiate between $P_{1,1}^1$ and $P_{1,1}^2$. By construction, the magnitude of the largest entrywise difference between $P_{1,1}^1$ and $P_{1,1}^2$ is lower bounded by $\Omega(1/\alpha^2)$. If the learner observes $Z$ transitions in the source MDPs, where $Z \leq O(\alpha^2)$ for some constant $C > 0$, then the probability of correctly identifying $\phi$ is upper bounded by $0.76$ [5]. Thus, if the learner does not observe $\Omega(\alpha^2)$ samples in the source phase, then the learner returns the incorrect feature mapping that is orthogonal to the true feature mapping with probability at least $0.24$. $\qquad\square$

Theorem 12 states that one must incur dependence on $\alpha$ in the source sample complexity to benefit from transfer learning as using a feature mapping that is orthogonal to the true one results in regret linear in $T$.

As the low rank MDP setting is analogous to our $(d, S, A)$ Tucker rank setting, we next present our assumptions in this setting,

**Assumption 9.** *In each of the $M$ source MDPs, the reward functions have rank $d$, and the transition kernels have Tucker rank $(d, S, A)$. Let the target MDP's reward functions have rank $d'$ and transition kernels have Tucker rank $(d', S, A)$ where $d' \leq dM$. Thus, there exists $S \times d$ tensors $U_{m,h}$, $S \times d'$ tensors $U_h$, $SA \times d$ matrices with orthonormal columns $\phi_{m,h}$, $SA \times d'$ matrices with orthonormal columns $\phi_h$, and $d$-dimensional vectors $W_{m,h}$, and $d'$-dimensional vectors $W_h$ such that*

$$P_{m,h}(s'|s,a) = \sum_{i \in [d]} \phi_{m,h}(s,a,i) U_{m,h}(s',i),$$
$$r_{m,h}(s,a) = \sum_{i \in [d]} \phi_{m,h}(s,a,i) W_{m,h}(i)$$

*and*

$$P_h(s'|s,a) = \sum_{i \in [d']} \phi_h(s,a,i) U_h(s',i),$$
$$r_h(s,a) = \sum_{i \in [d']} \phi_h(s,a,i) W_h(i)$$

*where $\|\sum_{s' \in \mathcal{S}} g(s') U_{m,h}(s')\|_2, \|\sum_{s' \in \mathcal{S}} g(s') U_h(s')\|_2 \leq \sqrt{d}, \|W_h\|_2$, and $\|W_{m,h}(s)\|_2 \leq \sqrt{S/\mu}$ for all $s \in \mathcal{S}$, $a \in \mathcal{A}$, $h \in [H]$, and $m \in [M]$ for any function $g : \mathcal{S} \to [0,1]$.*

While [4] assume that the target latent representation is a linear combination of the source latent representation, we allow for a more general setting to enable transfer learning. Specifically, we assume that the space spanned by the target latent representations is a subset of the space spanned by the source latent representations.

**Assumption 10.** *Suppose Assumption 9 holds. For latent factors $\phi_h$ from the target MDP and $\phi_{m,h}$ for the source MDPs, let $\phi_h = \{\phi_h, \phi_{h,m}\}_{m \in [M]}$. Then, there exists a non-empty set $\mathcal{B}(\phi_h)$ such that the elements in $\mathcal{B}(\phi_h)$ are $B_h \in \mathbb{R}^{d',d,M}$ such that*

$$\phi_h(s,i) = \sum_{j \in [d], m \in [M]} B_h(i,j,m) \phi_{m,h}(s,j)$$

*for all $i \in [d'], s \in S, h \in [H]$.*

As in the previous Tucker rank settings, we present the transfer-ability coefficient to quantify the difficulty transfer learning.

**Definition 5.** *Given a transfer RL problem that satisfies Assumptions 9 and 10. Let $\Phi$ be the set of all $\mathbb{R}^{SA \times d}$ latent factor matrices with orthonormal columns that span the column space of $P_{m,h}$ for all $m \in [M], h \in [H]$. Then, we define $\alpha$ as*

$$\alpha := \min_{\phi \in \Phi} \min_{B \in \mathcal{B}(\phi)} \max_{i \in [d'], j \in [d], m \in [M]} |B(i,j,m)|.$$

[4] provide an alternate definition of $\alpha$, which also measures the difficulty of transfer RL, and leave determining the optimal dependence on $\alpha$ as an interesting open question. Similarly to our lower bound in Theorem 1, we prove that one must incur a source sample complexity of $\Omega(\alpha^2)$ to benefit from using transfer learning. Note that in this Tucker rank setting, there is no reward function in the source tasks.

**Theorem 13.** *There exist two transfer RL instances such that (i) they satisfy Assumptions 9 and 10, (ii) they cannot be distinguished without observing $\Omega(\alpha^2)$ samples in the source phase, and (3) they have target state-action latent features that are orthogonal to each other.*

*Proof.* Consider the construction where all source and target MDPs $(\mathcal{S}, \mathcal{A}, P, 1, r)$ share the same state space, action space, and horizon with $H = 1$. As the horizon is one, we use the construction used to prove Theorem 7. We consider two transfer RL problems $i, i'$, in which the source MDPs have similar $Q_1^*$, but with orthogonal target $\phi_T$. For ease of notation, we let $a_1$ refer to any action in $[n]$, and $a_2$ refer to any action in $\{n+1, \ldots 2n\}$, respectively. The initial state distribution in the target MDP is uniform over $\mathcal{S}$. We now present two transfer RL problems with similar $Q$ functions that satisfy Assumptions 5 and 6 but have orthogonal target state latent factors. For ease of notation, the superscript $i$ of $Q_{m,h}^{*,i}, Q_h^{*,i}$ denotes transfer RL problem. Then, the optimal $Q$ functions (with

columns representing $a_1$ and $a_2$ and the row being any state) for transfer RL problem one are

$$Q_{1,1}^{*,1} = W_{1,h}^1 \phi_{1,1}^1 \begin{bmatrix} \sqrt{n} \\ \sqrt{n} \end{bmatrix} \begin{bmatrix} \sqrt{1/(2n)} & \sqrt{1/(2n)} \end{bmatrix} = \begin{bmatrix} 1/\sqrt{2} & 1/\sqrt{2} \end{bmatrix},$$

$$Q_{2,1}^{*,1} = W_{2,h}^1 \phi_{1,1}^1 \begin{bmatrix} \sqrt{n} \\ \sqrt{n} \end{bmatrix} \begin{bmatrix} \sqrt{1/(2n)} & \sqrt{1/(2n)} \end{bmatrix} = \begin{bmatrix} 1/\sqrt{2} & 1/\sqrt{2} \end{bmatrix},$$

$$Q_1^{*,1} = W_h \phi_1^1 = \begin{bmatrix} \sqrt{n} \\ \sqrt{n} \end{bmatrix} \begin{bmatrix} \sqrt{1/(2n)} & \sqrt{1/(2n)} \end{bmatrix} = \begin{bmatrix} 1/\sqrt{2} & 1/\sqrt{2} \end{bmatrix}$$

and for transfer RL problem two are

$$Q_{1,1}^{*,2} = W_{1,h}^2 \phi_{1,1}^2 \begin{bmatrix} \sqrt{n} \\ \sqrt{n} \end{bmatrix} \begin{bmatrix} \sqrt{1/(2n)} & \sqrt{1/(2n)} \end{bmatrix} = \begin{bmatrix} 1/\sqrt{2} & 1/\sqrt{2} \end{bmatrix},$$

$$Q_{2,1}^{*,2} = W_{2,h}^2 \phi_{2,1}^2 \begin{bmatrix} \sqrt{n} \\ \sqrt{n} \end{bmatrix} G' = \begin{bmatrix} (1+1/\alpha)/(c\sqrt{2}) & (1-1/\alpha)/(c\sqrt{2}) \end{bmatrix},$$

$$Q_2^{*,1} = W_h \phi_1^2 = \begin{bmatrix} \sqrt{n} \\ \sqrt{n} \end{bmatrix} \begin{bmatrix} \sqrt{1/(2n)} & -\sqrt{1/(2n)} \end{bmatrix} = \begin{bmatrix} 1/\sqrt{2} & -1/\sqrt{2} \end{bmatrix}$$

and $c = \sqrt{1 + 1/\alpha^2}$. Note that $c \in (1, 2dM)$ because $\alpha \geq 1/(dM)$ from Lemma 1 for $\alpha > 0$. The incoherence $\mu$, rank $d$, and condition number $\kappa$ of the above $Q$ functions are $O(1)$. Furthermore, the above construction satisfies Assumption 9 with Tucker rank $(1, S, A)$ and Assumption 10 because the source and target MDPs share the same feature representation $\phi_1^1 = \phi_{1,1}^1$ in the first transfer RL problem while in the second transfer RL problem,

$$\phi_1^2 = -\alpha \phi_{2,1}^2 + \alpha \phi_{2,1}^2.$$

Note that $\phi_1^1$ and $\phi_2^2$ are orthogonal to each other by construction.

The learner is given either transfer RL problem one or two with equal probability with a generative model in the source problem. When interacting with the source MDPs, the learner specifies a state-action pair $(s, a)$ and source MDP $m$ and observes a realization of a shifted and scaled Bernoulli random variable $X$. The distribution of $X$ is that $X = 1$ with probability $(Q_{m,1}^{*,i}(s, a) + 1)/2$ and $X = -1$ with probability $(1 - Q_{m,1}^{*,i}(s, a))/2$. Furthermore, the learner is given the knowledge that the state latent features lie in the function class $\Phi = \{\phi_1, \phi_2\}$ and must use the knowledge obtained from interacting with the source MDP to choose one to use in the target MDP, which is no harder than receiving no information about the function class.

To distinguish between the two transfer RL problems, one needs to differentiate between $Q_{2,1}^{*,1}$ and $Q_{2,1}^{*,2}$. By construction, the magnitude of the largest entrywise difference between $Q_{2,1}^{*,1}$ and $Q_{2,1}^{*,2}$ is lower bounded by $\Omega(1/\alpha)$.

If the learner observes $Z$ samples in the source MDPs, where $Z \leq O(\alpha^2)$ for some constant $C > 0$, then the probability of correctly identifying $G$ is upper bounded by 0.76 (Lemma 5.1 with $\delta = 0.24$ [5]). Thus, if the learner does not observe $\Omega(\alpha^2)$ samples in the source phase, then the learner returns the incorrect feature mapping that is orthogonal to the true feature mapping with probability at least 0.24. □

Theorem 13 states that one must incur dependence on $\alpha$ in the source sample complexity to benefit from transfer learning. With this lower bound, we've shown in all three Tucker rank settings that $\alpha$ determines the effectiveness of transfer learning under our transfer learning assumptions. We now present our algorithm that admits an efficient source sample complexity and target regret bound.

In contrast to our approaches in the other Tucker rank settings, one learns the latent representation in the source MDPs through the transition kernels. One potential issue arises when all entries of $P(\cdot|\cdot, a)$ are small; for example, each entry of the uniform transition kernel is $1/S$, which is not lower bounded by a constant. Thus, we let $D = \min_{m \in [M], h \in [H], a \in \mathcal{A}} \|P_{m,h}(\cdot|\cdot, a)\|_\infty$, and the dependence on $D$ is unsurprising as one needs to ensure the noise is sufficiently small to learn the latent representations from $P_{m,h}$. We now present and prove the main result in this Tucker rank setting.

---

**Algorithm 6** Source Phase

---

**Input:** $N$
1: **for** $(s, a) \in \mathcal{S} \times \mathcal{A}, h \in [H], m \in [M]$ **do**
2:     Collect $N$ transitions from source MDP $m$ at time step $h$ at state-action pair $(s, a)$.
3:     Use the samples to estimate the transition kernel with

$$\hat{P}_{m,h}(s'|s, a) = \frac{1}{N} \sum_{i \in [N]} \mathbf{1}_{X_{i,s,a}=s'},$$

    where $X_{i,s,a} \sim P_{m,h}(\cdot|s, a)$.
4: Compute the feature mapping $\hat{\phi} = \{\hat{\phi}_h\}_{h \in [H]}$ with

$$\hat{\phi}_h = \left\{ \frac{\sqrt{S}}{d\mu} \hat{F}_{m,h}(:, i)\hat{G}_{n,h}(:, j)|i, j \in [d], m, n \in [M] \right\}.$$

---

---

**Algorithm 7** Target Phase: LSVI-UCB

---

**Input:** $\lambda, \beta_{k,h}, \hat{\phi}$
1: Run LSVI-UCB with $\lambda, \beta_{k,h}$, and $\hat{\phi}$.

---

**Theorem 14.** *Suppose Assumptions 5, 6, and 3 hold in the $(d, S, A)$ Tucker Rank setting, and set $\delta \in (0, 1)$. Furthermore, assume that for any $\epsilon \in (0, \sqrt{H/T})$, $P_{m,h}(\cdot|\cdot, a)$ has rank $d$ and is $\mu$-incoherent with condition number $\kappa$ for all $a \in A$. Let $D = \min_{m \in [M], h \in [H], a \in \mathcal{A}} \|P_{m,h}(\cdot|\cdot, a)\|_\infty$ Then, for $T \geq H/\alpha^2$, using at most*

$$\tilde{O}\left(\frac{\alpha^2 d\mu^2 \kappa^2 M^2 TSA}{D^2}\right),$$

*samples in the source problems, our algorithm has regret at most*

$$\tilde{O}\left(\sqrt{(dMH)^3 T}\right)$$

*with probability at least $1 - \delta$ for $\lambda = 1$ and $\beta_{h,k}$, which is a function of $d, H, T, M$.*

*Proof.* Suppose the setting of Theorem 14 holds. Recall that we let $D = \min_{m \in [M], h \in [H], a \in \mathcal{A}} \|P_{m,h}(\cdot|\cdot, a)\|_\infty$. Then, observing $N$ transitions from each state-action pairs ensures that for each $(s', s, a) \in \mathcal{S} \times \mathcal{S} \times A$ and for each source MDP at each time step,

$$|P_{m,h}(s'|s, a) - \hat{P}_{m,h}(s'|s, a)| \leq \frac{\sqrt{H}}{\alpha\mu\kappa D\sqrt{dMT}}$$

with probability at least $1 - \delta/(2S^2 AMH)$ where $\hat{P}_{m,h}(s'|s, a) = \frac{1}{N} \sum_{i \in [N]} \mathbf{1}_{X_i(s,a)=s'}$ and $X_i \sim P(\cdot|s, a)$ for

$$N = \frac{\alpha^2 d\mu^2 \kappa^2 MT \log(4S^2 AMH/\delta)}{2HD^2}.$$

Let $\hat{P}_{m,h}(\cdot|\cdot, a) = \hat{F}_{m,h}^a \hat{\Sigma}_{m,h} \hat{G}_{m,h}\top$ be the singular value decomposition of our estimate where $\hat{F}$ corresponds to the state one transitions from. Since $T \geq \alpha^2/H$, it follows that $\bar{\gamma} \leq 1/2$. Thus, 3 states that there exists rotation matrices $R_{m,h,a}$ such that

$$\|(\hat{F}_{m,h}^a R_{m,h,a} - F_{m,h}^a)(s)\|_2 \leq \frac{d\sqrt{H\mu}}{\alpha\sqrt{MTS}},$$

for all $s \in \mathcal{S}$ where the the true transition kernel $P_{m,h}(\cdot|\cdot, a)$ has singular value decomposition $P_{m,h}(\cdot|\cdot, a) = F_{m,h}^a \Sigma_{m,h} G_{m,h}^\top$. We can re-express our terms in the more common Low rank MDP setting with $\phi_{m,h}(s, a) = F_{m,h}^a(s)$ and $\mu_{m,h}(s') = \Sigma_{m,h} G_{m,h}^\top$.

Let $\tilde{F}$ be the $SA \times dM$ matrix from joining all the source state latent factor matrices from $P(\cdot|\cdot, a)$ for each $a \in A$, and $\hat{\tilde{F}}_h$ be the $SA \times dM$ matrix from joining all $\bar{F}_{m,h}$. Let $R_h$ be the $d \times dM$ matrix that joins $R_{m,h}$ for all $m \in [M]$. Then, from Assumption 6 and Definition 4, it follows that for all $(s', s, a) \in \mathcal{S} \times \mathcal{S} \times A$, that

$$r_h(s, a) = \sum_{i \in [d]} F_h^a(s, i) W_h(i)$$

$$= \sum_{i \in [d]} \left( \sum_{j \in [d], m \in [M]} B(i, j, m) F_{h,m}^a(s, j) \right) W_h(i)$$

$$= \sum_{i, j \in [d], m \in [M]} B(i, j, m) F_{m,h}^a(s, j) W_h(i)$$

$$= \tilde{F}_h^a(s) W_h'^\top$$

and $P(s'|s, a) = \tilde{F}_h^a(s) U_h'(s')^\top$ (with the same argument) for some $dM$-dimension vector $W_h', U_h'(s')$. Note that $W_h'$ and $U_h'(s')$ are vectors that join $W_{m,h}$ and $U_{m,h}(s')$, respectively, with entries that at most $\alpha$ times larger than the original entry.

Therefore, it follows for all $(s', s, a, h) \in \mathcal{S} \times \mathcal{S} \times \mathcal{A} \times [H]$,

$$|r_h(s, a) - \hat{\tilde{F}}_h^a(s) R_h W_h'^\top| = |\tilde{F}_h^a(s) W_h'^\top - \hat{\tilde{F}}_h^a(s) R_h W_h'^\top|$$

$$= |(\tilde{F}_h^a(s) - \hat{\tilde{F}}_h^a(s) R_h) W_h'^\top|$$

$$\leq \alpha| \sum_{m \in M} (F_{m,h}^a(s) - \hat{F}_{m,h}^a(s) R_{m,h}) W_{m,h}|$$

$$\leq \alpha \sum_{m \in M} \|F_{m,h}^a(s) - \hat{F}_{m,h}^a(s) R_{m,h}\|_2 \|W_{m,h}\|_2$$

$$\leq \alpha \sqrt{\frac{S}{d\mu}} \sum_{m \in M} \|F_{m,h}^a(s) - \hat{F}_{m,h}^a(s) R_{m,h}\|_2$$

$$\leq \sqrt{\frac{dHM}{T}}$$

from the Cauchy-Schwarz inequality and our singular vector perturbation bound. From the same logic,

$$\left| \sum_{s' \in S} (P_h(s'|s, a) - \hat{\tilde{F}}_h^a(s) R_h U_h'(s')^\top) \right| \leq |\sum_{s' \in \mathcal{S}} P_h(s'|s, a) - \hat{\tilde{F}}_h^a(s) R_h U_h'(s')^\top|$$

$$= |\sum_{s' \in \mathcal{S}} \tilde{F}_h^a(s) U_h'(s') - \hat{\tilde{F}}_h^a(s) R_h U_h'(s')^\top|$$

$$\leq \alpha| \sum_{m \in M} (F_{m,h}^a(s) - \hat{F}_{m,h}^a(s) R_{m,h}) \sum_{s' \in \mathcal{S}} U_{m,h}(s')|$$

$$\leq \alpha \sum_{m \in M} \|F_{m,h}^a(s) - \hat{F}_{m,h}^a(s) R_{m,h}\|_2 \| \sum_{s' \in \mathcal{S}} U_{m,h}(s')^\top\|_2$$

$$\leq \sqrt{\frac{dHM}{T}}.$$

Therefore, $\hat{\tilde{F}}_h^a(s)$ satisfies Assumption B in [17] with $\xi = \sqrt{\frac{dMH}{T}}$ [5]. Then it follows that running LSVI-UCB using $\hat{\tilde{F}}_h^a(s)$ with $\delta' = \delta/2$ during the target phase of the algorithm admits a regret bound

---

[5]While the misspecified assumption in [17] uses the total variation distance, they refer to it as $|\sum_{s' \in S} P(s'|s, a) - \hat{P}(s'|s, a)|$ in the second to last inequality in the proof of Lemma C.1 and the second to last equation on page 23

of

$$Regret(T) \leq C''(\sqrt{d^3 M^3 H^3 T} + dMH\sqrt{\frac{dMH}{T}}) \in \tilde{O}(\sqrt{d^3 M^3 H^3 T})$$

with probability at least $1 - \delta$ from a final union bound. Furthermore, the sample complexity in the source phase is

$$\tilde{O}\left(\frac{\alpha^2 d\mu^2 \kappa^2 M^2 TSA}{D^2}\right),$$

which proves our result. $\qquad \square$

## H  $(d, d, d)$ Tucker Rank Setting

In this section, we present the assumptions, algorithm, and proof for Theorem 15. In the $(d, d, d)$ Tucker rank setting, we assume the following low-rank structure on the MDPs.

**Assumption 11.** *In each of the $M$ source MDPs, the reward functions have rank $d$, and the transition kernels have Tucker rank $(d, d, d)$. The target MDP's reward functions have rank $d'$ and transition kernels have Tucker rank $(d, d', d')$ where $d' \leq dM$. Thus, there exists $d \times d \times d$ tensors $U_{m,h}$, $d \times d' \times d'$ tensors $U_h$, $S \times d$ matrices $V_{m,h}$, $S \times d'$ matrices $V_h$, $S \times d$ matrices $F_{m,h}$, $S \times d'$ matrices $F_h$, $A \times d$ matrices $G_{m,h}$, and $A \times d'$ matrices $G_h$ all with orthonormal columns, and sequences of non-increasing singular values $\{\sigma_{m,h}(i) \geq \mathbb{R}_+\}_{i \in [d]}, \{\sigma_h(i) \geq \mathbb{R}_+\}_{i \in [d']}$ such that*

$$P_{m,h}(s'|s, a) = \sum_{i \in [d], j \in [d'], k \in [d]} U_{m,h}(i, j, k) V_{m,h}(s', i) F_{m,h}(s, j) G_{m,h}(a, k),$$
$$r_{m,h}(s, a) = \sum_{i \in [d]} \sigma_{m,h}(i) F_{m,h}(s, i) G_{m,h}(a, i)$$

*and*

$$P_h(s'|s, a) = \sum_{i \in [d], j \in [d'], k \in [d']} U_h(i, j, k) V_h(s', i) F_h(s, j) G_h(a, k),$$
$$r_h(s, a) = \sum_{i \in [d']} \sigma_h(i) F_h(s, i) G_h(a, i)$$

*where* $\|\sum_{i \in [d], s' \in \mathcal{S}} g(s') U_{m,h}(i, :, :) V_{m,h}(s', i)\|_F, \|\sum_{s' \in \mathcal{S}, i \in [d]} g(s') U_h(i, :, :) V_h(s', i)\|_F, \sigma_{m,h}(i), \sigma_h(i) \leq \frac{\sqrt{SA}}{d\mu}$ *for all* $s \in \mathcal{S}$, $a \in \mathcal{A}$, $h \in [H]$, $m \in [M]$, *and* $i \in [d]$ *for any function* $g : \mathcal{S} \to [0, 1]$.

To allow transfer learning to occur, we assume that the latent factors from the source MDPs span the space spanned by the target latent factors.

**Assumption 12.** *For latent factors $F_h$ from the target MDP and $F_{m,h}$ for the source MDPs, let $F = \{F_h, F_{h,m}\}_{h \in [H], m \in [M]}$. For latent factors $G_h$ from the target MDP and $G_{m,h}$ for the source MDPs, let $G = \{G_h, G_{h,m}\}_{h \in [H], m \in [M]}$. Then, there exists a non-empty set $\mathcal{B}(F), \mathcal{C}(G)$ such that the elements in $\mathcal{B}(F)$ are $B \in \mathbb{R}^{d', d, M}$ and the elements in $\mathcal{C}(G)$ are $C \in \mathbb{R}^{d', d, M}$ such that*

$$F_h(s, i) = \sum_{j \in [d], m \in [M]} B(i, j, m) F_{m,h}(s, j)$$

*and*

$$G_h(a, i) = \sum_{j \in [d], m \in [M]} C(i, j, m) G_{m,h}(a, j)$$

*for all* $i \in [d'], s \in \mathcal{S}, a \in \mathcal{A}$.

Similarly to the $(S, d, A)$ and $(S, S, d)$ Tucker rank settings, we introduce the transfer-ability coefficient that measures the difficulty of applying transfer learning.

**Definition 6** (Transfer-ability Coefficient). *Given a transfer RL problem that satisfies Assumptions 11, 12, and 3, let $\mathcal{F}$ be the set of all $\mathbb{R}^{S \times d}$ latent factor matrices with orthonormal columns that span the column space of $Q^*_{m,h}$ for all $m \in [M], h \in [H]$. Let $\mathcal{G}$ be the set of all $\mathbb{R}^{A \times d}$ latent factor matrices with orthonormal columns that span the row space of $Q^*_{m,h}$ for all $m \in [M], h \in [H]$. Then, we define $\alpha$ as*

$$\alpha := \max\left(\min_{F \in \mathcal{F}} \min_{B \in \mathcal{B}(F)} \max_{i \in [d'], j \in [d], m \in [M]} |B(i, j, m)|, \min_{G \in \mathcal{G}} \min_{C \in \mathcal{C}(F)} \max_{i \in [d'], j \in [d], m \in [M]} |C(i, j, m)|\right).$$

In contrast to the definition in the other Tucker rank settings, $\alpha$ bounds the largest coefficient used in the linear combination of $F_h$ and $G_h$. The algorithm that we use in this setting runs LR-EVI on each of the source MDPs, constructs a feature mapping $\hat{\phi}$ using the singular vectors from the singular value decomposition of $\bar{Q}_{m,h}$, and runs LSVI-UCB with $\hat{\phi}$. Since we use both sets of singular vectors to construct $\hat{\phi}$, our feature mapping has latent dimension $d^2 M^2$.

---

**Algorithm 8** Source Phase

---

**Input:** $\{N_h\}_{h\in[H]}$
 1: **for** $m \in [M]$ **do**
 2:  Run LR-EVI($\{N_h\}_{h\in[H]}$) on source MDP $m$ to obtain $\bar{Q}_{m,h}$.
 3:  Compute the singular value decomposition of $\bar{Q}_{m,h} = \hat{F}_{m,h}\hat{\Sigma}_{m,h}\hat{G}_{m,h}^{\top}$ .
 4: Compute the feature mapping $\hat{\phi} = \{\hat{\phi}_h\}_{h\in[H]}$ with

$$\hat{\phi}_h = \left\{ \frac{\sqrt{SA}}{d\mu}\hat{F}_{m,h}(:,i)\hat{G}_{n,h}(:,j) | i,j \in [d], m, n \in [M] \right\}.$$

---

---

**Algorithm 9** Target Phase: LSVI-UCB

---

**Input:** $\lambda, \beta_{k,h}, \hat{\phi}$
 1: Run LSVI-UCB with $\lambda, \beta_{k,h}$, and $\hat{\phi}$.

---

The above algorithm admits the following result.

**Theorem 15.** *Suppose Assumptions 11, 12, and 3 hold in the $(d,d,d)$ Tucker Rank setting, and set $\delta \in (0,1)$. Furthermore, assume that for any $\epsilon \in (0, \sqrt{H/T})$, $Q_{m,h} = r_{m,h} + P_{m,h}V_{m,h+1}$ has rank $d$ and is $\mu$-incoherent with condition number $\kappa$ for all $\epsilon$-optimal value functions $V_{h+1}$. Then, assuming $T \geq H/\alpha^2$, using at most*

$$\tilde{O}\left(d^{10}\mu^5\kappa^4(S+A)M^2H^4\alpha^4T\right)$$

*samples in the source problems, our algorithm has regret at most*

$$\tilde{O}\left(\sqrt{d^6M^6H^3T}\right)$$

*with probability at least $1-\delta$ for $\lambda = 1$ and $\beta_{h,k}$, which is a function of $d,H,T,M$.*

We now prove Theorem 15 using a similar argument to the one used in the proof of Theorem 10 except we show that the target MDP with our feature mapping satisfies Assumption B in [17] and apply LSVI-UCB.

*Proof.* Suppose the assumptions stated in Theorem 15 hold. Then, from Theorem 9[30], it follows that LR-EVI returns near-optimal $Q$ functions $\bar{Q}_{m,h}$ functions with singular value decomposition $\bar{Q}_{m,h} = \hat{F}_{m,h}\hat{\Sigma}_{m,h}\hat{G}_{m,h}$ that are $\sqrt{H}/(\kappa\alpha^2Md^4\mu\sqrt{T})$-optimal with probability at least $1-\delta/2$ using

$$\tilde{O}\left(d^{10}\mu^5\kappa^4(S+A)MH^4\alpha^4T\right)$$

samples on each source MDP. Since $T \geq H/\alpha^4$, from $\bar{\gamma} \leq \frac{1}{2}$ in Proposition 2. Thus, 3 states that there exists rotation matrices $R_{m,h,f}, R_{m,h,g}$ such that

$$\|(\hat{F}_{m,h}R_{m,h,f}-F_{m,h})(s)\|_2 \leq \frac{\sqrt{H}}{\alpha^4Md^2\sqrt{TS\mu}}, \quad \|(\hat{G}_{m,h}R_{m,h,g}-G_{m,h})(s)\|_2 \leq \frac{\sqrt{H}}{\alpha^4Md^2\sqrt{TA\mu}}$$

for all $s \in \mathcal{S}$ where the optimal $Q$ function have singular value decomposition $Q^*_{m,h} = F_{m,h}\Sigma_{m,h}G_{m,h}^{\top}$.

Then, from Assumption 6 and Definition 4, it follows that for all $(s', s, a) \in \mathcal{S} \times \mathcal{S} \times \mathcal{A}$, that

$$r_h(s, a) = \sum_{i \in [d]} \sigma_h(i) F_h(s, i) G_h(a, i)$$

$$= \sum_{i \in [d]} \sigma_h(i) \left( \sum_{j \in [d], m \in [M]} B(i, j, m) F_{h,m}(s, j) \right) \left( \sum_{l \in [d], n \in [M]} C(i, l, n) G_{h,n}(s, l) \right)$$

$$= \sum_{j, l \in [d], m, n \in [M]} F_{m,h}(s, j) G_{n,h}(a, l) \left( \sum_{i \in [d]} \sigma_h(i) B(i, j, m) C(i, l, n) \right)$$

$$= \theta_h \phi(s, a)_h^\top$$

where $\theta_h = \{ \sum_{i \in [d]} \sigma_h(i) B(i, j, m) C(i, l, n) \}_{j, l \in [d], m, n \in [M]}$ and $\phi_h = \{ F_{m,h}(s, j) G_{n,h}(a, l) \}_{j, l \in [d], m, n \in [M]}$ and $P(s'|\cdot, \cdot) = \mu_h(s') \phi_h^\top$ (with the same argument) for $\mu_h(s') = \{ \sum_{i,j,k \in [d]} U_h(i, j, k) V_h(s', i) B(j, l, m) C(k, x, n) \}_{l, x \in [d], m, n \in [M]}$. It follows that we can express $r_h = \tilde{F}_h \tilde{\Sigma}_h \tilde{G}_h^\top$ where $\tilde{F}_h$ be the $S \times d^2 M^2$ matrix from joining all $F_{m,h}$ $dM$ times, $\tilde{G}_h$ be the $A \times d^2 M^2$ matrix from joining all $G_{m,h}$ $dM$ times, and $\tilde{\Sigma}_h$ is the diagonal matrix with entries $\theta_h(j, l, m, n)$ for all $j, l \in [d], m, n \in [M]$. Note that the entries of $\tilde{\Sigma}_h$ and $\mu_h$ are at most $d\alpha^2$ larger than their original value.

Let $\hat{\tilde{F}}_h$ be the $S \times d^2 M^2$ matrix from joining all $\bar{F}_{m,h} R_{m,h,f}$ $dM$ times, and $\hat{\tilde{G}}_h$ be the $A \times d^2 M^2$ matrix from joining all $\bar{G}_{m,h} R_{m,h,g}$ $dM$ times. Let $\hat{\theta}_h$ satisfy

$$\tilde{\theta}_h \hat{\phi}_h^\top = \hat{\tilde{F}}_h \tilde{\Sigma}_h \hat{\tilde{G}}_h^\top.$$

Therefore, it follows for all $(s', s, a, h) \in \mathcal{S} \times \mathcal{S} \times \mathcal{A} \times [H]$,

$$|r_h(s, a) - \tilde{\theta}_h \hat{\phi}_h^\top(s, a)| = |\tilde{F}_h \tilde{\Sigma}_h \tilde{G}_h^\top(s, a) - \hat{\tilde{F}}_h \tilde{\Sigma}_h \hat{\tilde{G}}_h^\top(s, a)|$$

$$\leq |\tilde{F}_h \tilde{\Sigma}_h \tilde{G}_h^\top(s, a) - \hat{\tilde{F}}_h \tilde{\Sigma}_h \tilde{G}_h^\top(s, a)| + |\hat{\tilde{F}}_h \tilde{\Sigma}_h \tilde{G}_h^\top(s, a) - \hat{\tilde{F}}_h \tilde{\Sigma}_h \hat{\tilde{G}}_h^\top(s, a)|$$

$$\leq dM \sum_{m \in M} \|F_{m,h}(s) - \hat{F}_{m,h}(s) R_{m,h,f}\|_2 \|\tilde{\Sigma}_h\|_{op} \max_{m \in M} \sqrt{d} \|G_{m,h}\|_\infty$$

$$+ \sqrt{d} \max_{m \in M} \|\hat{F}_{m,h}\|_\infty \|\tilde{\Sigma}_h\|_{op} \|G_{m,h}(a) - \hat{G}_{m,h}(s) R_{m,h,g}\|_2$$

$$\leq d^{3/2} M \sum_{m \in M} \|F_{m,h}(s) - \hat{F}_{m,h}(s) R_{m,h,f}\|_2 \|\tilde{\Sigma}_h\|_{op} \max_{m \in M} \|G_{m,h}\|_\infty$$

$$+ \max_{m \in M} (\|F_{m,h}\| + \|F_{m,h} R - \hat{F}_{m,h}\|_\infty) \|\tilde{\Sigma}_h\|_{op} \|G_{m,h}(a) - \hat{G}_{m,h}(s) R_{m,h,g}\|_2$$

$$\leq C d^3 M \frac{\sqrt{S}}{\sqrt{\mu}} \sum_{m \in M} \|F_{m,h}(s) - \hat{F}_{m,h}(s) R_{m,h,f}\|_2$$

$$\leq C \frac{dM \sqrt{H}}{\sqrt{T}},$$

where the second inequality comes from the triangle inequality and Cauchy-Schwarz inequality, the third inequality comes from the triangle inequality, and the fourth inequality comes from the definition of incoherence, our bound on the entries of $\tilde{\Sigma}$ and the left term dominating the right term, and the final inequality is due to our singular vector perturbation bounds. Similarly,

$$\|P_h(\cdot|s, a) - \tilde{\mu_h}(s') \hat{\phi}_h(s, a)^\top\|_{TV} \leq C \frac{dM \sqrt{H}}{\sqrt{T}}$$

for some absolute constant $C > 0$ where $\tilde{\mu_h}(s')$ satisfies $\hat{\tilde{F}}_h \tilde{\Sigma}_h(s') \hat{\tilde{G}}_h^\top$ and $\tilde{\Sigma}_h(s')$ is the $d^2 M^2 \times d^2 M^2$ diagonal matrix with entries $\mu_h(s', l, k, m, n)$ with the same argument using the regularity condition on $\mu_h$.

Therefore, $\hat{\phi}$ satisfies Assumption B in [17] with $\xi = O(dM\sqrt{H/T})$. Running LSVI-UCB with $\hat{\phi}$ admits a regret bound of

$$Regret(T) \le C\left(\sqrt{d^6 M^6 H^3 T \iota^2} + \frac{d^3 M^3 H^{3/2} T \sqrt{\iota}}{\sqrt{T}}\right) \in \tilde{O}\left(d^6 M^6 H^3 T\right)$$

for some absolute constant $C > 0$ with probability at least $1 - \delta$ from a final union bound. The sample complexity in the source phase is

$$\tilde{O}\left(d^{10} \mu^5 \kappa^4 (S + A) M H^4 \alpha^4 T\right).$$

which completes the proof. $\qquad\square$

# I $\ell_\infty$ eigen perturbation bound with non-uniform noise: rank-$r$ case

Suppose $A = A^* + D \in \mathbb{R}^{m \times n}$, where $A^*$ is the true rank-$r$ matrix and $D$ is the noise matrix. Suppose $A^*$ has SVD $A^* = U^* \Sigma^* V^{*\top}$, where $U^* \in^{m \times r}, V^* \in^{n \times r}$ are orthonormal matrices containing the left and right singular vectors, respectively, and $\Sigma^* = \mathrm{diag}(\sigma_1^*, \ldots, \sigma_r^*)$ is a diagonal matrix of the singular values. Similarly, suppose $A$ has SVD $A = U \Sigma V^\top + U' \Sigma' V'^\top$, where $U \in^{m \times r}, V \in^{n \times r}$ and $\Sigma = \mathrm{diag}(\sigma_1, \ldots, \sigma_r)$ correspond to the top-$r$ singular vectors/values, and $U' \in^{m \times (m-r)}, V' \in^{m \times (n-r)}$ and $\Sigma \in^{(m-r) \times (n-r)}$ correspond to the bottom singular vectors/values.

The following proposition is a generalization of the perturbation bound in [12] to the setting with non-uniform noise.

**Proposition 2.** *Suppose* $\overline{\gamma} := \frac{\|D\|_{\mathrm{op}}}{\sigma_r^*} < \frac{1}{2}$. *There exist two orthonormal matrices* $\overline{R}, \underline{R} \in^{r \times r}$ *such that*

$$\left\|\left(U - U^* \overline{R}\right)_{i\cdot}\right\|_2 \le \|U_{i\cdot}^*\|_2 \cdot 8\overline{\gamma} + \|D_{i\cdot}\|_2 \cdot \frac{1}{\sigma_r^*(1 - \overline{\gamma})}, \qquad \forall i \in [m],$$

$$\left\|\left(V - V^* \underline{R}\right)_{j\cdot}\right\|_2 \le \|V_{j\cdot}^*\|_2 \cdot 8\overline{\gamma} + \|D_{\cdot j}\|_2 \cdot \frac{1}{\sigma_r^*(1 - \overline{\gamma})}, \qquad \forall j \in [n].$$

To prove Proposition 2, we need some notations. Let $\overline{H} := U^\top U^* \in^{r \times r}$ and its SVD be $\overline{H} = \overline{A} \overline{\Lambda} \overline{B}^\top$, where $\overline{A}, \overline{B} \in^{r \times r}$ are orthonormal, and $\overline{\Lambda} = \mathrm{diag}(\overline{\sigma}_1, \ldots, \overline{\sigma}_r) \in^{r \times r}$ is a diagonal matrix. Similarly, let $\underline{H} := V^\top V^* \in^{r \times r}$ and its SVD be $\underline{H} = \underline{A}\underline{\Lambda}\underline{B}^\top$. Finally, define the orthonormal matrices $\mathrm{sgn}(\overline{H}) := \overline{A}\overline{B}^\top$ and $\mathrm{sgn}(\underline{H}) := \underline{A}\underline{B}^\top$. (The function $\mathrm{sgn}(\cdot)$ is called the matrix sign function.)

We need a technical lemma, which generalizes [2, Lemma 3] to the asymmetric case.

**Lemma 3** (Properties of matrix sign function). *We have* $\left\|\overline{H}\right\|_{\mathrm{op}} \le 1$ *and* $\|\underline{H}\|_{\mathrm{op}} \le 1$. *When* $\overline{\gamma} := \frac{\|D\|_{\mathrm{op}}}{\sigma_r^*} < \frac{1}{2}$, *we have*

$$\sqrt{\left\|\overline{H} - \mathrm{sgn}(\overline{H})\right\|_{\mathrm{op}}} \le \left\|UU^\top - U^*U^{*\top}\right\|_{\mathrm{op}} \le \frac{\max\left\{\|U^*D\|_{\mathrm{op}}, \|DV^*\|_{\mathrm{op}}\right\}}{(1 - \overline{\gamma})\sigma_r^*} \le \frac{\overline{\gamma}}{1 - \overline{\gamma}},$$

$$\sqrt{\left\|\underline{H} - \mathrm{sgn}(\underline{H})\right\|_{\mathrm{op}}} \le \left\|VV^\top - V^*V^{*\top}\right\|_{\mathrm{op}} \le \frac{\max\left\{\|U^*D\|_{\mathrm{op}}, \|DV^*\|_{\mathrm{op}}\right\}}{(1 - \overline{\gamma})\sigma_r^*} \le \frac{\overline{\gamma}}{1 - \overline{\gamma}},$$

*and*

$$\left\|\Sigma\underline{H} - \overline{H}\Sigma\right\|_{\mathrm{op}} \le 2\|D\|_{\mathrm{op}}, \qquad \left\|\overline{H}^\top \Sigma - \Sigma\underline{H}^\top\right\|_{\mathrm{op}} \le 2\|D\|_{\mathrm{op}},$$

*and*

$$\left\|\overline{H}^{-1}\right\|_{\mathrm{op}} \le \frac{(1 - \overline{\gamma})^2}{(1 - 2\overline{\gamma})}, \qquad \left\|\underline{H}^{-1}\right\|_{\mathrm{op}} \le \frac{(1 - \overline{\gamma})^2}{(1 - 2\overline{\gamma})}.$$

*Proof.* We prove the bounds for $\overline{H}$. The proof for $\underline{H}$ is identical. Clearly $\left\|\overline{H}\right\|_{\mathrm{op}} \le \|U\|_{\mathrm{op}} \|U^*\|_{\mathrm{op}} = 1$.

By Weyl's inequality, we have $|\sigma_i^* - \sigma_i| \leq \|D\|_{\mathrm{op}}, \forall i \in [m]$, hence

$$\sigma_r^* - \sigma_{r+1} \geq \sigma_r^* - \|D\|_{\mathrm{op}} = (1 - \overline{\gamma})\sigma_r^* > \frac{1}{2}\sigma_r^* > 0$$

by assumption. On the other hand, by standard perturbation theory, $1 \geq \overline{\sigma}_1 \geq \cdots \geq \overline{\sigma}_r \geq 0$ are the cosines of the principal angles $0 \leq \overline{\theta}_1 \leq \cdots \leq \overline{\theta}_r \leq \pi/2$ between the column spaces of $U$ and $U^*$, and $\sin \overline{\theta}_r = \|UU^\top - U^*U^{*\top}\|_{\mathrm{op}}$. Hence $\|\mathrm{sgn}(\overline{H}) - \overline{H}\|_{\mathrm{op}} = 1 - \cos \overline{\theta}_r$. By Wedin's $\sin \Theta$ Theorem, we have

$$\sin \overline{\theta}_r \leq \frac{\max\left\{\|U^*D\|_{\mathrm{op}}, \|DV^*\|_{\mathrm{op}}\right\}}{\sigma_r^* - \sigma_{r+1}} \leq \frac{\max\left\{\|U^*D\|_{\mathrm{op}}, \|DV^*\|_{\mathrm{op}}\right\}}{(1 - \overline{\gamma})\sigma_r^*}.$$

Since $\cos \overline{\theta}_r \geq \cos^2 \overline{\theta}_r = 1 - \sin^2 \overline{\theta}_r$, we obtain

$$\sqrt{\|\mathrm{sgn}(\overline{H}) - \overline{H}\|_{\mathrm{op}}} = \sqrt{1 - \cos \overline{\theta}_r} \leq \sin \overline{\theta}_r \leq \frac{\max\left\{\|U^*D\|_{\mathrm{op}}, \|DV^*\|_{\mathrm{op}}\right\}}{(1 - \overline{\gamma})\sigma_r^*} \leq \frac{\|D\|_{\mathrm{op}}}{(1 - \overline{\gamma})\sigma_r^*} = \frac{\overline{\gamma}}{1 - \overline{\gamma}}.$$

Note that

$$U^\top DV^* = U^\top (A - A^*)V^* = \Sigma V^\top V^* - U^\top U^*\Sigma^* = \Sigma \underline{H} - \overline{H}\Sigma^*, \qquad \text{and}$$

Hence

$$\begin{aligned}\|\Sigma \underline{H} - \overline{H}\Sigma\|_{\mathrm{op}} &\leq \|\Sigma \underline{H} - \overline{H}\Sigma^*\|_{\mathrm{op}} + \|\overline{H}(\Sigma^* - \Sigma)\|_{\mathrm{op}} \\ &= \|U^\top DV^*\|_{\mathrm{op}} + \|\overline{H}(\Sigma^* - \Sigma)\|_{\mathrm{op}} \\ &\leq \|D\|_{\mathrm{op}} + \|\overline{H}\|_{\mathrm{op}}\|\Sigma^* - \Sigma\|_{\mathrm{op}} \\ &\leq 2\|D\|_{\mathrm{op}},\end{aligned}$$

where we use $\|\overline{H}\|_{\mathrm{op}} \leq 1$ and $\|\Sigma^* - \Sigma\|_{\mathrm{op}} \leq \|D\|_{\mathrm{op}}$. Similarly,

$$U^{*\top}DV = U^{*\top}(A - A^*)V = U^{*\top}U\Sigma - \Sigma^*V^{*\top}V^\top = \overline{H}^\top\Sigma - \Sigma^*\underline{H}^\top,$$

so a similar argument as above gives $\|\overline{H}^\top\Sigma - \Sigma\underline{H}^\top\|_{\mathrm{op}} \leq 2\|D\|_{\mathrm{op}}$.

Finally, recall that $\|\overline{H} - \mathrm{sgn}(\overline{H})\|_{\mathrm{op}} \leq \left(\frac{\overline{\gamma}}{1 - \overline{\gamma}}\right)^2$ and note the simple identity $X^{-1} - Y^{-1} = X^{-1}(Y - X)Y^{-1}$. It follows that

$$\begin{aligned}\|\overline{H}^{-1} - \mathrm{sgn}(\overline{H})^{-1}\|_{\mathrm{op}} &\leq \|\overline{H}^{-1}\|_{\mathrm{op}}\|\overline{H} - \mathrm{sgn}(\overline{H})\|_{\mathrm{op}}\|\mathrm{sgn}(\overline{H})^{-1}\|_{\mathrm{op}} \\ &= \left[\sigma_r\left(\mathrm{sgn}(\overline{H}) + (\overline{H} - \mathrm{sgn}(\overline{H}))\right)\right]^{-1}\|\overline{H} - \mathrm{sgn}(\overline{H})\|_{\mathrm{op}} \\ &\leq \frac{1}{1 - \left(\frac{\overline{\gamma}}{1 - \overline{\gamma}}\right)^2} \cdot \left(\frac{\overline{\gamma}}{1 - \overline{\gamma}}\right)^2 = \frac{\overline{\gamma}^2}{1 - 2\overline{\gamma}}.\end{aligned}$$

It follows that

$$\|H^{-1}\|_{\mathrm{op}} \leq \|\mathrm{sgn}(\overline{H})^{-1}\|_{\mathrm{op}} + \|\overline{H}^{-1} - \mathrm{sgn}(\overline{H})^{-1}\|_{\mathrm{op}} \leq 1 + \frac{\overline{\gamma}^2}{1 - 2\overline{\gamma}} = \frac{(1 - \overline{\gamma})^2}{1 - 2\overline{\gamma}}.$$

This completes the proof of Lemma 3. $\qquad\qquad\square$

We are now ready to prove Proposition 2. We have

$$\begin{aligned}U - U^*\mathrm{sgn}(\overline{H})^\top &= AV\Sigma^{-1} - U^*\mathrm{sgn}(\overline{H})^\top && A = U\Sigma V^\top \\ &= (A^* + D)V\Sigma^{-1} - U^*\mathrm{sgn}(\overline{H})^\top \\ &= U^*\Sigma^*V^{*\top}V\Sigma^{-1} - U^*\mathrm{sgn}(\overline{H})^\top + DV\Sigma^{-1} && A^* = U^*\Sigma^*V^{*\top} \\ &= U^*\Sigma^*\underline{H}^\top\Sigma^{-1} - U^*\mathrm{sgn}(\overline{H})^\top + DV\Sigma^{-1}. && \underline{H} = V^\top V^*\end{aligned}$$

But $\Sigma^*\underline{H}^\top = (\Sigma + \Sigma^* - \Sigma)\underline{H}^\top = \overline{H}^\top\Sigma + \left(\Sigma\underline{H}^\top - \overline{H}^\top\Sigma\right) + (\Sigma^* - \Sigma)\underline{H}^\top$. Hence

$$U - U^*\operatorname{sgn}(\overline{H})^\top = U^*\left[\overline{H}^\top\Sigma + \left(\Sigma\underline{H}^\top - \overline{H}^\top\Sigma\right) + (\Sigma^* - \Sigma)\underline{H}^\top\right]\Sigma^{-1} - U^*\operatorname{sgn}(\overline{H})^\top + DV\Sigma^{-1}$$

$$= U^*\overline{H}^\top + U^*\left(\Sigma\underline{H}^\top - \overline{H}^\top\Sigma\right)\Sigma^{-1} + U^*(\Sigma^* - \Sigma)\underline{H}^\top\Sigma^{-1} - U^*\operatorname{sgn}(\overline{H})^\top + DV\Sigma^{-1}$$

$$= U^*\left(\overline{H} - \operatorname{sgn}(\overline{H})\right)^\top + U^*\left(\Sigma\underline{H}^\top - \overline{H}^\top\Sigma\right)\Sigma^{-1} + U^*(\Sigma^* - \Sigma)\underline{H}^\top\Sigma^{-1} + DV\Sigma^{-1}.$$

For each $i \in [m]$, we can bound the $i$-th row of $U - U^*\operatorname{sgn}(\overline{H})^\top$ as

$$\left\|\left(U - U^*\operatorname{sgn}(\overline{H})^\top\right)_{i\cdot}\right\|_2 \leq \|U_{i\cdot}^*\|_2\left(\left\|\overline{H} - \operatorname{sgn}(\overline{H})\right\|_{\mathrm{op}} + \left\|\Sigma\underline{H}^\top - \overline{H}^\top\Sigma\right\|_{\mathrm{op}}\left\|\Sigma^{-1}\right\|_{\mathrm{op}} + \|\Sigma^* - \Sigma\|_{\mathrm{op}}\|\underline{H}\|_{\mathrm{op}}\left\|\Sigma^{-1}\right\|_{\mathrm{op}}\right)$$

$$+ \|D_{i\cdot}\|_2\|V\|_{\mathrm{op}}\left\|\Sigma^{-1}\right\|_{\mathrm{op}}$$

$$\overset{(i)}{\leq} \|U_{i\cdot}^*\|_2\left(\left(\frac{\overline{\gamma}}{1 - \overline{\gamma}}\right)^2 + 2\|D\|_{\mathrm{op}}\cdot\frac{1}{\sigma_r^*(1 - \overline{\gamma})} + \|D\|_{\mathrm{op}}\cdot 1\cdot\frac{1}{\sigma_r^*(1 - \overline{\gamma})}\right) + \|D_{i\cdot}\|_2\cdot\frac{1}{\sigma_r^*(1 - \overline{\gamma})}$$

$$= \|U_{i\cdot}^*\|_2\left(\left(\frac{\overline{\gamma}}{1 - \overline{\gamma}}\right)^2 + \frac{3\overline{\gamma}}{1 - \overline{\gamma}}\right) + \|D_{i\cdot}\|_2\cdot\frac{1}{\sigma_r^*(1 - \overline{\gamma})}$$

$$\overset{(ii)}{\leq} \|U_{i\cdot}^*\|_2\cdot 8\overline{\gamma} + \|D_{i\cdot}\|_2\cdot\frac{1}{\sigma_r^*(1 - \overline{\gamma})},$$

where step (i) follows from Lemma 3 and the bound

$$\left\|\Sigma^{-1}\right\|_{\mathrm{op}} = \frac{1}{\sigma_r} \leq \frac{1}{\sigma_r^* - \|D\|_{\mathrm{op}}} = \frac{1}{\sigma_r^*(1 - \overline{\gamma})},$$

and step (ii) holds since $\overline{\gamma} \leq \frac{1}{2}$. Setting $\overline{R} = \operatorname{sgn}(\overline{H})^\top$ proves the first inequality in Proposition 2.

The second inequality in Proposition 2 can be established by a similar argument.

**Corollary 3.** *Suppose the setting of Proposition 2 holds. Furthermore, assume that $A^*$ is $\mu$-incoherent with condition number $\kappa$ and $\|A^*\|_\infty \geq C$ for some constant $C > 0$. Then, there exist two orthonormal matrices $\overline{R}, \underline{R} \in^{r\times r}$ such that*

$$\left\|\left(U - U^*\overline{R}\right)_{i\cdot}\right\|_2 \leq \frac{16\kappa(\mu d)^{3/2}\|D\|_\infty}{C\sqrt{m}}, \qquad \forall i \in [m],$$

$$\left\|\left(V - V^*\underline{R}\right)_{j\cdot}\right\|_2 \leq \frac{16\kappa(\mu d)^{3/2}\|D\|_\infty}{C\sqrt{n}}, \qquad \forall j \in [n].$$

*Proof.* From the definition of the $\ell_\infty$ norm, there exists $(i, j) \in [m] \times [m]$ such that $\|A^*\|_\infty = |A_{i,j}|$. Then, it follows that

$$\|A^*\|_\infty = |A_{i,j}| = |(U^*\Sigma^*V^{*\top})_{i,j}|$$

$$\leq \|\Sigma^*\|_{op}\|U_i^*\|_2\|V_j^*\|_2$$

$$\leq \frac{\sigma_1\mu d}{\sqrt{mn}}$$

$$\leq \frac{\kappa\sigma_d\mu d}{\sqrt{mn}}.$$

Furthermore, we note that $\|D\|_{op} \leq \|D\|_\infty\sqrt{mn}$. Thus, the result follows from Proposition 2 and using the definition of incoherence and the above inequalities. $\square$

# J   Proofs of LSVI-UCB-(S, d, A) and LSVI-UCB-(S, S, d)

In this section, we present the proofs of Theorem 8 and 9 and the related lemmas. The proofs, theorems, and lemmas are modifications of the analogous result/proof in [17] tailored to our Tucker rank settings.

## J.1 LSVI-UCB-(S, d, A)

To prove Theorem 8, we first introduce notation used throughout this section and then present auxiliary lemmas. We reiterate that the steps and analysis in this subsection are essentially the same as the proofs in [17], except we have $A$ copies of the weight vectors and gram matrices. We let $\Lambda_{h,k}^a, w_{h,k}^a, Q_h^k, V_h^k, \pi^k$ as the parameters/estimates at episode $k$. Furthermore, let $V_h^k = \max_{a \in \mathcal{A}} Q_h^k(s, a)$ and $\pi^k$ be the greedy policy w.r.t. $Q_h^k$. We let $F_h^k := F_h(s_h^k)$.

To prove the desired regret bound, we first need to prove that $\mathbb{P}V - \hat{P}V$ concentrates about 0, which differs from typical concentration inequalities for self normalized processes due to the value function not being constant. Thus, we first bound the log of the covering number of our function class for the value function to control the error of our algorithm's estimate.

**Lemma 4.** *Let $\mathcal{V}$ denote a function class of mappings from $\mathcal{S}$ to $\mathbb{R}$ with the following parametric form*

$$V(\cdot) = \min \left\{ \max_{a \in \mathcal{A}} w_a^\top F(\cdot) + \beta \sqrt{F(\cdot)^\top \Lambda_a^{-1} F(\cdot)}, H \right\}$$

*where the parameters $w_a, \beta, \Lambda_a$ satisfy $\|w_a\| \leq L$ for all $a \in \mathcal{A}$, $\beta \in [0, B]$, and the minimum eigenvalue of $\Lambda_a$ is greater than $\lambda$. Assume that $\|F(s)\| \leq 1$ for all $s \in \mathcal{S}$, and let $\mathcal{N}_\epsilon$ be the $\epsilon$-covering number of $\mathcal{V}$ with respect to the distance $dist(V, V') = \sup_s |V(s) - V'(s)|$. Then,*

$$\log(\mathcal{N}_\epsilon) \leq d \log(1 + 4L/\epsilon) + d^2 \log(1 + 8d^{1/2} B^2/(\lambda \epsilon^2))$$

*Proof.* Equivalently, we reparameterize the function class by $D_a = \beta^2 \Lambda_a^{-1}$,

$$V(\cdot) = \min \left\{ \max_{a \in \mathcal{A}} w_a^\top F(\cdot) + \sqrt{F(\cdot)^\top D_a F(\cdot)}, H \right\}$$

where $\|w_a\| \leq L$ and $\|D_a\| \leq B^2/\lambda$ for all $a \in \mathcal{A}$. For any $V_1, V_2 \in \mathcal{V}$, let them take the form in the above equation with parameters $(w_a, D_a)$ and $(w_a', D_a')$. Since $\min(\cdot, H)$ and $\max_{a \in \mathcal{A}}$ are contraction maps,

$$dist(V_1 - V_2) \leq \sup_{s,a} |(w_a^\top F(s) + \sqrt{F(s)^\top D_a F(s)}) - (w_a'^\top F(s) + \sqrt{F(s)^\top D_a' F(s)})|$$

$$\leq \sup_{a, F:\|F\| \leq 1} |(w_a^\top F + \sqrt{F^\top D_a F}) - (w_a'^\top F + \sqrt{F^\top D_a' F})|$$

$$\leq \max_{a \in \mathcal{A}} \left[ \sup_{F:\|F\| \leq 1} |(w_a - w_a')^\top F| + \sup_{F:\|F\| \leq 1} |\sqrt{F^\top (D_a - D_a') F}| \right]$$

$$\leq \max_{a \in \mathcal{A}} \left[ \|w_a - w_a'\| + \sqrt{\|D_a - D_a'\|} \right] \leq \max_{a \in \mathcal{A}} \left[ \|w_a - w_a'\| + \sqrt{\|D_a - D_a'\|_F} \right]$$

where the holds because $|\sqrt{x} - \sqrt{y}| \leq \sqrt{|x - y|}$ for any $x, y \geq 0$. Let $C_w$ be an $\epsilon/2$-cover of $\{w \in \mathbb{R}^d | \|w\| \leq L\}$ with respect to the 2-norm, and $C_D$ be an $\epsilon^2/4$-cover of $\{D \in \mathbb{R}^{d \times d} | \|D\|_F \leq d^{1/2} B^2/\lambda\}$ with respect to the Frobenius norm. By Lemma D.5 from [17], we know that,

$$|C_w| \leq (1 + 4L/\epsilon)^d, \quad |C_D| \leq (1 + 8d^{1/2} B^2/(\lambda \epsilon^2))^{d^2}.$$

Hence, for any $V_1$, there exists a $w_a' \in C_w$ and $D_a' \in C_D$ for all $a \in \mathcal{A}$ such that $V_2$ parameterized by $(w_a', D_a')$ satisfies $dist(V_1, V_2) \leq \epsilon$. Hence, $\mathcal{N}_\epsilon \leq |C_w||C_D|$, which gives:

$$\log(\mathcal{N}_\epsilon) \leq \log(|C_w|) + \log(|C_D|) \leq d \log(1 + 4\epsilon L/\epsilon) + d^2 \log(1 + 8d^{1/2} B^2/(\lambda \epsilon^2)).$$

$\square$

We next bound the $\ell_2$-norm of the weight vector of any policy.

**Lemma 5.** *Let $\{w_h^a\}$ be the weight vector of some policy $\pi$, i.e., $Q_h^\pi(s, a) = F_h(s)^\top w_h^a$. Then,*

$$\|w_h^a\| \leq 2H\sqrt{d}.$$

*Proof.* Note that $Q_h^\pi(s,a) = r_h(s,a) + \sum_{s'} V_{h+1}^\pi(s')P_h(s'|s,a)$, so, using the low-rank representations of $r_h, P_h$, it follows that

$$\|w_h^a\|_2 = \|W_h(a,\cdot) + \sum_{s'} V^\pi(s')U_h(s',a,\cdot)\|_2 \leq \|W_h(a,\cdot)\|_2 + \|\sum_{s'} V^\pi(s')U_h(s',a,\cdot)\|_2 \leq \sqrt{d} + H\sqrt{d}.$$

Recall that since the feature mapping was scaled up $(F_h = \sqrt{\frac{S}{d\mu}}F')$ to ensure the max entries of $F$ and $W, U$ are on the same scale, it follows that $\|W_h(a)\|_2 \leq 1$ and $\|\sum_{s' \in \mathcal{S}} V_{h+1}^\pi(s')U_h(s',a)\|_2 \leq H$. Therefore, it follows that $\|w_h^a\|_2 \leq 2H\sqrt{d}$. $\qquad\square$

We next bound the 2-norm of the weight vector from the algorithm.

**Lemma 6.** *For any $k, h, a$, the weight vector from the above algorithm satisfies $\|w_{h,k}^a\| \leq 2H\sqrt{dk/\lambda}$.*

*Proof.* For any $F_h(s) = f$, we have for any $a \in \mathcal{A}$

$$|f^\top w_{h,k}^a| = |f^\top (\Lambda_h^a)^{-1} \sum_{t \in T_{h,k-1}^a} F_h(s_h^t)\left[r_h(s_h^t, a) + \max_{a' \in \mathcal{A}} Q_{h+1}(s_{h+1}^t, a')\right]|$$

$$\leq |\sum_{t \in T_{h,k-1}^a} f^\top (\Lambda_h^a)^{-1} F_h(s_h^t)|2H$$

$$\leq 2H\sqrt{(\sum_{t \in T_{h,k-1}^a} f^\top (\Lambda_h^a)^{-1} f^\top)(\sum_{t \in T_{h,k-1}^a} F_h(s_h^t)^\top (\Lambda_h^a)^{-1} F_h(s_h^t))}$$

$$\leq 2H\|f\|_2\sqrt{d|T_{h,k-1}^a|/\lambda} \leq 2H\sqrt{dk/\lambda},$$

where the second to last inequality is due to Lemma D.1 from [17] and that Rayleigh quotients are upper bounded by the largest eigenvalue of the matrix (Theorem 4.2.2 from [14]). The last inequality is due to $\|F_h(s)\|_2 \leq 1$ and $|T_{h,k-1}^a| \leq k$. $\qquad\square$

The above two lemmas are necessary in ensuring that the estimation error is small as they depend multiplicatively on the norm of the weight vectors. We next prove the main concentration lemma that controls the estimate.

**Lemma 7.** *Let $c'$ be the constant in the definition of $\beta_{k,h}^a = c'dH\sqrt{\iota}$. Then, there exists a constant $C > 0$ that is independent of $c'$ such that for any $\delta \in (0,1)$, if we let the event $\mathcal{X}$ be*

$$\forall (a,k,h) \in \mathcal{A} \times [K] \times [H] : \left\|\sum_{t \in T_{h,k-1}^a} F_h(s_h^t)(V_{h+1}^t(s_{h+1}^t) - P_h V_{h+1}^t(s_h^t, a))\right\|_{(\Lambda_{h,t}^a)^{-1}} \leq CdH\sqrt{\chi}$$

*where $\chi = \log(2(c'+1)dTA/\delta)$, then $\mathbb{P}(\mathcal{X}) \geq 1 - \delta/2$.*

*Proof.* From Lemma 6, we have $\|w_{h,k}^a\|_2 \leq 2H\sqrt{dk/\lambda}$. By construction, the minimum eigenvalue of $\Lambda_{h,t}^a$ is lower bounded by $\lambda$. Then, from Lemma D.4 in [17] (with $\delta' = \delta/A$), Lemma 4, and taking a union bound over actions, we have that with probability at least $1 - \delta'/2$, for all $a \in \mathcal{A}$ and $\epsilon > 0$,

$$\left\|\sum_{t \in T_{h,k-1}^a} F_h(s_h^t)(V_{h+1}^t(s_{h+1}^t) - P_h V_{h+1}^t(s_h^t, a))\right\|_{(\Lambda_{h,t}^a)^{-1}}^2 \leq 4H^2\left[\frac{d}{2}\log((k+\lambda)/\lambda)) + \log(2A\mathcal{N}_\epsilon/\delta)\right] + 8k^2\epsilon^2/\lambda.$$

Then, from Lemma 4, it follows that with probability at least $1 - \delta'/2$, for all $a \in A$ and $\epsilon > 0$,

$$\left\|\sum_{t \in T_{h,k-1}^a} F_h(s_h^t)(V_{h+1}^t(s_{h+1}^t) - P_h V_{h+1}^t(s_h^t, a))\right\|_{(\Lambda_{h,t}^a)^{-1}}^2$$

$$\leq 4H^2\left[\frac{d}{2}\log((k+\lambda)/\lambda)) + \log(2A/\delta) + d\log(1 + 4L/\epsilon) + d^2\log(1 + 8d^{1/2}\beta^2/(\lambda\epsilon^2))\right] + 8k^2\epsilon^2/\lambda.$$

Choosing $\lambda = 1, \beta_{k,h}^a = CdH\iota, \epsilon = dH/k$ gives that there exists some constant $C'$ such that

$$\left\| \sum_{t \in T_{h,k-1}^a} F_h(s_h^t)(V_{h+1}^t(s_{h+1}^t) - P_h V_{h+1}^t(s_h^t, a)) \right\|_{(\Lambda_{h,t}^a)^{-1}}^2 \leq C'd^2H^2 \log(2(c_\beta + 1)dTA/\delta).$$

$\square$

We next recursively bound the difference between the value function maintained in the algorithm and the true value function of any policy $\pi$. We upperbound this with their expected difference and an additional error term, which is bounded with high probability.

**Lemma 8.** *There exists a constant $c_\beta$ such that $\beta_{k,h}^a = c_\beta dH\sqrt{\iota}$ where $\iota = \log(2dAT/p)$ and for any fixed policy $\pi$, on the event $\mathcal{X}$ defined in Lemma 7, we have for all $(s, a, h, k) \in S \times A \times [H] \times [K]$:*

$$F_h(s)^\top w_{h,k}^a - Q_h^\pi(s, a) = P_h(V_{h+1}^k - V_{h+1}^\pi)(s, a) + \Delta_h^k(s, a),$$

*where $|\Delta_h^k(s, a)| \leq \beta_{k,h}^a \sqrt{F_h(s)^\top (\Lambda_{h,k}^a)^{-1} F_h(s)}.$*

*Proof.* From our low-rank assumption, it follows that

$$Q_h^\pi(s, a) = F_h(s)^\top w_h^a = (r_h + P_h V_{h+1}^\pi)(s, a),$$

which following the steps from the proof of Lemma B.4 in [17] gives

$$w_{h,k}^a - w_h^{a,\pi} = -\lambda(\Lambda_{h,k}^a)^{-1} w_h^{a,\pi}$$
$$+ (\Lambda_{h,k}^a)^{-1} \sum_{t \in T_{k-1,h}^a} F_h(s_h^t)(V_{h+1}^k(s_{h+1}^t) - P_h V_{h+1}^k(s_h^t, a))$$
$$+ (\Lambda_{h,k}^a)^{-1}) \sum_{t \in T_{k-1,h}^a} F_h(s_h^t) P_h(V_{h+1}^k - V_{h+1}^\pi)(s_h^t, a)$$

Now, we bound each term $(q_1^a, q_2^a, q_3^a)$ individually, for the first term, note for all $a \in \mathcal{A}$

$$|F_h(s)^\top q_1^a| = |\lambda F_h(s)^\top (\Lambda_{h,k}^a)^{-1} w_h^{a,\pi}| \leq \sqrt{\lambda} \|w_h^{a,\pi}\| \sqrt{F_h(s)^\top (\Lambda_{h,k}^a)^{-1} F_h(s)}$$

For the second term, given the event $\mathcal{X}$, we have from Cauchy Schwarz

$$|F_h(s)^\top q_2^a| \leq CdH\sqrt{\chi} \sqrt{F_h(s)^\top (\Lambda_{h,k}^a)^{-1} F_h(s)}$$

where $\chi = \log(2(c_\beta + 1)dTA/p)$. For the third term,

$$F_h(s)^\top q_3^a = F_h(s)^\top \left( (\Lambda_{h,k}^a)^{-1}) \sum_{t \in T_{k-1,h}^a} F_h(s_h^t) P_h(V_{h+1}^k - V_{h+1}^\pi)(s_h^t, a) \right)$$

$$= F_h(s)^\top \left( (\Lambda_{h,k}^a)^{-1}) \sum_{t \in T_{k-1,h}^a} F_h(s_h^t) F_h(s_h^t)^\top \sum_{s' \in \mathcal{S}} (V_{h+1}^k - V_{h+1}^\pi)(s') U_h(s', a) \right)$$

$$= F_h(s)^\top \left( \sum_{s' \in \mathcal{S}} (V_{h+1}^k - V_{h+1}^\pi)(s') U_h(s', a) \right)$$

$$- \lambda F_h(s)^\top \left( (\Lambda_{h,k}^a)^{-1}) \sum_{s' \in \mathcal{S}} (V_{h+1}^k - V_{h+1}^\pi)(s') U_h(s', a) \right)$$

We note that

$$F_h(s)^\top \left( \sum_{s' \in \mathcal{S}} (V_{h+1}^k - V_{h+1}^\pi)(s') U_h(s', a) \right) = P_h(V_{h+1}^k - V_{h+1}^\pi)(s, a)$$

and

$$|\lambda F_h(s)^\top \left((\Lambda_{h,k}^a)^{-1}\right) \sum_{s' \in \mathcal{S}} (V_{h+1}^k - V_{h+1}^k)(s') U_h(s', a)\right)| \le 2H\sqrt{d\lambda}\sqrt{F_h(s)^\top (\Lambda_{h,k}^a)^{-1} F_h(s)}$$

Since

$$|F_h(s)^\top w_{h,k}^a - Q_h^\pi(s,a)| = F_h(s)^\top (w_{h,k}^a - w_h^{a,\pi}) = F_h(s)^\top (q_1^a + q_2^a + q_3^a),$$

it follows that

$$|F_h(s)^\top w_{h,k}^a - Q_h^\pi(s,a) - P_h(V_{h+1}^k - V_{h+1}^\pi)(s,a)| \le C''dH\sqrt{\chi}\sqrt{F_h(s)^\top (\Lambda_{h,k}^a)^{-1} F_h(s)}.$$

Similarly as in [17], we choose $c_\beta$ that satisfies $C'' \sqrt{\log(2) + \log(c_\beta + 1)} \le c_\beta \sqrt{\log(2)}$. $\qquad\square$

This lemma implies that by adding the appropriate bonus, $Q_h^k$ is always an upperbound of $Q_h^*$ with high probability, i.e., achieving optimism.

**Lemma 9.** *On the event $\mathcal{X}$ defined in lemma 7, we have $Q_h^k(s,a) \ge Q_h^*(s,a)$ for all $(s,a,h,k) \in \mathcal{S} \times \mathcal{A} \times [H] \times [K]$.*

*Proof.* We prove this with induction. The base case holds because at step $H+1$, the value function is zero, so from Lemma 8,

$$|F_H(s)^\top w_{H,k}^a - Q_H^*(s,a)| \le \beta_{k,h}^a \sqrt{F_H(s)^\top (\Lambda_{H,k}^a)^{-1}) F_H(s)}$$

and thus,

$$Q_H^*(s,a) \le \min\left(H, F_H(s)^\top w_{H,k}^a + \beta_{k,h}^a \sqrt{F_H(s)^\top (\Lambda_{H,k}^a)^{-1}) F_H(s)}\right) = Q_H^k(s,a).$$

From the inductive hypothesis (assuming that $Q_{h+1}^k(s,a) \ge Q_{h+1}^*(s,a)$), it follows that $P_h(V_{h+1}^k - V_{h+1}^*)(s,a) \ge 0$. From Lemma 8, $F_h(s)^\top w_{h,k}^a - Q_h^*(s,a) \le \beta_{k,h}^a \sqrt{F_h(s)^\top (\Lambda_{h,k}^a)^{-1}) F_h(s)}$. It follows that

$$Q_h^*(s,a) \le \min\left(H, F_h(s)^\top w_{h,k}^a + \beta_{k,h}^a \sqrt{F_h(s)^\top (\Lambda_{h,k}^a)^{-1}) F_h(s)}\right) = Q_h^k(s,a).$$

$\qquad\square$

We next show that Lemma 8 transforms to the recursive formula $\delta_h^k = V_h^k(s_h^k) - V_h^{\pi_k}(s_h^k)$, which is used in proving our regret bound.

**Lemma 10.** *Let $\delta_h^k = V_h^k(s_h^k) - V_h^{\pi_k}(s_h^k)$ and $\xi_{h+1}^k = \mathbb{E}[\delta_{h+1}^k | s_h^k, a_h^k] - \delta_{h+1}^k$. Then, on the event $\mathcal{X}$ defined in Lemma 7, we have for any $(h,k)$:*

$$\delta_h^k \le \delta_{h+1}^k + \xi_{h+1}^k + \beta_{k,h}^a \sqrt{F_h(s_h^k)^\top (\Lambda_{h,k}^{a_h^k})^{-1} F_h(s_h^k)}$$

*Proof.* From Lemma 8, we have for any $(s,a,h,k)$ that

$$Q_h^k(s,a) - Q_h^{\pi_k}(s,a) \le P_h(V_{h+1}^k - V_{h+1}^{\pi_k})(s,a) + \beta_{k,h}^a \sqrt{F_h(s)^\top (\Lambda_{h,k}^a)^{-1} F_h(s)}.$$

From the definition of $V^{\pi_k}$, we have

$$\delta_h^k = Q_h^k(s_h^k, a_h^k) - Q_h^{\pi_k}(s_h^k, a_h^k).$$

$\qquad\square$

Finally, we prove the main theorem, Theorem 8, which asserts that the regret bound of our algorithm is efficient with respect to $A$ and $T$.

*Proof.* Suppose that the Assumptions required in Theorem 8 hold. We condition on the event $\mathcal{X}$ from Lemma 7 with $p = \delta/2$ and use the notation for $\delta_h^k, \xi_h^k$ as in Lemma 10. From Lemmas 9 and 10, we have

$$Regret(K) = \sum_{k=1}^{K}(V_1^*(s_1^k) - V_1^{\pi_k}(s_1^k)) \leq \sum_{k=1}^{K} \delta_1^k \leq \sum_{k \in [K]} \sum_{h \in [H]} \xi_h^k + \beta_{k,h}^a \sum_{k \in [K]} \sum_{h \in [H]} \sqrt{F_h(s_h^k)^\top (\Lambda_{h,k}^{a_h^k})^{-1} F_h(s_h^k)}$$

Since the observations at episode $k$ are independent of the computed value function (this uses the trajectories from episodes 1 to $k - 1$, it follows $\{\xi_h^k\}$ is a martingale difference sequence with $|\xi_h^k| \leq 2H$ for all $(k, h)$. Thus, from the Azuma Hoeffding inequality, we have

$$\sum_{k \in [K], h \in H} \xi_h^k \leq \sqrt{2TH^2 \log(2/p)} \leq 2H\sqrt{T\iota}$$

with probability at least $1 - \delta/2$. For the second term, we note that the minimum eigenvalue of $\Lambda_{h,k}^{a_h^k}$ is at least one by construction and $\|F_h(s)\| \leq 1$. From the Elliptical Potential Lemma (Lemma D.2 [17]), we have for all $a \in \mathcal{A}$ and $h \in [H]$,

$$\sum_{k=1}^{K} F_h(s_k)^\top (\Lambda_{h,k}^a)^{-1} F_h(s_k) \leq 2\log(det(\Lambda_{h,k+1}^a)/det(\Lambda_{h,0}^a))$$

Furthermore, we have $\|\Lambda_{h,k+1}^a\| = \|\sum_{t \in T_{K,h}^a} F_h(s_h^t)F_h(s_h^t)^\top + \lambda I\| \leq \lambda + |T_{K,h}^a| \leq \lambda + K$. It follows that

$$\sum_{k=1}^{K} F_h(s_k)^\top (\Lambda_{h,k}^a)^{-1} F_h(s_k) \leq 2d\log(1 + K/\lambda) \leq 2d\iota$$

Next, by Cauchy-Schwarz and grouping the regret by each action, it follows that

$$\sum_{k \in [K]} \sum_{h \in [H]} \sqrt{F_h(s_h^k)^\top (\Lambda_{h,k}^{a_h^k})^{-1} F_h(s_h^k)} \leq \sum_{h \in [H]} \sqrt{K} \left[ \sum_{k \in [K]} F_h(s_h^k)^\top (\Lambda_{h,k}^{a_h^k})^{-1} F_h(s_h^k) \right]^{1/2}$$

$$= \sum_{h \in [H]} \sqrt{K} \left[ \sum_{a \in \mathcal{A}} \sum_{t \in T_{K,h}^a} F_h(s_h^t)^\top (\Lambda_{h,t}^a)^{-1} F_h(s_h^t) \right]^{1/2}$$

$$\leq \sum_{h \in [H]} \sqrt{K} \left[ \sum_{a \in \mathcal{A}} 2d\iota \right]^{1/2}$$

$$\leq H\sqrt{2KAd\iota}.$$

Since $\beta_{k,h}^a = cdH\sqrt{\iota}$ for some constant $c$, from a union bound and the previous bounds, it follows that

$$Regret(K) \leq 2H\sqrt{T\iota} + \beta_{k,h}^a H\sqrt{2KAd\iota} \in \tilde{O}(\sqrt{d^3 H^3 AT}).$$

with probability at least $1 - \delta$. $\qquad\square$

To use LSVI-UCB-$(S, d, A)$ in our transfer RL setting, we next show that it is robust to misspecification error. Similarly, we first prove the helper lemmas needed in the misspecified setting and follow the proof structure and techniques used in [17].

**Lemma 11.** *For a $\xi$-approximate $(S, d, A)$ Tucker rank MDP (Assumption 7), then for any policy $\pi$ there exists corresponding weight vectors $\{w_h^a\}$ where $w_h^a = W_h(a) + \sum_{s' \in \mathcal{S}} V_{h+1}^\pi(s') U_h(s', a)$ such that for all $(s, a) \in \mathcal{S} \times \mathcal{A}$*

$$|Q_h^\pi(s, a) - F_h(s)w_h^{a\top}| \leq 3H\xi.$$

*Proof.* Note that $Q_h^\pi(s, a) = r_h(s, a) + \sum_{s'} V_{h+1}^\pi(s') P_h(s'|s, a)$, so, using the low-rank representations of $r_h, P_h$, it follows that

$$|Q_h^\pi(s, a) - F_h(s) w_h^{a\top}| \leq |r_h(s, a) - F_h(s) W_h(a)^\top|$$

$$+ \left| \sum_{s'} V_{h+1}^\pi(s') P_h(s'|s, a) - F_h(s) \sum_{s' \in \mathcal{S}} V_{h+1}^\pi(s') U_h(s', a) \right|$$

$$\leq \xi + 2H\xi$$

$$\leq 3H\xi$$

$\square$

**Lemma 12.** *Suppose Assumption 7 holds. Let $\{w_h^a\}$ be the weight vector of some policy $\pi$, i.e., $Q_h^\pi(s, a) = F_h(s)^\top w_h^a$. Then,*

$$\|w_h^a\| \leq 2Hd.$$

*Proof.* Recall that $w_h^a = W_h(a) + \sum_{s' \in \mathcal{S}} V_{h+1}^\pi(s') U_h(s', a)$. Since the feature mapping was scaled up ($F_h = \sqrt{\frac{S}{d\mu}} F'$) to ensure the max entries of $F$ and $W, U$ are on the same scale, it follows that $\|W_h(a)\|_2 \leq 1$ and $\|\sum_{s' \in \mathcal{S}} V_{h+1}^\pi(s') U_h(s', a)\|_2 \leq H$. Thus, $\|w_h^a\|_2 \leq 2Hd$. $\square$

Similarly, we bound the stochastic noise but account for the misspecification error.

**Lemma 13.** *Suppose Assumption 7 holds. Let $c'_m$ be the constant in the definition of $\beta_{k,h}^a = c'_m H(d\sqrt{\iota} + \xi\sqrt{|T_{k,h}^a|d})$. Then, there exists a constant $C > 0$ that is independent of $c'_m$ such that for any $\delta \in (0, 1)$, if we let the event $\mathcal{X}$ be*

$$\forall (a, k, h) \in \mathcal{A} \times [K] \times [H]: \quad \left\| \sum_{t \in T_{h,k-1}^a} F_h(s_h^t)(V_{h+1}^t(s_{h+1}^t) - P_h V_{h+1}^t(s_h^t, a)) \right\|_{(\Lambda_{h,t}^a)^{-1}} \leq CdH\sqrt{\chi}$$

*where $\chi = \log(2(c'_m + 1)dTA/\delta)$, then $\mathbb{P}(\mathcal{X}) \geq 1 - \delta/2$.*

*Proof.* The proof is the same as the proof of Lemma 7 except we increase $\beta_{k,h}^a$ to account for the misspecifiction error. Since $\xi \leq 1$ by assumption, the modification to $\beta_{k,h}^a$ only affects $C$. $\square$

We next account for the noise being adversarial due to the misspecification error.

**Lemma 14.** *Let $\{\epsilon_t\}$ be any sequence such that $|\epsilon_\tau| \leq B$ for any t. Then, for any $(h, k, a) \in [H] \times [K] \times \mathcal{A}$ and $F \in \mathbb{R}^d$, we have*

$$|F^\top (\Lambda_{k,h}^a)^{-1} \sum_{t \in T_{k-1,h}^a} F_h(s_h^t) \epsilon_t| \leq B\sqrt{d|T_{k-1,h}^a| F^\top (\Lambda_{k,h}^a)^{-1} F}.$$

*Proof.* From the Cauchy-Schwarz inequality and Elliptical Potential Lemma (Lemma D.1 [17]), we have

$$|F^\top (\Lambda_{k,h}^a)^{-1} \sum_{t \in T_{k-1,h}^a} F_h(s_h^t) \epsilon_t| \leq B\sqrt{\left( \sum_{t \in T_{k-1,h}^a} F^\top (\Lambda_{k,h}^a)^{-1} F \right) \left( \sum_{t \in T_{k-1,h}^a} F_h(s_h^t)^\top (\Lambda_{k,h}^a)^{-1} F_h(s_h^t) \right)}$$

$$\leq B\sqrt{d|T_{k-1,h}^a| F^\top (\Lambda_{k,h}^a)^{-1} F}.$$

$\square$

Next, we bound the error between a policies $Q$ function and our low-rank estimate.

**Lemma 15.** *There exists a constant $c'_m$ such that $\beta^a_{k,h} = c'_m H(d\sqrt{\iota} + \xi\sqrt{|T^a_{k,h}|d})$ where $\iota = \log(2dAT/p)$ and for any fixed policy $\pi$, on the event $\mathcal{X}$ defined in Lemma 13, we have for all $(s,a,h,k) \in \mathcal{S} \times \mathcal{A} \times [H] \times [K]$:*

$$F_h(s)^\top w^a_{h,k} - Q^\pi_h(s,a) = P_h(V^k_{h+1} - V^\pi_{h+1})(s,a) + \Delta^k_h(s,a),$$

*where $|\Delta^k_h(s,a)| \leq \beta^a_{k,h}\sqrt{F_h(s)^\top(\Lambda^a_{h,k})^{-1}F_h(s)} + 4H\xi$.*

*Proof.* From Lemma 12, it follows that there exists a weight vector $w^a_h = W_h(a) + \sum_{s' \in \mathcal{S}} V^\pi_{h+1}(s')U_h(s',a)$, such that for all $(s,a) \in \mathcal{S} \times \mathcal{A}$,

$$|Q^\pi_h(s,a) - F_h(s)^\top w^a_h| \leq 2H\xi.$$

Let $\tilde{P}(\cdot|s,a) = F_h(s)^\top U_h(\cdot,a)$, so we have for any $(s,a) \in \mathcal{S} \times \mathcal{A}$, $F_h(s)^\top w^a_h = F_h(s)^\top W_h(a) + \tilde{P}V^\pi_{h+1}(s,a)$. Therefore,

$$
\begin{aligned}
w^a_{h,k} - w^{a,\pi}_h = {} & -\lambda(\Lambda^a_{h,k})^{-1}w^{a,\pi}_h \\
& + (\Lambda^a_{h,k})^{-1} \sum_{t \in T^a_{k-1,h}} F_h(s^t_h)(V^k_{h+1}(s^t_{h+1}) - P_h V^k_{h+1}(s^t_h,a)) \\
& + (\Lambda^a_{h,k})^{-1} \sum_{t \in T^a_{k-1,h}} F_h(s^t_h)\tilde{P}_h(V^k_{h+1} - V^\pi_{h+1})(s^t_h,a) \\
& + (\Lambda^a_{h,k})^{-1} \sum_{t \in T^a_{k-1,h}} F_h(s^t_h)\left(r_h(s^t_h,a) - F_h(s^t_h)W_h(a) + (P_h - \tilde{P}_h)V^k_{h+1}(s^t_h,a)\right)
\end{aligned}
$$

Now, we bound each of the four terms above $(q^a_1, q^a_2, q^a_3, q^a_4)$ individually, for the first term, note for all $a \in \mathcal{A}$

$$|F_h(s)^\top q^a_1| = |\lambda F_h(s)^\top(\Lambda^a_{h,k})^{-1}w^{a,\pi}_h| \leq \sqrt{\lambda}\|w^{a,\pi}_h\|\sqrt{F_h(s)^\top(\Lambda^a_{h,k})^{-1}F_h(s)}$$

For the second term, given the event $\mathcal{X}$, we have from Cauchy Schwarz

$$|F_h(s)^\top q^a_2| \leq CdH\sqrt{\chi}\sqrt{F_h(s)^\top(\Lambda^a_{h,k})^{-1}F_h(s)}$$

where $\chi = \log(2(c_\beta + 1)dTA/p)$. For the third term,

$$
\begin{aligned}
F_h(s)^\top q^a_3 &= F_h(s)^\top \left((\Lambda^a_{h,k})^{-1} \sum_{t \in T^a_{k-1,h}} F_h(s^t_h)\tilde{P}_h(V^k_{h+1} - V^\pi_{h+1})(s^t_h,a)\right) \\
&= F_h(s)^\top \left((\Lambda^a_{h,k})^{-1} \sum_{t \in T^a_{k-1,h}} F_h(s^t_h)F_h(s^t_h)^\top \sum_{s' \in \mathcal{S}}(V^k_{h+1} - V^\pi_{h+1})(s')U_h(s',a)\right) \\
&= F_h(s)^\top \left(\sum_{s' \in \mathcal{S}}(V^k_{h+1} - V^\pi_{h+1})(s')U_h(s',a)\right) \\
&\quad - \lambda F_h(s)^\top \left((\Lambda^a_{h,k})^{-1} \sum_{s' \in \mathcal{S}}(V^k_{h+1} - V^\pi_{h+1})(s')U_h(s',a)\right),
\end{aligned}
$$

and note that

$$F_h(s)^\top \left(\sum_{s' \in \mathcal{S}}(V^k_{h+1} - V^\pi_{h+1})(s')U_h(s',a)\right) = \tilde{P}_h(V^k_{h+1} - V^\pi_{h+1})(s,a)$$

and

$$\left|\lambda F_h(s)^\top \left((\Lambda^a_{h,k})^{-1} \sum_{s' \in \mathcal{S}}(V^k_{h+1} - V^k_{h+1})(s')U_h(s',a)\right)\right| \leq 2H\sqrt{d\lambda}\sqrt{F_h(s)^\top(\Lambda^a_{h,k})^{-1}F_h(s)}.$$

Since $\|\tilde{P}_h(\cdot|s,a) - P_h(\cdot|s,a)\|_\infty \le 1$ for all $(s,a) \in \mathcal{S} \times \mathcal{A}$, it follows that

$$|\tilde{P}_h(V_{h+1}^k - V_{h+1}^\pi)(s,a) - P_h(V_{h+1}^k - V_{h+1}^\pi)(s,a)| \le |(\tilde{P}_h - P_h)(V_{h+1}^k - V_{h+1}^\pi)(s,a)| \le 2H\xi.$$

From Lemma 14, we have $|F_h(s), q_4^a| \le 2H\xi\sqrt{d|T_{k,h}^a|F_h(s)^\top (\Lambda_{h,k}^{-1})^{-1}F_h(s)}$. Since

$$|F_h(s)^\top w_{h,k}^a - Q_h^\pi(s,a)| = F_h(s)^\top (w_{h,k}^a - w_h^{a,\pi}) = F_h(s)^\top (q_1^a + q_2^a + q_3^a + q_4^a),$$

it follows that

$$|F_h(s)^\top w_{h,k}^a - Q_h^\pi(s,a) - P_h(V_{h+1}^k - V_{h+1}^\pi)(s,a)| \le \sqrt{\chi}(C"d\sqrt{\chi} + 2\xi\sqrt{|T_{k,h}^a|d})H\sqrt{F_h(s)^\top (\Lambda_{h,k}^a)^{-1}F_h(s)} + 4H\xi.$$

Similarly as in [17], we choose $c_\beta$ that satisfies $C"\sqrt{\log(2) + \log(c_\beta + 1)} \le c_\beta\sqrt{\log(2)}$. $\qquad \square$

Now, we prove that $Q_h^k$ is an upperbound of $Q_h^*$ conditioned on the event in Lemma 13.

**Lemma 16.** *Suppose Assumption 7 holds. On the event $\mathcal{X}$ defined in lemma 13, we have $Q_h^k(s,a) \ge Q_h^*(s,a) - 4H(H+1-h)\xi$ for all $(s,a,h,k) \in \mathcal{S} \times \mathcal{A} \times [H] \times [K]$.*

*Proof.* We prove this with induction. The base case holds because at step $H+1$, the value function is zero, so from Lemma 15,

$$|F_H(s)^\top w_{H,k}^a - Q_H^*(s,a)| \le \beta_{k,h}^a\sqrt{F_H(s)^\top (\Lambda_{H,k}^a)^{-1})F_H(s)} + 4H\xi$$

and thus,

$$Q_H^*(s,a) - 4H\xi \le \min\left(H, F_H(s)^\top w_{H,k}^a + \beta_{k,h}^a\sqrt{F_H(s)^\top (\Lambda_{H,k}^a)^{-1})F_H(s)}\right) = Q_H^k(s,a).$$

From the inductive hypothesis (assuming that $Q_{h+1}^k(s,a) \ge Q_{h+1}^*(s,a) - 4H(H-h)\xi$), it follows that $P_h(V_{h+1}^k - V_{h+1}^*)(s,a) \ge -4\xi$. From Lemma 15, $F_h(s)^\top w_{h,k}^a - Q_h^*(s,a) \le \beta_{k,h}^a\sqrt{F_h(s)^\top (\Lambda_{h,k}^a)^{-1})F_h(s)} + 4H\xi$. It follows that

$$Q_h^*(s,a) - 4H(H-h+1)\xi \le \min\left(H, F_h(s)^\top w_{h,k}^a + \beta_{k,h}^a\sqrt{F_h(s)^\top (\Lambda_{h,k}^a)^{-1})F_h(s)}\right) = Q_h^k(s,a).$$

This completes the proof. $\qquad \square$

Similarly, to the regular case, the gap in the misspecified setting has a recursive formula.

**Lemma 17.** *Let $\delta_h^k = V_h^k(s_h^k) - V_h^{\pi_k}(s_h^k)$ and $\xi_{h+1}^k = \mathbb{E}[\delta_{h+1}^k|s_h^k, a_h^k] - \delta_{h+1}^k$. Then, on the event $\mathcal{X}$ defined in Lemma 13, we have for any $(h,k)$:*

$$\delta_h^k \le \delta_{h+1}^k + \xi_{h+1}^k + \beta_{k,h}^a\sqrt{F_h(s_h^k)^\top (\Lambda_{h,k}^{a_h^k})^{-1}F_h(s_h^k)} + 4H\xi$$

*Proof.* From Lemma 15, we have for any $(s,a,h,k)$ that

$$Q_h^k(s,a) - Q_h^{\pi_k}(s,a) \le P_h(V_{h+1}^k - V_{h+1}^{\pi_k})(s,a) + \beta_{k,h}^a\sqrt{F_h(s)^\top (\Lambda_{h,k}^a)^{-1}F_h(s)} + 4H\xi.$$

From the definition of $V^{\pi_k}$, we have

$$\delta_h^k = Q_h^k(s_h^k, a_h^k) - Q_h^{\pi_k}(s_h^k, a_h^k),$$

and substituting this into the first equation finishes the proof. $\qquad \square$

Finally, we prove the main result in the misspecified setting Theorem 9.

*Proof.* Suppose that the Assumptions required in Theorem 9 hold. We condition on the event $\mathcal{X}$ from Lemma 13 with $p = \delta/2$ and use the notation for $\delta_h^k, \xi_h^k$ as in Lemma 17. From Lemma 16, we have $Q_1^k(s,a) \geq Q_1^*(s,a) - 4H^2\xi$, which implies $V_1^*(s) - V_1^{\pi_k}(s) \leq \delta_1^k + 4H^2\xi$. Thus, from Lemma 15, on the event $\mathcal{X}$, it follows that

$$Regret(K) = \sum_{k=1}^{K}(V_1^*(s_1^k) - V_1^{\pi_k}(s_1^k)) \leq \sum_{k=1}^{K}\delta_1^k + 4H^2\xi$$

$$\leq \sum_{k\in[K]}\sum_{h\in[H]}\xi_h^k + \sum_{k\in[K]}\beta_{k,h}^a\sum_{h\in[H]}\sqrt{F_h(s_h^k)^\top(\Lambda_{h,k}^{a_h^k})^{-1}F_h(s_h^k)} + 4HT\xi$$

since $HK = T$. Since the observations at episode $k$ are independent of the computed value function (this uses the trajectories from episodes 1 to $k-1$, it follows $\{\xi_h^k\}$ is a martingale difference sequence with $|\xi_h^k| \leq 2H$ for all $(k,h)$. Thus, from the Azuma Hoeffding inequality, we have

$$\sum_{k\in[K],h\in H}\xi_h^k \leq \sqrt{2TH^2\log(2/p)} \leq 2H\sqrt{T\iota}$$

with probability at least $1 - \delta/2$. By construction of $\beta_{k,h}^a$, we have

$$\sum_{k\in[K]}\beta_{k,h}^a\sqrt{F_h(s_h^k)^\top(\Lambda_{h,k}^{a_h^k})^{-1}F_h(s_h^k)} \leq CH\Big(\sum_{k\in[K]}d\sqrt{\iota}\sqrt{F_h(s_h^k)^\top(\Lambda_{h,k}^{a_h^k})^{-1}F_h(s_h^k)}$$

$$+ \sum_{k\in[K]}\xi\sqrt{|T_{k,h}^a|d}\sqrt{F_h(s_h^k)^\top(\Lambda_{h,k}^{a_h^k})^{-1}F_h(s_h^k)}\Big).$$

From the Cauchy-Schwarz inequality, it follows that

$$\sum_{k\in[K]}\sqrt{F_h(s_h^k)^\top(\Lambda_{h,k}^{a_h^k})^{-1}F_h(s_h^k)} \leq \left(\sqrt{K}\right)\left(\sqrt{\sum_{k\in[K]}F_h(s_h^k)^\top(\Lambda_{h,k}^{a_h^k})^{-1}F_h(s_h^k)}\right).$$

Since the minimum eigenvalue of $\Lambda_{h,k}^{a_h^k}$ is at least one by construction and $\|F_h(s)\| \leq 1$, from the Elliptical Potential Lemma (Lemma D.2 [17]), we have for all $a \in \mathcal{A}$ and $h \in [H]$,

$$\sum_{k=1}^{K}F_h(s_k)^\top(\Lambda_{h,k}^a)^{-1}F_h(s_k) \leq 2\log(det(\Lambda_{h,k+1}^a)/det(\Lambda_{h,0}^a))$$

Furthermore, we have $\|\Lambda_{h,k+1}^a\| = \|\sum_{t\in T_{K,h}^a}F_h(s_h^t)F_h(s_h^t)^\top + \lambda I\| \leq \lambda + |T_{K,h}^a| \leq \lambda + K$. It follows that

$$\sum_{k=1}^{K}F_h(s_k)^\top(\Lambda_{h,k}^a)^{-1}F_h(s_k) \leq 2d\log(1 + K/\lambda) \leq 2d\iota,$$

so grouping the episodes by actions gives

$$\left[\sum_{k\in[K]}F_h(s_h^k)^\top(\Lambda_{h,k}^{a_h^k})^{-1}F_h(s_h^k)\right]^{1/2} = \left[\sum_{a\in\mathcal{A}}\sum_{t\in T_{K,h,a}}F_h(s_h^t)^\top(\Lambda_{h,t}^a)^{-1}F_h(s_h^t)\right]^{1/2} \leq \sqrt{2dA\iota}.$$

For the next term, let $k_{a,i}$ refer to the episode in which action $a$ was taken at time step $h$ for the $i$th time. Then, we first re-index the summation and then use the Cauchy-Schwarz inequality to get

$$\sum_{k\in[K]} \sqrt{|T_{k,h}^a|}\sqrt{F_h(s_h^k)^\top(\Lambda_{h,k}^{a_h^k})^{-1}F_h(s_h^k))} = \sum_{a\in\mathcal{A}}\sum_{i=1}^{|T_{K,h}^a|}\sqrt{i}\sqrt{F_{k_{a,i},h}^\top(\Lambda_{k_{a,i},h}^a)^{-1}F_{k_{a,i},h}}$$

$$\leq \sum_{a\in\mathcal{A}}\left(\sum_{i=1}^{|T_{K,h}^a|}i\right)^{1/2}\left(\sum_{t\in T_{K,h}^a}F_{t,h}^\top(\Lambda_{t,h}^a)^{-1}F_{t,h}\right)^{1/2}$$

$$\leq C'\sum_{a\in\mathcal{A}}|T_{K,h}^a|\left(\sum_{t\in T_{K,h}^a}F_{t,h}^\top(\Lambda_{t,h}^a)^{-1}F_{t,h}\right)^{1/2}$$

$$\leq C'\sum_{a\in\mathcal{A}}|T_{K,h}^a|\sqrt{2d\iota},$$

where the last inequality comes from holds from the above elliptical potential lemma. Applying the results to the initial inequality, we have

$$\sum_{k\in[K]}\beta_{k,h}^a\sqrt{F_h(s_h^k)^\top(\Lambda_{h,k}^{a_h^k})^{-1}F_h(s_h^k)} \leq CH(\sqrt{2d^3AK\iota^2}+\xi\sqrt{d}C'\sum_{a\in\mathcal{A}}|T_{K,h}^a|\sqrt{2d\iota}) \leq C(\sqrt{2d^3AHT\iota^2}+\xi\sqrt{2\iota}dT.$$

Substituting these results into the original regret bound gives

$$Regret(K) \leq C'''(\sqrt{d^3H^3AT\iota^2} + dTH\xi\sqrt{\iota}).$$

for some constant $C''' > 0$ with probability at least $1 - \delta$. $\qquad\square$

## J.2  LSVI-UCB-(S, S, d)

We reiterate that the steps and analysis in this subsection are essentially the same as the proofs in [17], except we have $S$ copies of the weight vectors and gram matrices. The results and proofs follow the same structure and logic as in The proof of all the theorems and lemmas follow the same steps as the proofs for the analogous result in the $(S, d, A)$ setting except we replace $F(s)$ with $G(a)$ and estimate the Gram matrix and weight vectors for each state instead of each action.

To prove Theorem 5, we first introduce notation used throughout this section and then present auxiliary lemmas. We let $\Lambda_{h,k}^s, w_{h,k}^s, Q_h^k, V_h^k, \pi^k$ as the parameters/estimates at episode $k$. Furthermore, let $V_h^k = \max_{s\in\mathcal{S}}Q_h^k(s,a)$ and $\pi^k$ be the greedy policy w.r.t. $Q_h^k$. We let $G_h^k := G_h(a_h^k)$.

We first bound the log of the covering number of our function class for the value function to control the error of our algorithm's estimate, which allows us to prove our concentration result as the value function estimate is not constant.

**Lemma 18.** *Let $\mathcal{V}$ denote a function class of mappings from $\mathcal{S}$ to $\mathbb{R}$ with the following parametric form for all $s \in \mathcal{S}$*

$$V(s) = \min\left\{\max_{a\in\mathcal{A}}w_s^\top G(a) + \beta\sqrt{G(a)^\top\Lambda_s^{-1}G(a)}, H\right\}$$

*where the parameters $w_s, \beta, \Lambda_s$ satisfy $\|w_s\| \leq L$ for all $s \in \mathcal{S}$, $\beta \in [0, B]$, and the minimum eigenvalue of $\Lambda_s$ is greater than $\lambda$. Assume that $\|G(a)\| \leq 1$ for all $s \in \mathcal{S}$, and let $\mathcal{N}_\epsilon$ be the $\epsilon$-covering number of $\mathcal{V}$ with respect to the distance $dist(V, V') = \sup_s|V(s) - V'(s)|$. Then,*

$$\log(\mathcal{N}_\epsilon) \leq d\log(1 + 4L/\epsilon) + d^2\log(1 + 8d^{1/2}B^2/(\lambda\epsilon^2))$$

*Proof.* Equivalently, we reparameterize the function class by $D_s = \beta^2\Lambda_s^{-1}$, for all $s \in \mathcal{S}$

$$V(s) = \min\left\{\max_{a\in\mathcal{A}}w_s^\top G(a) + \sqrt{G(a)^\top D_sG(a)}, H\right\}$$

where $\|w_s\| \leq L$ and $\|D_s\| \leq B^2/\lambda$ for all $s \in \mathcal{S}$. For any $V_1, V_2 \in \mathcal{V}$, let them take the form in the above equation with parameters $(w_s, D_s)$ and $(w_{s'}, D_{s'})$. Since $\min(\cdot, H)$ and $\max_{s \in \mathcal{S}}$ are contraction maps,

$$dist(V_1 - V_2) \leq \sup_{s,a} |(w_s^\top G(a) + \sqrt{G(a)^\top D_s G(a)}) - (w_{s'}^\top G(a) + \sqrt{G(a)^\top D_{s'} G(a)})|$$

$$\leq \sup_{s, G:\|G\|\leq 1} |(w_s^\top G + \sqrt{G^\top D_s G}) - (w_{s'}^\top G + \sqrt{G^\top D_{s'} G})|$$

$$\leq \max_{s \in \mathcal{S}} \left[ \sup_{G:\|G\|\leq 1} |(w_s - w_{s'})^\top G| + \sup_{G:\|G\|\leq 1} |\sqrt{G^\top(D_s - D_{s'})G}| \right]$$

$$\leq \max_{s \in \mathcal{S}} \left[ \|w_s - w_{s'}\| + \sqrt{\|D_s - D_{s'}\|} \right] \leq \max_{s \in \mathcal{S}} \left[ \|w_s - w_{s'}\| + \sqrt{\|D_s - D_{s'}\|_F} \right]$$

where the third inequality holds because $|\sqrt{x} - \sqrt{y}| \leq \sqrt{|x - y|}$ for any $x, y \geq 0$. Let $C_w$ be an $\epsilon/2$-cover of $\{w \in \mathbb{R}^d | \|w\| \leq L\}$ with respect to the 2-norm, and $C_D$ be an $\epsilon^2/4$-cover of $\{D \in \mathbb{R}^{d \times d} | \|D\|_F \leq d^{1/2} B^2/\lambda\}$ with respect to the Frobenius norm. By Lemma D.5 from [17], we know that,

$$|C_w| \leq (1 + 4L/\epsilon)^d, \quad |C_D| \leq (1 + 8d^{1/2} B^2/(\lambda \epsilon^2))^{d^2}.$$

Hence, for any $V_1$, there exists a $w_{s'} \in C_w$ and $D_{s'} \in C_D$ for all $a \in \mathcal{A}$ such that $V_2$ parameterized by $(w_{s'}, D_{s'})$ satisfies $dist(V_1, V_2) \leq \epsilon$. Hence, $\mathcal{N}_\epsilon \leq |C_w||C_D|$, which gives:

$$\log(\mathcal{N}_\epsilon) \leq \log(|C_w|) + \log(|C_D|) \leq d \log(1 + 4\epsilon L/\epsilon) + d^2 \log(1 + 8d^{1/2} B^2/(\lambda \epsilon^2)).$$

$\square$

We next bound the $\ell_2$-norm of the weight vector of any policy.

**Lemma 19.** *Let $\{w_h^s\}$ be the weight vector of some policy $\pi$, i.e., $Q_h^\pi(s, a) = G_h(a)^\top w_h^s$. Then,*

$$\|w_h^s\| \leq 2H\sqrt{d}.$$

*Proof.* Note that $Q_h^\pi(s, a) = r_h(s, a) + \sum_{s'} V_{h+1}^\pi(s') P_h(s'|s, a)$, so, using the low-rank representations of $r_h, P_h$, it follows that

$$\|w_h^s\|_2 = \|W_h(s, \cdot) + \sum_{s'} V^\pi(s') U_h(s', s, \cdot)\|_2 \leq \|W_h(s, \cdot)\|_2 + \|\sum_{s'} V^\pi(s') U_h(s', s, \cdot)\|_2.$$

Recall that since the feature mapping was scaled up ($G_h = \sqrt{\frac{A}{d\mu}} G'$) to ensure the max entries of $G$ and $W, U$ are on the same scale, it follows that $\|W_h(s)\|_2 \leq 1$ and $\|\sum_{s' \in \mathcal{S}} V_{h+1}^\pi(s') U_h(s', s)\|_2 \leq H$. Therefore, it follows that $\|w_h^s\|_2 \leq 2H\sqrt{d}$. $\square$

We next bound the 2-norm of the weight vector from the algorithm.

**Lemma 20.** *For any $k, h, s$, the weight vector from the above algorithm satisfies $\|w_{h,k}^s\| \leq 2H\sqrt{dk/\lambda}$.*

*Proof.* For any $G_h(a) = g$, we have for all $s \in \mathcal{S}$,

$$|g^\top w_{h,k}^s| = |g^\top (\Lambda_h^s)^{-1} \sum_{t \in T_{k-1,h}^s} G_h(a_h^t) \left[ r_h(s, a_h^t) + \max_{a' \in \mathcal{A}} Q_{h+1}(s_{h+1}^t, a') \right] |$$

$$\leq |\sum_{t \in T_{k-1,h}^s} g^\top (\Lambda_h^s)^{-1} G_h(a_h^t)| 2H$$

$$\leq 2H\sqrt{(\sum_{t \in T_{k-1,h}^s} g^\top (\Lambda_h^s)^{-1} g)(\sum_{t \in T_{k-1,h}^s} G_h(a_h^t)^\top (\Lambda_h^s)^{-1} G_h(a_h^t))}$$

$$\leq 2H\|g\|_2 \sqrt{d|T_{k-1,h}^s|/\lambda} \leq 2H\sqrt{dk/\lambda}$$

where the second to last inequality is due to Lemma D.1 from [17] and that Rayleigh quotients are upper bounded by the largest eigenvalue of the matrix (Theorem 4.2.2 from [14]). The last inequality is due to $\|g\|_2 \leq 1$ and $|T^s_{k-1,h}| \leq k$. $\qquad\square$

We next prove the main concentration lemma that controls the estimate.

**Lemma 21.** *Let $c'$ be the constant in the definition of $\beta^s_{k,h} = c'dH\sqrt{\iota}$. Then, there exists a constant $C > 0$ that is independent of $c'$ such that for any $\delta \in (0,1)$, if we let the event $\mathcal{X}$ be*

$$\forall (s,k,h) \in \mathcal{S} \times [K] \times [H]: \quad \left\| \sum_{t \in T^s_{k-1,h}} G_h(a^t_h)(V^t_{h+1}(s^t_{h+1}) - P_h V^t_{h+1}(s,a^t_h)) \right\|_{(\Lambda^s_{h,t})^{-1}} \leq CdH\sqrt{\chi}$$

*where $\chi = \log(2(c'+1)dTS/\delta)$, then $\mathbb{P}(\mathcal{X}) \geq 1 - \delta/2$.*

*Proof.* From Lemma 20, we have $\|w^s_{h,k}\|_2 \leq 2H\sqrt{dk/\lambda}$. By construction, the minimum eigenvalue of $\Lambda^s_{h,t}$ is lower bounded by $\lambda$. Then, from Lemma D.4 in [17] (with $\delta' = \delta/S$), Lemma 18, and taking a union bound over all states, we have that with probability at least $1 - \delta'/2$, for all $s \in \mathcal{S}$ and $\epsilon > 0$,

$$\left\| \sum_{t \in T^s_{k-1,h}} G_h(a^t_h)(V^t_{h+1}(s^t_{h+1}) - P_h V^t_{h+1}(s,a^t_h)) \right\|^2_{(\Lambda^s_{h,t})^{-1}} \leq 4H^2 \left[ \frac{d}{2} \log((k+\lambda)/\lambda)) + \log(2S\mathcal{N}_\epsilon/\delta) \right] + 8k^2\epsilon^2/\lambda.$$

Then, from Lemma 18, it follows that with probability at least $1 - \delta'/2$, for all $a \in S$ and $\epsilon > 0$,

$$\left\| \sum_{t \in T^s_{k-1,h}} G_h(a^t_h)(V^t_{h+1}(s^t_{h+1}) - P_h V^t_{h+1}(s,a^t_h)) \right\|^2_{(\Lambda^s_{h,t})^{-1}}$$

$$\leq 4H^2 \left[ \frac{d}{2} \log((k+\lambda)/\lambda)) + \log(2S/\delta) + d\log(1 + 4L/\epsilon) + d^2 \log(1 + 8d^{1/2}\beta^2/(\lambda\epsilon^2)) \right] + 8k^2\epsilon^2/\lambda.$$

Choosing $\lambda = 1, \beta^s_{k,h} = CdH\iota, \epsilon = dH/k$ gives that there exists some constant $C'$ such that

$$\left\| \sum_{t \in T^s_{k-1,h}} G_h(a^t_h)(V^t_{h+1}(s^t_{h+1}) - P_h V^t_{h+1}(s,a^t_h)) \right\|^2_{(\Lambda^s_{h,t})^{-1}} \leq C'd^2H^2 \log(2(c_\beta+1)dTS/\delta).$$

$\qquad\square$

We next recursively bound the difference between the value function maintained in the algorithm and the true value function of any policy $\pi$. We upperbound this with their expected difference and an additional error term, which is bounded with high probability.

**Lemma 22.** *There exists a constant $c_\beta$ such that $\beta^s_{k,h} = c_\beta dH\sqrt{\iota}$ where $\iota = \log(2dST/p)$ and for any fixed policy $\pi$, on the event $\mathcal{X}$ defined in Lemma 21, we have for all $(s,a,h,k) \in S \times S \times [H] \times [K]$:*

$$G_h(a)^\top w^s_{h,k} - Q^\pi_h(s,a) = P_h(V^k_{h+1} - V^\pi_{h+1})(s,a) + \delta^k_h(s,a),$$

*where $|\delta^k_h(s,a)| \leq \beta^s_{k,h} \sqrt{G_h(a)^\top (\Lambda^s_{h,k})^{-1} G_h(a)}$.*

*Proof.* From our low-rank assumption, it follows that

$$Q^\pi_h(s,a) = G_h(a)^\top w^s_h = (r_h + P_h V^\pi_{h+1})(s,a),$$

which following the steps from the proof of Lemma B.4 in [17] gives

$$w_{h,k}^s - w_h^{s,\pi} = -\lambda(\Lambda_{h,k}^s)^{-1}w_h^{s,\pi}$$
$$+ (\Lambda_{h,k}^s)^{-1} \sum_{t \in T_{k-1,h}^s} G_h(a_h^t)(V_{h+1}^k(s_{h+1}^t) - P_h V_{h+1}^k(s, a_h^t))$$
$$+ (\Lambda_{h,k}^s)^{-1}) \sum_{t \in T_{k-1,h}^s} G_h(a_h^t) P_h(V_{h+1}^k - V_{h+1}^\pi)(s, a_h^t)$$

Now, we bound each term $(q_1^s, q_2^s, q_3^s)$ individually, for the first term, note for all $s \in S, a \in \mathcal{A}$

$$|G_h(a)^\top q_1^s| = |\lambda G_h(a)^\top(\Lambda_{h,k}^s)^{-1}w_h^{s,\pi}| \leq \sqrt{\lambda}\|w_h^{s,\pi}\|\sqrt{G_h(a)^\top(\Lambda_{h,k}^s)^{-1}G_h(a)}$$

For the second term, given the event $\mathcal{X}$, we have from Cauchy Schwarz

$$|G_h(a)^\top q_2^s| \leq CdH\sqrt{\chi}\sqrt{G_h(a)^\top(\Lambda_{h,k}^s)^{-1}G_h(a)}$$

where $\chi = \log(2(c_\beta + 1)dTS/p)$. For the third term,

$$G_h(a)^\top q_3^s = G_h(a)^\top \left( (\Lambda_{h,k}^s)^{-1}) \sum_{t \in T_{k-1,h}^s} G_h(a_h^t) P_h(V_{h+1}^k - V_{h+1}^\pi)(s, a_h^t) \right)$$

$$= G_h(a)^\top \left( (\Lambda_{h,k}^s)^{-1}) \sum_{t \in T_{k-1,h}^s} G_h(a_h^t) G_h(a_h^t)^\top \sum_{s' \in S} (V_{h+1}^k - V_{h+1}^\pi)(s') U_h(s', s) \right)$$

$$= G_h(a)^\top \left( \sum_{s' \in S} (V_{h+1}^k - V_{h+1}^\pi)(s') U_h(s', s) \right)$$

$$- \lambda G_h(a)^\top \left( (\Lambda_{h,k}^s)^{-1}) \sum_{s' \in S} (V_{h+1}^k - V_{h+1}^\pi)(s') U_h(s', s) \right)$$

We note that

$$G_h(a)^\top \left( \sum_{s' \in S} (V_{h+1}^k - V_{h+1}^\pi)(s') U_h(s', s) \right) = P_h(V_{h+1}^k - V_{h+1}^\pi)(s, a)$$

and

$$\left|\lambda G_h(a)^\top \left( (\Lambda_{h,k}^s)^{-1}) \sum_{s' \in S} (V_{h+1}^k - V_{h+1}^k)(s') U_h(s', s) \right)\right| \leq 2H\sqrt{d\lambda}\sqrt{G_h(a)^\top(\Lambda_{h,k}^s)^{-1}G_h(a)}$$

Since

$$|G_h(a)^\top w_{h,k}^s - Q_h^\pi(s, a)| = G_h(a)^\top(w_{h,k}^s - w_h^{s,\pi}) = G_h(a)^\top(q_1^s + q_2^s + q_3^s),$$

it follows that

$$|G_h(a)^\top w_{h,k}^s - Q_h^\pi(s, a) - P_h(V_{h+1}^k - V_{h+1}^\pi)(s, a)| \leq C"dH\sqrt{\chi}\sqrt{G_h(a)^\top(\Lambda_{h,k}^s)^{-1}G_h(a)}.$$

Similarly as in [17], we choose $c_\beta$ that satisfies $C"\sqrt{\log(2) + \log(c_\beta + 1)} \leq c_\beta\sqrt{\log(2)}$.     $\square$

This lemma implies that by adding the appropriate bonus, $Q_h^k$ is always an upperbound of $Q_h^*$ with high probability

**Lemma 23.** *On the event $\mathcal{X}$ defined in lemma 21, we have $Q_h^k(s, a) \geq Q_h^*(s, a)$ for all $(s, a, h, k) \in S \times \mathcal{A} \times [H] \times [K]$.*

*Proof.* We prove this with induction. The base case holds because at step $H + 1$, the value function is zero, so from Lemma 22,

$$|G_h(a)^\top w_{h,k}^s - Q_H^*(s,a)| \leq \beta_{k,h}^s \sqrt{G_h(a)^\top (\Lambda_{h,k}^s)^{-1}) G_h(a)}$$

and thus,

$$Q_H^*(s,a) \leq \min\left(H, G_h(a)^\top w_{h,k}^s + \beta_{k,h}^s \sqrt{G_h(a)^\top (\Lambda_{h,k}^s)^{-1}) G_h(a)}\right) = Q_H^k(s,a).$$

From the inductive hypothesis (assuming that $Q_{h+1}^k(s,a) \geq Q_{h+1}^*(s,a)$), it follows that $P_h(V_{h+1}^k - V_{h+1}^*)(s,a) \geq 0$. From Lemma 22, $G_h(a)^\top w_{h,k}^s - Q_h^*(s,a) \leq \beta_{k,h}^s \sqrt{G_h(a)^\top (\Lambda_{h,k}^s)^{-1} G_h(a)}$. It follows that

$$Q_h^*(s,a) \leq \min\left(H, G_h(a)^\top w_{h,k}^s + \beta_{k,h}^s \sqrt{G_h(a)^\top (\Lambda_{h,k}^s)^{-1}) G_h(a)}\right) = Q_h^k(s,a).$$

$\square$

We next show that Lemma 22 transforms to the recursive formula $\delta_h^k = V_h^k(s_h^k) - V_h^{\pi_k}(s_h^k)$.

**Lemma 24.** *Let $\delta_h^k = V_h^k(s_h^k) - V_h^{\pi_k}(s_h^k)$ and $\xi_{h+1}^k = \mathbb{E}[\delta_{h+1}^k | s_h^k, s_h^k] - \delta_{h+1}^k$. Then, on the event $\mathcal{X}$ defined in Lemma 21, we have for any $(h,k)$:*

$$\delta_h^k \leq \delta_{h+1}^k + \xi_{h+1}^k + \beta_{k,h}^s \sqrt{G_h(a_h^k)^\top (\Lambda_{h,k}^{s_h^k})^{-1} G_h(a_h^k)}$$

*Proof.* From Lemma 22, we have for any $(s,a,h,k)$ that

$$Q_h^k(s,a) - Q_h^{\pi_k}(s,a) \leq P_h(V_{h+1}^k - V_{h+1}^{\pi_k})(s,a) + \beta_{k,h}^s \sqrt{G_h(a)^\top (\Lambda_{h,k}^s)^{-1} G_h(a)}.$$

From the definition of $V^{\pi_k}$, we have

$$\delta_h^k = Q_h^k(s_h^k, a_h^k) - Q_h^{\pi_k}(s_h^k, a_h^k).$$

$\square$

Finally, we prove the main theorem, Theorem 5.

*Proof.* Suppose that the Assumptions required in Theorem 5 hold. We condition on the event $\mathcal{X}$ from Lemma 21 with $p = \delta/2$ and use the notation for $\delta_h^k, \xi_h^k$ as in Lemma 24. From Lemmas 9 and 24, we have

$$Regret(K) = \sum_{k=1}^K (V_1^*(s_1^k) - V_1^{\pi_k}(s_1^k)) \leq \sum_{k=1}^K \delta_1^k \leq \sum_{k \in [K]} \sum_{h \in [H]} \xi_h^k + \beta_{k,h}^s \sum_{k \in [K]} \sum_{h \in [H]} \sqrt{G_h(a_h^k)^\top (\Lambda_{h,k}^{s_h^k})^{-1} G_h(a_h^k)}$$

Since the observations at episode $k$ are independent of the computed value function (this uses the trajectories from episodes 1 to $k - 1$, it follows $\{\xi_h^k\}$ is a martingale difference sequence with $|\xi_h^k| \leq 2H$ for all $(k,h)$. Thus, from the Azuma Hoeffding inequality, we have

$$\sum_{k \in [K], h \in H} \xi_h^k \leq \sqrt{2TH^2 \log(2/p)} \leq 2H\sqrt{T\iota}$$

with probability at least $1 - \delta/2$. For the second term, we note that the minimum eigenvalue of $\Lambda_{h,k}^{s_h^k}$ is at least one by construction and $\|G_h(a)\| \leq 1$. From the Elliptical Potential Lemma (Lemma D.2 [17]), we have for all $s \in \mathcal{S}$ and $h \in [H]$,

$$\sum_{k=1}^K G_h(a_h^k)^\top (\Lambda_{h,k}^s)^{-1} G_h(a_h^k) \leq 2\log(det(\Lambda_{h,k+1}^s)/\det(\Lambda_{h,0}^s))$$

Furthermore, we have $\|\Lambda_{h,k+1}^s\| = \|\sum_{t\in T_{K,h}^s} G_h(a_h^t)G_h(a_h^t)^\top + \lambda I\| \le \lambda + |T_{K,h}^s| \le \lambda + K$. It follows that

$$\sum_{k=1}^K G_h(a_h^k)^\top (\Lambda_{h,k}^s)^{-1} G_h(a_h^k) \le 2d\log(1 + K/\lambda) \le 2d\iota.$$

Next, by Cauchy-Schwarz and grouping the regret by each state, it follows that

$$\sum_{k\in[K]}\sum_{h\in[H]} \sqrt{G_h(a_h^k)^\top (\Lambda_{h,k}^{s_h^k})^{-1} G_h(a_h^k)} \le \sum_{h\in[H]} \sqrt{K}\left[\sum_{k\in[K]} G_h(a_h^k)^\top (\Lambda_{h,k}^{s_h^k})^{-1} G_h(a_h^k)\right]^{1/2}$$

$$= \sum_{h\in[H]} \sqrt{K}\left[\sum_{s\in\mathcal{S}}\sum_{t\in T_{K,h}^s} G_h(a_h^t)^\top (\Lambda_{h,t}^s)^{-1} G_h(a_h^t)\right]^{1/2}$$

$$\le \sum_{h\in[H]} \sqrt{K}\left[\sum_{s\in\mathcal{S}} 2d\iota\right]^{1/2}$$

$$\le H\sqrt{2KSd\iota}.$$

Since $\beta_{k,h}^s = cdH\sqrt{\iota}$ for some constant $c$, from a union bound and the previous bounds, it follows that

$$Regret(K) \le 2H\sqrt{T\iota} + \beta_{k,h}^s H\sqrt{2KSd\iota} \in \tilde{O}(\sqrt{d^3H^3ST}).$$

with probability at least $1 - \delta$. $\qquad\square$

Next, we prove the helper lemmas needed in the misspecified setting and follow the proof structure and techniques used in [17].

**Lemma 25.** *For a $\xi$-approximate $(S, S, d)$ Tucker rank MDP (Assumption 4), then for any policy $\pi$ there exists corresponding weight vectors $\{w_h^s\}$ where $w_h^s = \Sigma_h W_h(s) + \sum_{s'\in\mathcal{S}} V_{h+1}^\pi(s')U_h(s', s)$ such that for all $(s, a) \in \mathcal{S} \times \mathcal{A}$*

$$|Q_h^\pi(s, a) - G_h(a)w_h^{s\top}| \le 3H\xi.$$

*Proof.* Note that $Q_h^\pi(s, a) = r_h(s, a) + \sum_{s'} V_{h+1}^\pi(s')P_h(s'|s, a)$, so, using the low-rank representations of $r_h, P_h$, it follows that

$$|Q_h^\pi(s, a) - G_h(a)w_h^{s\top}|$$

$$\le |r_h(s, a) - G_h(a)W_h(s)^\top| + \left|\sum_{s'} V_{h+1}^\pi(s')P_h(s'|s, a) - G_h(a)\sum_{s'\in\mathcal{S}} V_{h+1}^\pi(s')U_h(s', s)\right|$$

$$\le \xi + 2H\xi$$

$$\le 3H\xi.$$

$\qquad\square$

**Lemma 26.** *Suppose Assumption 7 holds. Let $\{w_h^s\}$ be the weight vector of some policy $\pi$, i.e., $Q_h^\pi(s, a) = G_h(a)^\top w_h^s$. Then,*

$$\|w_h^s\| \le 2H\sqrt{d}.$$

*Proof.* Recall that $w_h^s = \Sigma_h W_h(s) + \sum_{s'\in\mathcal{S}} V_{h+1}^\pi(s')U_h(s', s)$. From the same argument used to prove Lemma 19, it follows that $\|w_h^s\|_2 \le 2H\sqrt{d}$ $\qquad\square$

Similarly, we bound the stochastic noise but account for the misspecification error.

**Lemma 27.** *Suppose Assumption 7 holds. Let $c'_m$ be the constant in the definition of $\beta^s_{k,h} = c'_m H(d\sqrt{\iota} + \xi\sqrt{|T^s_{k,m}|d})$. Then, there exists a constant $C > 0$ that is independent of $c'_m$ such that for any $\delta \in (0, 1)$, if we let the event $\mathcal{X}$ be*

$$\forall (a, k, h) \in \mathcal{A} \times [K] \times [H]: \quad \left\| \sum_{t \in T^s_{k-1,h}} G_h(a^t_h)(V^t_{h+1}(s^t_{h+1}) - P_h V^t_{h+1}(s, a^t_h)) \right\|_{(\Lambda^s_{h,t})^{-1}} \leq CdH\sqrt{\chi}$$

*where $\chi = \log(2(c'_m + 1)dTS/\delta)$, then $\mathbb{P}(\mathcal{X}) \geq 1 - \delta/2$.*

*Proof.* The proof is the same as the proof of Lemma 21 except we increase $\beta^s_{k,h}$ to account for the misspecifiction error. Since $\xi \leq 1$ by assumption, the modification to $\beta^s_{k,h}$ only affects $C$. $\quad\square$

Next, we account for the adversarial misspecification error.

**Lemma 28.** *Let $\{\epsilon_t\}$ be any sequence such that $|\epsilon_\tau| \leq B$ for any t. Then, for any $(h, k, s) \in [H] \times [K] \times \mathcal{S}$ and $G \in \mathbb{R}^d$, we have*

$$|G^\top (\Lambda^s_{k,h})^{-1} \sum_{t \in T^s_{k-1,h}} G_h(a^t_h)\epsilon_t| \leq B\sqrt{d|T^s_{k-1,h}|G^\top(\Lambda^s_{k,h})^{-1}G}.$$

*Proof.* From the Cauchy-Schwarz inequality and Elliptical Potential Lemma (Lemma D.1 [17]), we have

$$|G^\top (\Lambda^s_{k,h})^{-1} \sum_{t \in T^s_{k-1,h}} G_h(a^t_h)\epsilon_t| \leq B\sqrt{\left( \sum_{t \in T^s_{k-1,h}} G^\top(\Lambda^s_{k,h})^{-1}G \right) \left( \sum_{t \in T^s_{k-1,h}} G_h(a^t_h)^\top(\Lambda^s_{k,h})^{-1}G_h(a^t_h) \right)}$$

$$\leq B\sqrt{d|T^s_{k-1,h}|G^\top(\Lambda^s_{k,h})^{-1}G}.$$

$\square$

Next, we bound the error between a policies $Q$ function and our low-rank estimate.

**Lemma 29.** *There exists a constant $c'_m$ such that $\beta^s_{k,h} = c'_m H(d\sqrt{\iota} + \xi\sqrt{|T^s_{k,h}|d})$ where $\iota = \log(2dST/p)$ and for any fixed policy $\pi$, on the event $\mathcal{X}$ defined in Lemma 27, we have for all $(s, a, h, k) \in S \times S \times [H] \times [K]$:*

$$G_h(a)^\top w^s_{h,k} - Q^\pi_h(s, a) = P_h(V^k_{h+1} - V^\pi_{h+1})(s, a) + \delta^k_h(s, a),$$

*where $|\delta^k_h(s, a)| \leq \beta^s_{k,h}\sqrt{G_h(a)^\top (\Lambda^s_{h,k})^{-1} G_h(a)} + 4H\xi$.*

*Proof.* From Lemma 25, it follows that there exists a weight vector $w^s_h = W_h(a) + \sum_{s' \in \mathcal{S}} V^\pi_{h+1}(s')U_h(s', s)$, such that for all $(s, a) \in \mathcal{S} \times \mathcal{A}$,

$$|Q^\pi_h(s, a) - G_h(a)^\top w^s_h| \leq 2H\xi.$$

Let $\tilde{P}(\cdot|s, a) = G_h(a)^\top U_h(\cdot, s)$, so we have for any $(s, a) \in \mathcal{S} \times \mathcal{A}$, $G_h(a)^\top w^s_h = G_h(a)^\top W_h(a) + \tilde{P}V^\pi_{h+1}(s, a)$. Therefore,

$$\begin{aligned}
w^s_{h,k} - w^{s,\pi}_h = &-\lambda(\Lambda^s_{h,k})^{-1}w^{s,\pi}_h \\
&+ (\Lambda^s_{h,k})^{-1} \sum_{t \in T^s_{k-1,h}} G_h(a^t_h)(V^k_{h+1}(s^t_{h+1}) - P_h V^k_{h+1}(s, a^t_h)) \\
&+ (\Lambda^s_{h,k})^{-1} \sum_{t \in T^s_{k-1,h}} G_h(a^t_h)\tilde{P}_h(V^k_{h+1} - V^\pi_{h+1})(s, a^t_h) \\
&+ (\Lambda^s_{h,k})^{-1} \sum_{t \in T^s_{k-1,h}} G_h(a^t_h)\left( r_h(s, a^t_h) - G_h(a^t_h)W_h(a) + (P_h - \tilde{P}_h)V^k_{h+1}(s, a^t_h) \right)
\end{aligned}$$

Now, we bound each of the four terms above $(q_1^s, q_2^s, q_3^s, q_4^s)$ individually, for the first term, note for all $a \in \mathcal{A}$

$$|G_h(a)^\top q_1^s| = |\lambda G_h(a)^\top (\Lambda_{h,k}^s)^{-1} w_h^{s,\pi}| \le \sqrt{\lambda}\|w_h^{s,\pi}\|\sqrt{G_h(a)^\top (\Lambda_{h,k}^s)^{-1} G_h(a)}$$

For the second term, given the event $\mathcal{X}$, we have from Cauchy Schwarz

$$|G_h(a)^\top q_2^s| \le CdH\sqrt{\chi}\sqrt{G_h(a)^\top (\Lambda_{h,k}^s)^{-1} G_h(a)}$$

where $\chi = \log(2(c_\beta + 1)dTS/p)$. For the third term,

$$
\begin{aligned}
G_h(a)^\top q_3^s &= G_h(a)^\top \left( (\Lambda_{h,k}^s)^{-1} \sum_{t \in T_{k-1,h}^s} G_h(a_h^t) \tilde{P}_h(V_{h+1}^k - V_{h+1}^\pi)(s, a_h^t) \right) \\
&= G_h(a)^\top \left( (\Lambda_{h,k}^s)^{-1} \sum_{t \in T_{k-1,h}^s} G_h(a_h^t) G_h(a_h^t)^\top \sum_{s' \in \mathcal{S}} (V_{h+1}^k - V_{h+1}^\pi)(s') U_h(s', s) \right) \\
&= G_h(a)^\top \left( \sum_{s' \in \mathcal{S}} (V_{h+1}^k - V_{h+1}^\pi)(s') U_h(s', s) \right) \\
&\quad - \lambda G_h(a)^\top \left( (\Lambda_{h,k}^s)^{-1} \sum_{s' \in \mathcal{S}} (V_{h+1}^k - V_{h+1}^\pi)(s') U_h(s', s) \right),
\end{aligned}
$$

and note that

$$G_h(a)^\top \left( \sum_{s' \in \mathcal{S}} (V_{h+1}^k - V_{h+1}^\pi)(s') U_h(s', s) \right) = \tilde{P}_h(V_{h+1}^k - V_{h+1}^\pi)(s, a)$$

and

$$\left| \lambda G_h(a)^\top \left( (\Lambda_{h,k}^s)^{-1} \sum_{s' \in \mathcal{S}} (V_{h+1}^k - V_{h+1}^k)(s') U_h(s', s) \right) \right| \le 2H\sqrt{d\lambda}\sqrt{G_h(a)^\top (\Lambda_{h,k}^s)^{-1} G_h(a)}.$$

Since $\|\tilde{P}_h(\cdot|s, a) - P_h(\cdot|s, a)\|_\infty \le 1$ for all $(s, a) \in \mathcal{S} \times \mathcal{A}$, it follows that

$$|\tilde{P}_h(V_{h+1}^k - V_{h+1}^\pi)(s, a) - P_h(V_{h+1}^k - V_{h+1}^\pi)(s, a)| \le |(\tilde{P}_h - P_h)(V_{h+1}^k - V_{h+1}^\pi)(s, a)| \le 2H\xi.$$

From Lemma 28, we have $|G_h(a), q_4^s| \le 2H\xi\sqrt{d|T_{k,h}^s|G_h(a)^\top (\Lambda_{h,k}^{-1})^{-1} G_h(a)}$. Since

$$|G_h(a)^\top w_{h,k}^s - Q_h^\pi(s, a)| = G_h(a)^\top (w_{h,k}^s - w_h^{s,\pi}) = G_h(a)^\top (q_1^s + q_2^s + q_3^s + q_4^s),$$

it follows that

$$|G_h(a)^\top w_{h,k}^s - Q_h^\pi(s, a) - P_h(V_{h+1}^k - V_{h+1}^\pi)(s, a)| \le \sqrt{\chi}(C"d\sqrt{\chi} + 2\xi\sqrt{|T_{k,h}^s|d})H\sqrt{G_h(a)^\top (\Lambda_{h,k}^s)^{-1} G_h(a)} + 4H\xi.$$

Similarly as in [17], we choose $c_\beta$ that satisfies $C"\sqrt{\log(2) + \log(c_\beta + 1)} \le c_\beta\sqrt{\log(2)}$. $\qquad \square$

Now, we prove that $Q_h^k$ is an upperbound of $Q_h^*$ conditioned on the event in Lemma 27.

**Lemma 30.** *Suppose Assumption 7 holds. On the event $\mathcal{X}$ defined in lemma 27, we have $Q_h^k(s, a) \ge Q_h^*(s, a) - 4H(H + 1 - h)\xi$ for all $(s, a, h, k) \in \mathcal{S} \times \mathcal{A} \times [H] \times [K]$.*

*Proof.* We prove this with induction. The base case holds because at step $H + 1$, the value function is zero, so from Lemma 29,

$$|G_h(a)^\top w_{h,k}^s - Q_H^*(s, a)| \le \beta_{k,h}^s \sqrt{G_h(a)^\top (\Lambda_{h,k}^s)^{-1} G_h(a)} + 4H\xi$$

and thus,

$$Q_H^*(s,a) - 4H\xi \le \min\left(H, G_h(a)^\top w_{h,k}^s + \beta_{k,h}^s\sqrt{G_h(a)^\top(\Lambda_{h,k}^s)^{-1})G_h(a)}\right) = Q_H^k(s,a).$$

From the inductive hypothesis (assuming that $Q_{h+1}^k(s,a) \ge Q_{h+1}^*(s,a) - 4H(H-h)\xi$), it follows that $P_h(V_{h+1}^k - V_{h+1}^*)(s,a) \ge -4\xi$. From Lemma 29, $G_h(a)^\top w_{h,k}^s - Q_h^*(s,a) \le \beta_{k,h}^s\sqrt{G_h(a)^\top(\Lambda_{h,k}^s)^{-1})G_h(a)} + 4H\xi$. It follows that

$$Q_h^*(s,a) - 4H(H-h+1)\xi \le \min\left(H, G_h(a)^\top w_{h,k}^s + \beta_{k,h}^s\sqrt{G_h(a)^\top(\Lambda_{h,k}^s)^{-1})G_h(a)}\right) = Q_h^k(s,a).$$

$\square$

Similarly, to the regular case, the gap in the misspecified setting has a recursive formula.

**Lemma 31.** *Let $\delta_h^k = V_h^k(s_h^k) - V_h^{\pi_k}(s_h^k)$ and $\xi_{h+1}^k = \mathbb{E}[\delta_{h+1}^k|s_h^k, a_h^k] - \delta_{h+1}^k$. Then, on the event $\mathcal{X}$ defined in Lemma 27, we have for any $(h,k)$:*

$$\delta_h^k \le \delta_{h+1}^k + \xi_{h+1}^k + \beta_{k,h}^s\sqrt{G_h(a_h^k)^\top(\Lambda_{h,k}^{s_h^k})^{-1}G_h(a_h^k)} + 4H\xi$$

*Proof.* From Lemma 29, we have for any $(s,a,h,k)$ that

$$Q_h^k(s,a) - Q_h^{\pi_k}(s,a) \le P_h(V_{h+1}^k - V_{h+1}^{\pi_k})(s,a) + \beta_{k,h}^s\sqrt{G_h(a)^\top(\Lambda_{h,k}^s)^{-1})G_h(a)} + 4H\xi.$$

From the definition of $V^{\pi_k}$, we have

$$\delta_h^k = Q_h^k(s_h^k, a_h^k) - Q_h^{\pi_k}(s_h^k, a_h^k),$$

and substituting this into the first equation finishes the proof. $\square$

Finally, we prove the main result in the misspecified setting Theorem 9.

*Proof.* Suppose that the Assumptions required in Theorem 6 hold. We condition on the event $\mathcal{X}$ from Lemma 27 with $p = \delta/2$ and use the notation for $\delta_h^k, \xi_h^k$ as in Lemma 31. From Lemma 16, we have $Q_1^k(s,a) \ge Q_1^*(s,a) - 4H^2\xi$, which implies $V_1^*(s) - V_1^{\pi_k}(s) \le \delta_1^k + 4H^2\xi$. Thus, from Lemma 29, on the event $\mathcal{X}$, it follows that

$$Regret(K) = \sum_{k=1}^K (V_1^*(s_1^k) - V_1^{\pi_k}(s_1^k)) \le \sum_{k=1}^K \delta_1^k + 4H^2\xi$$

$$\le \sum_{k\in[K]}\sum_{h\in[H]} \xi_h^k + \sum_{k\in[K]} \beta_{k,h}^s \sum_{h\in[H]} \sqrt{G_h(a_h^k)^\top(\Lambda_{h,k}^{s_h^k})^{-1}G_h(a_h^k)} + 4HT\xi$$

since $HK = T$. Since the observations at episode $k$ are independent of the computed value function (this uses the trajectories from episodes 1 to $k-1$, it follows $\{\xi_h^k\}$ is a martingale difference sequence with $|\xi_h^k| \le 2H$ for all $(k,h)$. Thus, from the Azuma Hoeffding inequality, we have

$$\sum_{k\in[K],h\in H} \xi_h^k \le \sqrt{2TH^2\log(2/p)} \le 2H\sqrt{T\iota}$$

with probability at least $1 - \delta/2$. Note that

$$\sum_{k\in[K]} \beta_{k,h}^s\sqrt{G_h(a_h^k)^\top(\Lambda_{h,k}^{s_h^k})^{-1}G_h(a_h^k)}$$

$$\le CH(\sum_{k\in[K]} d\sqrt{\iota}\sqrt{G_h(a_h^k)^\top(\Lambda_{h,k}^{s_h^k})^{-1}G_h(a_h^k)} + \sum_{k\in[K]} \xi\sqrt{|T_{k,h}^s|d}\sqrt{G_h(a_h^k)^\top(\Lambda_{h,k}^{s_h^k})^{-1}G_h(s_h^k)}).$$

From the Cauchy-Schwarz inequality, it follows that

$$\sum_{k\in[K]} \sqrt{G_h(a_h^k)^\top (\Lambda_{h,k}^{s_h^k})^{-1} G_h(a_h^k)} \leq \left(\sqrt{K}\right) \left(\sqrt{\sum_{k\in[K]} G_h(a_h^k)^\top (\Lambda_{h,k}^{s_h^k})^{-1} G_h(a_h^k)}\right).$$

Let $k_i$ refers to the episode in which state $s$ was seen at time step $h$ for the $i$th time. Then, for the other term, we first re-index the summation and then use the Cauchy-Schwarz inequality to get

$$\sum_{k\in[K]} \sqrt{|T_{k,h}^s|}\sqrt{G_h(a_h^k)^\top (\Lambda_{h,k}^{s_h^k})^{-1} G_h(a_h^k))} = \sum_{s\in\mathcal{S}} \sum_{i=1}^{|T_{K,h}^s|} \sqrt{i}\sqrt{G_{k_i,h}^\top (\Lambda_{k_i,h}^a)^{-1} G_{k_i,h}}$$

$$\leq \left(\sum_{i=1}^{|T_{K,h}^s|} i\right)^{1/2} \left(\sum_{t\in T_{K,h}^s} G_{k,h}^\top (\Lambda_{k,h}^{s_h^k})^{-1} G_{k,h}\right)^{1/2}$$

$$\leq C'|T_{K,h}^s| \left(\sum_{t\in T_{K,h}^s} G_{k,h}^\top (\Lambda_{k,h}^{s_h^k})^{-1} G_{k,h}\right)^{1/2}$$

for some absolute constant $C' > 0$. Since the minimum eigenvalue of $\Lambda_{h,k}^{s_h^k}$ is at least one by construction and $\|G_h(s)\| \leq 1$, from the Elliptical Potential Lemma (Lemma D.2 [17]), we have for all $a \in \mathcal{A}$ and $h \in [H]$,

$$\sum_{k=1}^{K} G_h(s_k)^\top (\Lambda_{h,k}^a)^{-1} G_h(s_k) \leq 2\log(det(\Lambda_{h,k+1}^a)/det(\Lambda_{h,0}^a))$$

Furthermore, we have $\|\Lambda_{h,k+1}^a\| = \|\sum_{t\in T_{k,h}^s} G_h(s_h^t)G_h(s_h^t)^\top + \lambda I\| \leq \lambda + |T_{k,h}^s| \leq \lambda + K$. It follows that

$$\sum_{k=1}^{K} G_h(s_k)^\top (\Lambda_{h,k}^a)^{-1} G_h(s_k) \leq 2d\log(1 + K/\lambda) \leq 2d\iota,$$

so grouping the episodes by actions gives

$$\left[\sum_{k\in[K]} G_h(a_h^k)^\top (\Lambda_{h,k}^{s_h^k})^{-1} G_h(a_h^k)\right]^{1/2} = \left[\sum_{s\in\mathcal{S}} \sum_{t\in T_{K,h,a}} G_h(a_h^t)^\top (\Lambda_{h,t}^a)^{-1} G_h(a_h^t)\right]^{1/2} \leq \sqrt{2dS\iota}$$

Thus,

$$\sum_{k\in[K]} \beta_{k,h}^s \sqrt{G_h(a_h^k)^\top (\Lambda_{h,k}^{s_h^k})^{-1} G_h(a_h^k)} \leq CH(\sqrt{2d^3 SK\iota^2} + \xi\sqrt{d}C'\sum_{s\in\mathcal{S}} |T_{k,h}^s|\sqrt{2d\iota})$$

$$\leq C(\sqrt{2d^3 SHT\iota^2} + \xi\sqrt{2\iota}dT.$$

Substituting these results into the original regret bound gives

$$Regret(K) \leq C'''(\sqrt{d^3 H^3 ST\iota^2} + dTH\xi\sqrt{\iota}).$$

for some constant $C''' > 0$ with probability at least $1 - \delta$. $\qquad\square$

