# OpenReview forum: "The Limits of Transfer Reinforcement Learning with Latent Low-rank Structure"
_NeurIPS.cc/2024/Conference — NeurIPS 2024 poster_

### Official Review · Reviewer_LmTe · 2024-07-11

**Soundness:** 2
**Presentation:** 1
**Contribution:** 3
**Rating:** 5
**Confidence:** 2

**Summary:**

The paper investigates transfer in reinforcement learning, where one tries to exploit latent structure common across several MDPs. Specifically the paper consider M "source" episodic MDPs sharing a latent low-rank structure of the transition matrix and one "target" episodic MDP whose transition matrix is also-low rank, with latent features on the span of the source features. The main results take the form of minimal sample-complexity bounds on the source MDPs that induce a regret bound on the target MDP that avoid a dependence in the size of the state-action space. Since the transition matrix of a MDP is actually a 3-tensor, several notions of low-rank are considered in the sense of the Tucker rank of a tensor, which correspond to different ways of factorizing $P(s' | s,a)$ as a product of two matrices. One such low-rank assumption covers the case of low-rank and linear MDPs. For each of these cases, the paper also identifies a transferability coefficient from which a sample-complexity lower bound is also established.

**Strengths:**

The numerous results are novel and of interest. The paper is really complete in considering basically all the possible assumptions of low-rank that can be made on the transition matrix.

**Weaknesses:**

The paper is quite long, technical and I think some effort could be made to make it easier to read. I have not gone carefully over all the proofs but I have found quite a number of statements that I did not understand. There is also a large number of typos. A typo here and there should normally not be a big problem, but their number is quite irritating. I think a certain number of them arises from the similarity between the different Tucker rank assumptions.

**Questions:**

Essentially, I simply suggest improving the clarity of the paper.  I have made a list of a few things which caught my attention, but this is probably not exhaustive.

1. I do not understand the proof of Theorem 1:

a. It is really difficult to identify the different quantities involved in Assumptions 1 and 2, such as the latent factors $G_i$, from what is given in the proof. This all the more so true as the notation is really heavy (e.g. $Q_{1,1}^{\ast,1}$) and inconsistent, resulting in conflicts of notation: for example in line 513 $G_i$ refers to the target latent factor of problem $i$ but this notation is also used in Assumption 1 with $i$ referring to a time step. From the notation used in the proof $i$ should be used as an exponent to refer to problem $i$.

b. In line 508 $c = \sqrt{1+1/\alpha^2}$ is not bounded by $2$ as $\alpha \rightarrow 0$...

c. There is a $c$ missing in the equation of line 512: $[\sqrt{1/2n} \quad - \sqrt{1/2n}] = - \alpha [\sqrt{1/2n} \quad - \sqrt{1/2n}] + \alpha c G'$.

d. I guess the parameter $\alpha$ of the proof is supposed to be the transferability coefficient of Definition 3. Is it still the case with the missing $c$? Furthermore the relation between Definition $3$ and the difference between entries of $Q^{\ast}$-s is not clear.

e. The end of the argument (l. 526 - 529) is too elusive: the relation with the rest of the proof is not explained. One does not understand how the Bernoulli variable $X$ comes into play.

 f. l. 527 is the reader supposed to read all the 389 pages of [5] to be convinced that ”the probability of correctly identifying $G$ is upper bounded by 0.76 [5]"?

g. Theorem 1 requires Assumptions 1 and 2 but the proof involves Assumptions 5 and 6

h. Assumptions 1 and 2 are assumptions on transition matrices but in the proof these are deduced directly from the form of the $Q$-functions

 i. The matrices given in the proof have size $2 \times 1, 1 \times 2$ or $2 \times 2$ but the example is about a state space of size $[2n]$. If the matrices are block matrices this needs to be emphasized. What about orthonormality of columns?

j. Finally, there is (at least) one typo in the second equation of line 505: $Q_{1,2}^{\ast,1}$ should be $Q_{2,1}^{\ast,1}$.

2.Some notations change throughout the paper. For instance, the number of episodes (which i think is not defined anywhere) is sometimes written $K$, sometimes $T$. Same for the rank of the matrices considered, it is sometimes $r$, sometimes $d$ (e.g. from Prop. 2 to Corollary 3)

3. p.2, l 113-116, there is a problematic conflict of notation in having $P_{h}$ stand for both the transition matrix of the source MDP at time $h$ and the family of transition matrices of the $h$-th MDP

4. p.4 In Definition 1 the factors $G_i$ belong to $\mathbb{R}^{n_i \times d_i}$ instead of just $\mathbb{R}^{n_i}$. Strictly speaking orthonormal matrices need to be square matrices so if $d_i \neq n_i$ they only have orthonormal columns. It should be $G_{h}(a,i)$ instead of $G_{h}(s,i)$

5. p.4 l.150 "Figure 1 pictorially displays the $(S,d,A)$ Tucker rank decomposition": as indicated by the caption, the case displayed is $(S,S,d)$...

6. p.5 In Assumption 1 l; 192 It should be $G_{h}(a,i)$ instead of $G_{h}(s,i)$. Same remark as above, orthonormality refers to orthonormality of the columns.

7. p.7 l.281 $Q_{h,m}^{\ast}$ should be $Q_{m,h}^{\ast}$

8. p.7 l.290: I think $s_{h+1}^{k}$ should be $s_{h+1}^{t}$.

9. p.9 l.352 "Algorithm 2, 2": remove one 2

10. p.15 in the proof of Theorem 2 how is $\bar{\gamma}$ bounded? Shouldn't there be an additional factor $8 / \sqrt{\mu}$ in l.556?

11. p.16 l.568 I assume $|| \cdot ||_{TV}$ is total variation distance. It should be defined explicitly. Also there shouldn't be an $s'$ in the left hand side, and the sum in the right hand side should be inside the absolute value.

12. In Appendices F and G, I suggest adding more references, links between proofs and equations, recalling some assumptions that are made to simplify the exposition.

12. p.36 in Prop.2 the second bound should involve $V_j^{\ast}$ instead of $U_j^{\ast}$ in the right hand side

13. p.38 I do not understand the proof of Corollary 3: from Prop.2 one should bound $\bar{\gamma}$. The proof gives a bound on $\|| A^{\ast} \||_{\infty}$.

14. p.39 l.1131 in the first equation there is a $\top$ sign that should be in exponent

15. p.48 l.1323 $s$ appears both as an argument of $V$ and in the maximum

16. p.49 l.1331 a word is missing in "where the holds"

17. p.49 l.1340 I do not understand where the last inequality comes from

18. p.49 in the proof of Lemma 20 I do not understand where the third inequality comes from. Also the indices $h$ and $k$ are reversed in $T_{h,k-1}$ are reversed compared to when the notation was introduced in l.286

19. p.49 in the 3rd equation of l.1345, there is no $\top$ on the second $g$ term

**Limitations:**

The authors addressed the limitations of the paper.

---

> ### Author Rebuttal · Authors · 2024-08-07
>
> We thank reviewer LmTe for the suggestions on how to improve the presentation of our paper and have made the modifications for the final version. We have corrected the typos addressed by points 1a, 1c, 1g, 1j, 3, 4, 5, 6, 7, 8, 9, 13, 15, 16, 17, and 20.
> Regarding Theorem 1, the key idea is that we can construct two transfer RL problems with similar source $Q^*$s but use orthogonal feature representations in the target phase. As the learner is given one of the transfer RL problems with equal probability, they must identify which one it is to avoid using an orthogonal feature mapping with high probability. To identify the transfer RL problem, one must distinguish between the $Q^*$ from the different source MDPs, which depends on $\alpha$, the parameter to the result. From standard hypothesis testing sample complexity lower bounds (we need the noise variable to be Bernoulli to use this result), one must incur a sample complexity of $o(\alpha^2)$ to identify the correct transfer RL problem and feature mapping.
>
> 1b: Thank you for pointing this out. We will clarify that $\alpha$ is lower bounded by 1/(dM) (see Lemma 2). With a loose upper bound, it follows that $c \in (1, 2dM)$. As $d, M > 1$, we have that $2\alpha dM > \alpha$ and still require that one must observe $o(\alpha^2)$ samples to benefit from transfer learning.
>
> 1e: The Bernoulli variable specifies the noise model in our setting; one observes a realization of a Bernoulli random variable when specifying a $Q^*$ and state-action pair to observe from. We chose Bernoulli random variable as our noise model to use the result from [5].
>
> 1f: Thank you for pointing this out. The relevant result is Lemma 5.1 from [5] by setting $\delta = 0.24$, and we have added this to our paper.
>
> 1i: Yes, these matrices are block matrices. Thank you for your suggestions. We will emphasize this. Also, as the matrices are rank 1, the columns are orthogonal to each other. Furthermore, the norm of each block vector is 1 when accounting for the size of the blocks.
>
> 10: First, note that from Assumption 3, $\|Q^\*\|\_{\infty} \geq C$, so $\frac{1}{\sigma\_d} \leq  \frac{\|Q^\*\|\_\infty }{ C\sigma\_d}$. From the last equation on page 48 from [28], we have that $\frac{\|Q^\*\|\_\infty}{C\sigma\_d} \leq \frac{\kappa d \mu}{\sqrt{S A} C}$. Thus, we have $\bar{\gamma} \leq \frac{ \kappa d \mu }{ (\sqrt{S A} C)}  \sqrt{S A} \|D\|\_\infty$. From the source sample complexity, it follows that $\|D\|\_\infty \leq \frac{C}{2d\mu}$, which proves that $\bar{\gamma} \leq \frac{1}{2}$. Thank you for pointing this out, we will add more intermediate steps and clarification in the text. We have updated Corollary 3 as it was missing a factor of $\mu$.
>
> 11: Thank you for catching these errors. We have redefined our definition of misspecified low Tucker rank MDPs to $| \sum\_{s’ \in \mathcal{S}} P(s’|s, a) - G(a)U(s’, s) | \leq \xi$ instead of using the total variation distance as our analysis holds with the above definition.
>
> 14: We first show that $\bar{\gamma} = \frac{\|D\|\_{op}}{\sigma\_r} = \frac{\kappa d \mu \|D\|\_\infty}{C}$. Combining this with the definition of incoherence and Proposition 2, we get the result of Corollary 3.
>
> 18: recall that since the feature mapping was scaled up ($G_h = \sqrt{\frac{A }{d\mu}} G'$) to ensure the max entries of $G$ and $W, U$ are on the same scale, it follows that $|W\_{h}(s)|\_2 \leq 1$ and $ \|\sum\_{s' \in \mathcal{S}} V\_{h+1}^\pi(s') U\_{h}(s', s) \|\_2 \leq H$. Therefore, it follows that $\|w\_h^s\|\_2 \leq 2H\sqrt{d}$.
>
> 19: The rightmost term is upper bounded by $\sqrt{d}$ from Lemma D.1 in [15]. As $|T_{k-1, h}^s| \leq k$, it follows that $\sqrt{ \sum_{t \in T_{k-1, h}^s} g^\top (\Lambda\_h^s)^{-1} g} \leq \sqrt{k  g^\top (\Lambda\_h^s)^{-1} g}$. Since $ \Lambda\_h^s $is real and symmetric with smallest eigenvalue lower bounded by $\lambda$, it follows that the largest eigenvalue of $(\Lambda\_h^s)^{-1}$ is upper bounded by $\lambda$. It follows that  $g^\top (\Lambda\_h^s)^{-1} g \leq \|g\|_2^2/\lambda$ from the fact that the maximum value a Rayleigh quotient $ (g^\top (\Lambda\_h^s)^{-1} g/ g^\top g) $is the largest eigenvalue of $(\Lambda\_h^s)^{-1} $(see Theorem 4.2.2 from Matrix Analysis by Horn et al).

---

> > ### Comment · Reviewer_LmTe · 2024-08-12
> > **Answer to rebuttal**
> >
> > I thank the authors for their answers to my questions. I am really satisfied with the explanation regarding the lower bound which was my main concern. A precise reference to Lemma 5.1 in [5] was undoubtedly the missing ingredient to make it clear and convincing. I think there remains a small typo in that the "magnitude of the largest entrywise difference" (l.524) is $\Omega(1/\alpha)$, not $\Omega(1/\alpha^2)$, the square coming from the application of the lemma.
> >
> > The other (less important) points raised have also found satisfying clarification. I'll add the remark however that these were intended to be "samples" of what I considered flaws in the paper, so that "more intermediate steps and clarification" do not need to restrict to these of course.

---

> > > ### Author Response · Authors · 2024-08-12
> > >
> > > We thank reviewer LmTe for catching this typo and have corrected it for the final version.

---

### Official Review · Reviewer_nw9E · 2024-07-12

**Soundness:** 3
**Presentation:** 3
**Contribution:** 2
**Rating:** 6
**Confidence:** 3

**Summary:**

This paper addresses the computational and data inefficiencies of reinforcement learning (RL) algorithms due to large state and action spaces. It introduces a transfer learning approach that utilizes latent low-rank structures in the transition kernels of source and target Markov Decision Processes (MDPs). The paper presents a new algorithm that achieves efficient transfer by learning and utilizing latent representations, significantly reducing the dependency on state and action spaces in the target MDP. The paper provides theoretical guarantees for their algorithms, including source sample complexity and target regret bounds for each Tucker rank setting. The authors also discuss connections to related work in linear MDPs, low-rank MDPs, and other transfer RL approaches

**Strengths:**

- The work introduces novel algorithms for transfer RL with latent low-rank structures. It explores multiple Tucker rank settings, offering a comprehensive framework not fully addressed in prior works.
- The introduction of the transfer-ability coefficient $\alpha$, which quantifies the difficulty of transferring latent representations, is a novel concept that enhances the understanding of transfer dynamics in RL.
- The submission is technically sound, with rigorous theoretical analyses supporting the claims, though the reviewer couldn’t check the correctness of all details. The problem is well-formulated, and the methods used are appropriate and well-executed, indicating a complete work.

**Weaknesses:**

- While the paper is mostly original, it builds significantly on existing concepts in low-rank MDPs and linear MDPs. More explicit discussion on how this work diverges from traditional approaches could enhance its originality.
- The assumptions required for the theoretical results, such as the specifics of the Tucker rank and the incoherence conditions, might limit the applicability of the results in practical, real-world scenarios where such assumptions may not hold.
- The practical implications and applications of the proposed methods could be highlighted more. Discussions on how these methods could be integrated into existing RL systems or specific real-world applications would enhance the paper's impact.

**Questions:**

- The paper assumes specific structures for the MDPs. How robust are the proposed methods to deviations from these assumptions?
- How sensitive is the algorithm's performance to changes in the transfer-ability coefficient $\alpha$? What are the practical steps for estimating $\alpha$ in a new environment?
- Could the authors conduct numerical experiments like [4, 28] did in their paper? It would be valuable to include empirical comparisons against relevant baselines on benchmark RL tasks to demonstrate practical performance gains.
- Could the authors clarify the practical implications of the transfer-ability coefficient in real-world scenarios? How can one estimate this coefficient in practice?
- The current setup focuses on transferring from the source to a single target task. Is there potential to extend this framework to continual or multi-task learning settings? How does this approach relate to other forms of transfer in RL, such as policy distillation or meta-learning?
- Typos
    - Inconsistency in Line 150 and Figure 1 caption. It seems $(S, S, d)$ Tucker rank.
    - Line 352: Algorithm 2, 2,

**Limitations:**

The paper discusses the minimax optimality of theoretical guarantees with respect to the parameters or some assumptions of the problem, but a more direct mention of this paper's limitations would make the paper more complete. Furthermore, practical limitations, such as the algorithms' scalability to extremely large state and action spaces or their performance under model misspecification, are not thoroughly examined.

---

> ### Author Rebuttal · Authors · 2024-08-06
>
> We appreciate reviewer nw9e’s feedback and suggestions on how to improve our paper. We completely agree that our paper would benefit from numerical experiments and will look into running simulations to illustrate the benefits of our algorithm.
>
> **Q2&4:** Regarding estimating $\alpha$, $\alpha$ is a fundamental quantity in each transfer RL problem. The significance of $\alpha$ is that for large $\alpha$, transfer learning approaches will perform worse than tabular MDP algorithms which ignore the source MDPs. While our source sample complexity and target regret bound depend on $\alpha$, we emphasize that **one does not need to know $\alpha$ to run our algorithm.** When running our algorithm in practice, one should observe as many samples in the source phase as allowed by their computational budget/constraint to obtain the best performance on the target problem.
>
> **Q5:** Our approach will work for a multi-task learning setting with multiple target MDPs, as long as each of the target MDPs satisfy our transfer learning assumption (Assumption 2), i.e., each target task’s feature representation lies in the space spanned by the source MDP feature representations. Our algorithm would proceed in the same way to estimate the subspaces in the source phase, and then would run LSVI-UCB-(S, S, d) on each target MDP subsequently, assuming that all MDPs have transition kernels with Tucker rank $(S, S, d)$. The same approach works in the other Tucker rank settings.
> Our approach differs from policy distillation algorithms as we only require the feature representation from the target MDP to be contained in the space spanned by the source MDP feature representation. While one typically transfers a policy or $Q$ function from a teacher (source MDP) to a student (target MDP) in policy distillation, in our setting, the optimal $Q$ function and policy in the source MDPs can be very different from the optimal $Q$ function and policy from the target MDP. Therefore, with our assumptions, transferring only a policy or $Q$ function can lead to no improvement on the student or target MDP.
> Our setting is similar to the meta RL setting as the agent attempts to learn information from the source MDPs or meta tasks to improve the efficiency of learning in the target MDP. By learning a sufficient feature representation from the source MDPs (meta tasks), one can learn a good policy with significantly fewer samples, which is the goal in meta RL.

---

> ### Comment · Reviewer_nw9E · 2024-08-11
> **Thank you for your response**
>
> I really appreciate your detailed response to my questions. After reading the authors' responses and comments from other reviewers, I will keep my score.

---

### Official Review · Reviewer_Stu9 · 2024-07-31

**Soundness:** 3
**Presentation:** 2
**Contribution:** 3
**Rating:** 7
**Confidence:** 3

**Summary:**

This work considers transfer RL, where the source and target MDPs admit low Tucker rank. An information-theoretic lower bound is derived for the source sample complexity, and the proposed algorithm is minimax optimal respecting the transfer-ability coefficient $\alpha$ (in the case of $(d,S,A)$). The results do not assume the representations lie within the given function class.

**Strengths:**

(+) The paper considers different types of low-Tucker-rank MDPs, corresponding to different factorizations of the transition kernel, which can provide new insights into the literature.

(+) The derived source sample complexity does not scale with the size of the given function class.

(+) An information-theoretic lower bound is derived for the source sample complexity, and the proposed algorithm for $(d,S,A)$-Tucker-rank is minimax optimal respecting the transfer-ability coefficient $\alpha$.

**Weaknesses:**

(-) The target regret bound grows polynomially in the number $M$ of source MDPs.

(-) Assumption 3 of full-rank $Q^*$-function can be quite restrictive, and does not hold in simple settings such as goal-reaching tasks in a binary tree (i.e., yielding reward only when reaching some leaf node at the last layer).

**Questions:**

1. Does alg in [4] also work for MDPs with Tucker ranks $(S,S,d),(S,d,A)$, and $(d,d,d)$? If yes, are the source sample complexity and the target regret bound the same to the case of $(d,S,A)$? If not, is there any reason?
2. As different algorithms are required for various types of low-Tucker-rank MDPs, could you provide examples of how to choose the algorithm in practice?
3. I am quite confused by the statement in Lines 325-328. What is the difference between "a subset of the space spanned by the source features" and "being a linear combination of the source features"? According to (Assumption 2.2, [4]), it is possible that some $\alpha_{k;h}$'s are zero.
4. What is $Q_{2,1}^{\*,1}$ at Line 523 (should it be $Q_{1,2\}^{*,1}$)? Also, could you provide more explanations on why the agent needs to differentiate between the two optimal $Q$-functions to enable the transferring?
5. To improve interpretability, I suggest adding a column in Table 1 and listing the target regret bounds with known latent representations.

**Limitations:**

The limitations are addressed in the paper.

---

> ### Author Rebuttal · Authors · 2024-08-06
>
> We appreciate reviewer Stu9’s feedback and suggestions on how to improve our paper. We completely agree that our paper would benefit from adding a column listing the target regret bounds in Table 1 and will add it to the final version.
>
> **Weakness 2:** We realize that our terminology is misleading in Assumption 3. Our “full-rank” $Q^*$ assumption asserts that $rank(Q^*) = d$, not that $rank(Q^*) = \min(S, A)$. We have fixed this for the final version and have removed the mention of “full-rank”.
>
> **Q1:** The algorithm from [4] does not work in the $(S, S, d)$ and $(S, d, A)$ Tucker rank settings due to the difference in structural assumptions on the transition kernel. Specifically, [4] assumes low rank structure across the transition forward in time (between $s’$ and $(s, a)$) whereas in our work, we assume that there is low rank structure between $(s, s’)$ and $a$ or $(a, s’)$ and $s$. As a result, our algorithm learns feature mappings and weight vectors with different dimensions than in [4]. In the $(d, d, d)$ Tucker rank setting, one could use the algorithm in [4], which admits a target regret bound that matches ours with respect to $d, T, S, A$. However, our source sample complexities differ as the one using algorithm [4] depends on the size of the function class that the feature representations belong to instead of $\mathcal{S}$ or $\mathcal{A}$, and our algorithm does not require a computationally inefficient oracle in the source phase.
>
> **Q3:** We apologize for the typo/mistake as they mean the same thing. The main difference in our models is that in [4] each MDP, source and target, share the same d-dimensional feature mapping while ours only assumes that the space spanned by the target feature mapping lies in the space spanned by the source feature mappings. This means that our results actually allow the target MDP to have up to $dM$-feature mapping if each of the $M$ source MDP subspaces are disjoint.
>
> **Q4:** In our lower bound, the learner is given one of two transfer RL problems, 1 and 2, with equal probability where the one latent feature is optimal in problem 1 and the other is optimal in problem 2. The two latent factors are orthogonal to each other, so choosing the wrong latent feature is no better than randomly guessing a latent feature. Thus, to benefit from transfer learning with high probability, one must identify which transfer RL problem they are dealing with. To do so, one must determine if the $Q$ function from the second source MDP is $Q_{2, 1}^{ \*, 1} $ or $Q^{\*, 2}_{2, 1}$.

---

> > ### Comment · Reviewer_Stu9 · 2024-08-09
> > **Follow-up Questions**
> >
> > I thank the authors for their in-depth responses to my questions. Responding:
> >
> > **Dependence on $M$:** I am confused by the statement "...the feature mapping we learn in the source phase has dimension $dM$". Do you mean the feature mapping $\tilde{G}_h$ computed in Line 4 of Algorithm 1 has dimension $dM$ (but the equation between Lines 290 and 291 implies that $\tilde{G}_h\in\mathbb{R}^d$)? Also, the definition of $\tilde{G}_h$ is a bit confusing. Is it a set?
> >
> > My understanding is that when the features lie in the same $d$-dimensional space (i.e., linear combination), then $\tilde{G}_h$ would be $d$-dimensional; however, without such prior knowledge (i.e., disjoint subspaces), then we need to construct a $dM$-dimensional $\tilde{G}_h$. Is my understanding correct?
> >
> > I am also wondering whether the algorithm can distinguish between these two cases (and hence remove the dependence on $M$).
> >
> >
> > **Q1.** Can the algorithm in [4] assume access to representations of the form $\phi(\cdot,\cdot):\mathcal{S}\times\mathcal{S}\to\mathbb{R}^d$ and $\mu(\cdot):\mathcal{A}\to\mathbb{R}^d$ for MDPs with $(S,S,d)$-Tucker rank? I think this would work.
> >
> > **Q4.** Is there a typo in the second equation between Lines 505 and 506, where $Q\_{1,2}^{\*,1}$ on the LHS should be $Q\_{2,1}^{*,1}$?

---

> > > ### Author Response · Authors · 2024-08-12
> > >
> > > **Dependence on $M$**: Thank you for pointing this out, we realize our notation of $\tilde{G}\_h$ is confusing.
> > >
> > > We first remark that our current assumptions in our work are stronger than what we actually need. Instead, our analysis only requires that $G\_h$ satisfy Assumption 2 instead of both Assumptions 1 and 2, which allows the target MDP to have Tucker rank at most $(S, S, dM)$ if the source subspaces are disjoint.
> > >
> > > $\tilde{G}_h$ is not a set; it is a $S \times dM$ matrix. In step 4 of Algorithm 1 (line 287), to construct $\tilde{G}\_h$, we concatenated the estimated singular vectors $\tilde{G}_h = \text{Concat}(\hat{G}\_{1, h}, \hat{G}\_{2, h}, \ldots, \hat{G}\_{m, h})$ from each source MDP so that $\tilde{G}\_h$ has dimension $S \times dM$.
> > >
> > > This approach as stated cannot distinguish between the two cases you presented, i.e., whether the subspaces of the feature mappings from the source MDPs are disjoint or not, and the constructed feature mapping has dimension $dM$ in both cases. Nonetheless, this issue can be readily fixed, as we described below.
> > >
> > > To improve our results in the case when the feature mappings from the source MDPs lie in the same d-dimensional subspace, we can add the following procedure: after computing $\tilde{G}\_h$ via concatenation of $\hat{G}\_{m, h}$ for all $m \in [M]$, we perform a singular value decomposition of $\tilde{G}\_h$ and threshold the singular values to remove unneeded dimensions; the constructed feature mapping is then the concatenation of the singular vectors of  $\tilde{G}\_h$ with sufficiently large singular values. Specifically, a short calculation shows the following: we can zero out the  singular values smaller than  $$\sqrt{\frac{d \mu H S}{\alpha^2 T A M}}, $$ and doing so would remove the dependence on $M$ in the target phase regret bound.
> > >
> > > However, in the case when the feature mappings from each source MDP lie in orthogonal d-dimensional subspaces, we cannot discard any unique dimension of the $M$ subspaces as we cannot tell which of the $d$ dimensions match the ones in the target MDP without interacting with it. Thus, we need to include all of them, which still results in at worst a $dM$ dimensional feature representation.
> > >
> > > Thank you for pointing out the issue in lines 290-291, we will fix it to use the correct dimension.
> > >
> > >
> > > **Q1**: If the learner had access to representations  of $\phi(\cdot,\cdot):\mathcal{S}\times\mathcal{S}\to\mathbb{R}^d$ and $\mu(\cdot):\mathcal{A}\to\mathbb{R}^d$ for MDPs with $(S,S,d)$-Tucker rank, we agree that the algorithm from [4] would work in the source phase provided that one adapts the algorithms and analysis to the $(S, S, d)$-Tucker rank setting. However, one must still use our target phase algorithm to get the $\sqrt{A}$ dependence in the regret bound. Using the target phase algorithm from [4] results in a $\sqrt{A^3}$ regret bound. We also note that with this modification, it is possible that the source sample complexity will depend on $S$, unlike the vanilla linear MDP setting where the regret/complexity is independent of $S$.
> > >
> > > **Q4**: Thank you for catching this typo, we have corrected it for the final version.

---

> > > > ### Comment · Reviewer_Stu9 · 2024-08-13
> > > > **Thank you**
> > > >
> > > > I appreciate the detailed clarification from the authors. I have raised my score accordingly. Responding:
> > > >
> > > > 1. I wonder if the thresholding procedure is also applicable to (intermediate) cases where the feature mappings lie in an $n$-dimensional subspace with $d<n<dM$.
> > > >
> > > > 2. To improve readability, I suggest including a notation list in the Appendix with clear explanations for (at least the essential) variables used in the analysis.

---

> ### Author Response · Authors · 2024-08-13
>
> **1:** Yes, the thresholding procedure also works for the intermediate case. This procedure computes the union of the dimensions from all the source MDP subspaces. It then discards the repeated dimensions from each source MDP (as well as superfluous dimensions due to estimation error in the source MDPs). Thus, if the feature mapping lies in a $n$-dimensional subspace where $d < n < dM$, then the union of the subspaces of all the source MDPs is also $n$-dimensional, in which case the thresholding procedure will output $\tilde{G}_h$ with dimension $n$ (with high probability with respects to the estimation errors).
>
>  **2:** We thank reviewer Stu9 for the suggestion and will include a notation table in the appendix in the final version.

---

### Author Rebuttal · Authors · 2024-08-06

We appreciate the reviewer’s feedback and suggestions. We discuss the primary shared concerns of the reviewers below, and have deferred addressing clarification questions to the individual reviewer rebuttals. We will incorporate the below discussion into the final paper.

**Intellectual novelty (nw9E):** Our major contribution is understanding the benefits of assuming low-rankness along any one of the three modes of the transition kernel in transfer RL, whereas [4] only considers a single mode of low rank structure. Our algorithms in the $(S, S, d)$ and $(S, d, A)$ Tucker rank settings are novel as directly using any linear MDP algorithm, e.g., UCB-LSVI from [15] used in [4], does not achieve our $\sqrt{S}$ or $\sqrt{A}$ dependence (it attains a suboptimal scaling of $S$ or $A$). Additionally, we provide information theoretic lower bounds that prove the optimality of our algorithms with respect to $\alpha$, which is an open question posed in [4], and extend the concept of the transferability coefficient to the additional modes of low rank structure beyond what is considered by [4]. Finally, in contrast to the algorithm in [4], our algorithm does not require access to a computationally inefficient oracle.

**Motivation of our low Tucker rank and incoherence assumptions (Stu9, nw9E):**  Any block MDP satisfies our low Tucker rank assumption, whether the block membership is known or not, as the complexity of the problem scales with the number of latent states/actions (i.e. blocks) instead of the size of the observable spaces. As long as the blocks are not too small, then the MDP will also satisfy incoherence. Another example in which approximately low rank structure will likely hold is a discrete MDP with state and action spaces that can be well approximated by a smooth continuous population distribution. For example, a large population of users and products from a recommender system could likely be approximated by a smooth distribution of user types and product types. Alternatively, the MDP could have been constructed by a discretization of a continuous MDP, e.g. stochastic control tasks. In these MDPs, the complexity of the discrete state and action space depends primarily on the smooth continuous distribution instead of the size of the discretized space. Thus, the complexity, or rank or Tucker rank, of the reward function and transition kernel, respectively, is likely to be approximately low dimensional, i.e. independent of $S, A$. [Udell et al. 2018] formalized this argument for matrices, showing that smoothness conditions lead to approximate low rank structure; a slight modification of their analysis should also extend to tensors. Additionally, due to the smoothness conditions, the MDP would likely satisfy incoherence as there cannot be a small number of states or actions that deviate significantly from all other states and actions.

When the MDPs have only approximately low Tucker rank, our results degrade smoothly with respect to the misspecification error. In particular, assuming that the source MDP’s and target MDPs’ reward functions are $\tau$ away from a low rank matrix with respect to the $\ell_\infty$ norm and the transition kernels are $\tau$ away from a low Tucker rank transition kernel with respect to the total variation distance (see Assumption 5 from [29]), in the $(S, S, d)$ Tucker rank setting, our algorithm’s source sample complexity remains the same while the target regret bound becomes
$$
{O}(\sqrt{(dMH)^3A T} + \kappa^3\mu^3d^4M^2H^3 \tau T \sqrt{\frac{A }{S }}).
$$
Thus, if the cardinality of the state and action space are proportional to each other, our algorithm degrades smoothly with respect to the misspecification error, incurring an additional regret of ${O}(\tau T \text{poly}(d, \kappa, \mu, M , H))$.
	Determining which of our algorithms to use depends on what structure is present in the problem. For example, if one believes that only the actions live in a much smaller dimensional space, then one should use the $(S, S, d)$ Tucker rank algorithm. However, if both states and actions have low rank representations, one should use the $(d, d, d)$ Tucker rank algorithm.

**Dependence on $M$ (Stu9):** Our target regret bound grows polynomially in $M$ because the feature mapping we learn in the source phase has dimension $dM$ as we take the union of the feature mappings from each of the $m \in [M]$ source MDPs. As one has not interacted with the target MDP, they cannot identify which of the $dM$ dimensions of the learned feature mapping are unnecessary, and removing a necessary dimension will lead to a linear in $T$ regret bound. Thus, one must use the full $dM$ feature mapping to ensure the feature mapping is sufficient. We remark that our results hold for relaxed assumptions when the Tucker rank of the source MDPs are $(S, S, d)$ and the Tucker rank of the target MDP can be as large as $(S, S, dM)$, as the main assumption we need is only that the feature mapping in the target MDP can be represented by a linear combination of the feature mappings from the source MDPs.

Finally, we thank the reviewers for catching mistakes and typos and have fixed them in the final version.

---

### Decision · Program_Chairs · 2024-09-25

**Decision:**

Accept (poster)

**Comment:**

The paper studies the problem of transfer learning in RL under low-rank assumptions on the dynamics. While some of the reviewers raised concerns about the novelty of the contribution, there is general consensus about the fact that the main merit of the paper is to provide a comprehensive treatment of different low-rank settings and a somehow general and unified solution for them. Furthermore, for most of the assumptions considered in the paper, the results are minimax optimal. As such, I think this is an important contribution to the community and I believe this paper could become a go-to reference for this setting. Hence, my recommendation for acceptance.

I still strongly encourage the authors to revise the current manuscript:
* Fix all the typos identified by rev. LmTe. In particular, given that the paper reviews many different settings, it is easy to get "lost" in the different results of the paper. So it is really important to have very rigorous writing.
* Please include all the discussion and clarification from the rebuttal. For instance, the dependence on M may indeed by a source of confusion.